# Augmenting Biological Fitness Prediction Benchmarks with Landscapes Features from GraphFLA

**Mingyu Huang**[1]**, Shasha Zhou**[2]**, and Ke Li**[2]
[1]School of Computer Science and Engineering,
University of Electronic Science and Technology of China
[2]Department of Computer Science, University of Exeter
m.huang.gla@outlook.com; {sz484, k.li}@exeter.ac.uk

## Abstract

Machine learning models increasingly map biological sequence-fitness landscapes to predict mutational effects. Effective performance evaluation of these models demands comprehensive benchmarks curated from empirical data. Despite their impressive scale, existing benchmarks lack topographical information regarding the underlying fitness landscapes, which hampers interpretation and comparison of model performance beyond simple averaged scores. To address this, here we present `GraphFLA`, a Python framework that constructs and analyzes fitness landscapes from mutagenesis data in diverse sequence modalities (e.g., DNA, RNA, protein and beyond) with up to millions of mutants. `GraphFLA` calculates a holistic set of 20 biologically relevant features that characterize 4 fundamental aspects of landscape topography: ruggedness, epistasis, navigability and neutrality. By applying `GraphFLA` to over 5,300 empirical landscapes from ProteinGym, RNAGym, and CIS-BP, we demonstrate its utility in interpreting and comparing the performance of dozens of fitness prediction models, highlighting factors influencing model accuracy and respective advantages of different models. All the resources are available at `https://github.com/COLA-Laboratory/GraphFLA`.

## 1 Introduction

The fitness landscape is a nearly century-old foundational concept rooted in evolutionary biology [1] with profound implications on the understanding of biological principles in all 3 modalities of the central dogma (DNA, RNA, protein)—from drug resistance [2, 3], enzyme activity [4–8], protein stability and expression [9–11], RNA folding and function [12–16], to transcription factor binding [17, 18]. Efficiently and accurately mapping these fitness landscape surfaces is critical to enable various downstream tasks [19–23], and has been recently advanced by machine learning (ML) models that can capture complex and high-dimensional patterns of the sequence-fitness map [24–30].

A critical step in developing models is their proper performance evaluation to understand limitations and enable comparisons with existing ones. For this purpose, large-scale benchmarks have been established across different modalities. For example, ProteinGym [31] offers more than 250 tasks curated from deep mutational scanning (DMS) assays for proteins, while RNAGym [32] incorporates over 30 standardized RNA DMS assays. Considering their impressive scales and the famous "no free lunch" theorem [33], it is often unrealistic to expect one single model to dominate on *all* tasks. For instance, although the VenusREM model [24] yields the highest *average* score across all 217 DMS substitution tasks in ProteinGym, it leads in only $14$ ($6.5\%$) individual tasks. Meanwhile, $44$ out of the evaluated 89 models leads in *at least one* task. This reality prompts critical questions: **Q1:** *"Why did one model perform well on one set of tasks but poorly on another?"*, **Q2:** *"why did one model outperform baseline on one task, but not on the other?"*

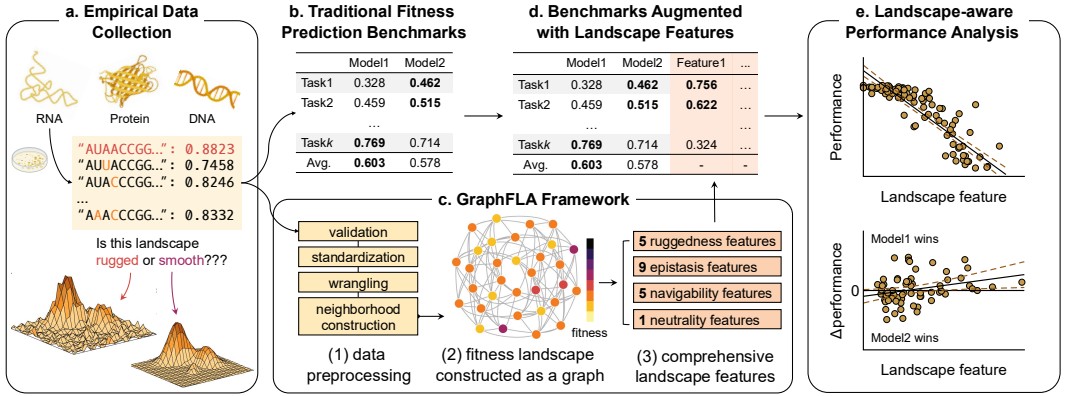

Figure 1: **Overview of how GraphFLA contributes to the performance benchmarking of fitness prediction models.** Existing biological fitness prediction benchmarks (**b**) are often curated from empirical fitness landscape datasets without interrogating landscape topography (**a**). `GraphFLA` constructs these landscapes and offers a comprehensive suite of features characterizing their topography (**c**). Such landscape features can then augment existing benchmarks (**d**) and thus assist performance interpretation (**e, upper**) and comparison (**e, lower**).

Answering these questions necessitates informative features that characterize each task. Unfortunately, existing benchmarks typically provide only basic labels (e.g., taxon) or statistics (e.g., sequence length), which are insufficient to fully elucidate the 2 questions above. Consequently, users often rely on average scores for decision-making and comparison, which can lead to biased conclusions.

For decades, evolutionary biologists have developed various features to quantitatively characterize topographical aspects of fitness landscapes, including ruggedness [4, 34, 35, 18], navigability [18, 4, 34, 17, 36], epistasis [37–42], and neutrality [43–45, 22]. These features have been extensively applied to unveil fundamental principles governing evolutionary dynamics [2–18]. As biological sequence models essentially aim to learn these landscape surfaces, we hypothesize that these same features can explain why models perform differently across tasks and address the previous 2 questions.

Yet, despite decades of study, landscape analysis remains a highly specialized biological field. As a result, standard open-source implementations for calculating many landscape features are rarely available. Also, existing research often targets specific landscape aspects, leaving no consensus on a comprehensive feature set. In addition, empirical landscapes span diverse biological modalities and scales, further complicating the development of unified analysis tools. The rapid growth in empirical data also demands highly scalable methods. Consequently, researchers currently lack accessible, broadly applicable tools for characterizing fitness landscape features in common benchmarking tasks.

To address this lack of analysis tooling, we present `GraphFLA`, a versatile, comprehensive, scalable and end-to-end Python framework for streamlining fitness landscape analysis. `GraphFLA` constructs fitness landscapes from biological sequence-fitness data in diverse modalities (including, but not limited to DNA, RNA, and protein) and is heavily optimized to scale to datasets with even millions of mutants. It is interoperable with established fitness prediction benchmarks by using an API and data format similar to that used for model training, and is essentially applicable to empirical data in other databases and in the literature. Once a landscape is constructed, `GraphFLA` offers a rich suite of 20 features compiled from thousands of papers characterizing 4 fundamental aspects of landscape topography: ruggedness, navigability, epistasis, and neutrality, which can then serve as biologically meaningful meta-features for each benchmark task to better interpret model performance.

We extensively compared `GraphFLA`'s scalability to existing tools and validated its reliability via a large-scale replication study using 155 *combinatorially complete* empirical landscapes collected from 61 works (Section 4.1), which are released as part of `GraphFLA`. We then demonstrated `GraphFLA`'s robustness to data missing, biased sampling, as well as noise in Section 4.2 with synthetic landscapes. To further demonstrate its versatility, we applied `GraphFLA` to analyze 5,300+ empirical landscapes from ProteinGym, RNAGym, and the CIS-BP database [46]. By employing landscape features from `GraphFLA` to interpret the performance of dozens of established models on these landscapes, we illustrate that: ▶ Model performance strongly depends on landscape topography; landscapes that are more rugged, epistatic, neutral, while less navigable, are harder for models to predict accurately (**Q1**;

Section 4.3); ▶ Different models, even with similar overall performance, can excel at different types of landscapes; performance gaps between them can change with specific landscape characteristics (**Q2**; Section 4.4). Finally, we showcase the wider utility of `GraphFLA` by applying it to analyze results of ML-guided directed evolution (MLDE) and phenotype landscapes in Section 4.5.

## 2   Background and Related Work

**Fitness landscapes.** In his pioneering work in 1932, Wright first described the concept of a fitness landscape by analogy to a physical landscape, where each spatial location represents a genotype, and the elevation indicates its fitness. Though this landscape metaphor is initially used to describe the genotype-fitness map, its influence quickly extended to other biological modalities and scales, e.g., molecules like RNA [12–16], proteins [47, 10, 11], genes [48, 49], and even communities [50–54].

**Landscape topography.** Since adaptation can be viewed as navigating fitness landscapes towards their highest peaks, their topography is essential for understanding the course of evolution. The most intuitive and widely studied aspect is **ruggedness** [4, 34, 35, 18], often characterized by the presence of multiple local optima (peaks). A necessary condition for landscape ruggedness is **epistasis** [38–42], which occurs when one or more mutations interact. In contrast, a purely additive landscape, where mutational effects are independent, would be fairly smooth with a single global optimum. Ruggedness along with pervasive epistasis can pose a fundamental challenge to an evolving population's ability to find the highest peak, thus reducing the landscape's **navigability** [18, 4, 34, 17, 36], another important topography aspect. Finally, many studies also interrogate **neutrality** [43–45, 22], which describes the presence of "plateaus" consisting of genotypes sharing the same fitness.

**Software packages for landscape analysis.** The only biological landscape analysis package known to us, `MAGALLEN` [55], offers several quantitative metrics but is limited in scope. In contrast, `GraphFLA` offers a holistic suite of 20 features covering all 4 fundamental aspects above. Also, while `MAGALLEN` is written in C, it can only rapidly handle landscapes at the scale of $10^5$ variants. `GraphFLA`, however, easily scales to landscapes of $10^7$. Furthermore, `MAGALLEN`'s pure-C implementation also hinders interoperability with modern ML ecosystems, unlike `GraphFLA`'s native Python API.

**Empirical fitness landscapes.** While early studies of landscape topography often relied on theoretical models (e.g., the NK model [56]), advancements in experimental methodologies have enabled the empirical assessment of increasingly large fitness landscapes [31, 57, 58]. These empirical landscapes are usually constructed by either ▶ randomly sampling a vast number of single- or multi-mutants for a wild-type (WT) sequence (e.g., [10–12]), or ▶ systematically assaying all possible sequences in a predefined space (e.g., [4, 7, 42, 59]). The first approach probes a fairly large area of the sequence space, but the resulting landscape is of narrow depth by containing only immediate neighbors of the WT. In contrast, the second approach generates *combinatorially complete* landscapes that allow exact analysis of topography and enable testing of model predictions on combined effects of mutations.

**Fitness prediction benchmarks.** Apart from driving biological insights, empirical landscape data also give rise to the wealth of benchmarking tasks for fitness prediction in different modalities [31, 32, 58, 60–64]. While earlier benchmarks like FLIP [58], [62] and [63] comprise only a handful of tasks, recent ones like ProteinGym [31] and RNAGym [32] now offer dozens to hundreds of tasks to enable more robust evaluation. Yet this scale also makes it harder to interpret the results, and users often abandon task-level scores and resort to averages [30, 63]. Though grouping scores based on basic task features (e.g., mutational type, taxon) or analyzing performance distribution can offer additional information [31, 32], they are not sufficient to fully address **Q1** and **Q2** that we previously posed. As a result, these benchmarks have not yet been fully leveraged. `GraphFLA` contributes augmenting them with fitness landscape features that enable biologically meaningful task-level analysis.

**Landscape analysis in other domains.** Landscape features have also been widely used to describe problem characteristics in black-box optimization (BBO). For example, the classic 24 BBO benchmarking functions included the `COCO` platform [65] are classified into 5 groups (from easy to hard) based on features like separability, modality, etc. Another R-package, `flacco` [66], offers 17 sets of features describing diverse characteristics of the optimization landscape. Similar features also exist for multi-objective optimization problems [67], and they can enable more informed algorithm testing, comparison, selection [68], and configuration [69]. Yet, all these features are designed for general continuous BBO problems. In contrast, `GraphFLA` is rooted in evolutionary biology for analyzing sequence-fitness landscapes, and goes beyond simple statistics to biologically meaningful ones.

Table 1: Collection of 20 essential landscapes features in `GraphFLA`

| Class | Index | Feature | Range | Higher value indicates |
|---|---|---|---|---|
| | F1 | Fraction of local optima | $[0, 1]$ | ↑ more peaks |
| | F2 | Roughness-slope ratio | $[0, \infty)$ | ↑ ruggedness |
| Ruggedness | F3 | Autocorrelation | $[-1, 1]$ | ↓ ruggedness |
| | F4 | Gamma statistic | $[-1, 1]$ | ↑ ruggedness |
| | F5 | Neighbor-fitness correlation | $[-1, 1]$ | ↓ ruggedness |
| | F6 | Magnitude epistasis | $[0, 1)$ | ↓ evolutionary constraints |
| | F7 | Sign epistasis | $[0, 1]$ | ↑ evolutionary constraints |
| | F8 | Reciprocal sign epistasis | $[0, 1]$ | ↑ evolutionary constraints |
| | F9 | Positive epistasis | $[0, 1]$ | ↑ synergistic effects |
| Epistasis | F10 | Negative epistasis | $[0, 1]$ | ↑ antagonistic effects |
| | F11 | Global idiosyncratic index | $[0, 1]$ | ↑ specific interactions |
| | F12 | Diminishing return epistasis | $[0, 1]$ | ↑ flat peaks |
| | F13 | Increasing cost epistasis | $[0, 1]$ | ↑ steep descents |
| | F14 | Pairwise epistasis | $[0, 1]$ | ↓ higher-order interactions |
| | F15 | Fitness-distance correlation | $[-1, 1]$ | ↑ navigation |
| | F16 | Glocal optima accessibility | $[0, 1]$ | ↑ access to global peaks |
| Navigability | F17 | Basin-fitness corr. (accessible) | $[-1, 1]$ | ↑ access to fitter peaks |
| | F18 | Basin-fitness corr. (greedy) | $[-1, 1]$ | ↑ access to fitter peaks |
| | F19 | Evol-enhancing mutation | $[0, 1]$ | ↑ evolvability |
| Neutrality | F20 | Neutrality | $[0, 1]$ | ↑ neutrality |

# 3  GraphFLA: A Framework for Fitness Landscape Analysis

The `GraphFLA` framework mainly consists of 3 parts (Fig. 1c): (1) data preprocessing, (2) landscape construction, and (3) landscape analysis. We purpose-built it to meet 4 key desiderata: ▶ **Applicability** across empirical landscapes from diverse biological modalities and scales. ▶ **Interoperability** with existing ML-ready data. ▶ **Scalability** to efficiently handle landscapes containing millions of genetic variants. ▶ **Extendability** to include new analysis methods via an unified API.

**Data input.** To ensure compatibility with existing fitness prediction benchmarks, `GraphFLA`'s API accepts the standard inputs used by typical ML frameworks. It takes a list of biological sequences (`X`) and their corresponding fitness values (`f`), which can be obtained from either random, site-saturation, or combinatorial mutagenesis, or other analogous design. `GraphFLA` supports sequences of length $n$ where each locus $i \in \{1, \dots, n\}$ can take distinct values from a predefined set $\mathcal{A}_i$ ($|\mathcal{A}_i| \geq 2$; e.g., for DNA sequences, $\mathcal{A}_i = \{A, C, G, T\}$). This general input form allows `GraphFLA` to handle data from diverse biological modalities, such as DNA, RNA and protein sequences, or single-cell profiles. We also include built-in classes optimized for common sequence types (DNA, RNA, proteins, and binary data) to enhance performance. Additionally, `GraphFLA`'s preprocessing pipeline automatically detects the composition of the sequence space, standardizes the input data, and identifies duplicates or missing values. This preprocessing ensures robust results in subsequent analyses.

**Neighborhood identification.** Next, `GraphFLA` determines a neighborhood structure, which specifies which input variants are genetically adjacent in the sequence space. To this end, traditional methods calculate genetic distances between all possible pairs of variants to find one-mutant neighbors [4, 55]. However, this pairwise calculation quickly becomes impractical because it requires quadratic time and memory resources. To overcome this scalability issue, `GraphFLA` employs a more efficient approach: instead of comparing every pair, it directly generates all potential single-mutation neighbors for each variant. This new strategy achieves nearly linear complexity and significantly outperforms existing implementations in both runtime and memory efficiency (Section 4.1).

**Landscape as a variant network.** Once the neighborhood is identified, `GraphFLA` constructs the fitness landscape as a directed, attributed graph (Fig. 1c), where each node represents a variant and is associated with its fitness; any two variants that are neighbors to each other are connected by an directed edge, which represents a single mutational step towards higher fitness. This graph representation, backend by the `igraph` package in C, allows many landscape analysis to be implemented via efficient graph mining algorithms for significant speed up. For example, locating local optima can be done by finding all *sinks* in this graph, while classifying different types of epistasis is equivalent to finding specific types of 4-node motifs (Appendix C).

Table 2: Summary of combinatorially complete datasets in `GraphFLA`

| Modality | Space | No. of Datasets | No. of Mutants |
|----------|-------|-----------------|----------------|
| DNA | Genomic sequence | 55 | 724k |
| Protein | Transcript sequence | 63 | 1.1M |
| RNA | Amino acid sequence | 37 | 348k |
| Total | | 155 | 2.2M |

**Landscape analysis.** For each constructed landscape, `GraphFLA` offers a comprehensive suite of 20 features characterizing their 4 fundamental topographical aspects: ruggedness, navigability, epistasis, and neutrality (Table 1). In selection of these features, we conducted an large language model-assisted, data-driven survey of 1, 673 papers on landscape analysis and evolutionary biology (Appendix B), which aims to identify all prevalemt quantitative indicators of landscape topography in literature. From more than 100 initial candidates, we compiled this final collection of 20 essential features based on their (1) frequency of appearance in literature, (2) biological significance, (3) coverage across different aspects, (4) computational feasibility, and (5) compatibility with data modalities, sizes, and structures (see details in Appendix B). A full introduction to these features is available in Appendix C

**Empirical and theoretical landscapes.** Beyond the main modules depicted in Fig. 1c, `GraphFLA` provides a data module featuring 155 empirical fitness landscapes that are combinatorially complete, covering more than 2.2M total sequence variants (Table 2; Table A3). These landscapes, gathered via another extensive literature survey (Appendix B), span multiple modalities (DNA, RNA, protein) and taxa with diverse fitness metrics. The comprehensive combinatorial nature of this collection distinguishes it from current benchmarks, which mainly consist of randomly generated mutagenesis libraries. This structure enables systematic evaluation of model predictions on combined mutations of varying orders and aids in interpreting results through landscape topography. In addition to these empirical landscapes, `GraphFLA` provides 5 theoretical models for generating synthetic landscapes with tunable characteristics (e.g., dimension, ruggedness; Appendix D).

## 4 Results

### 4.1 GraphFLA Enables Efficient and Accurate Landscape Analysis Across Modalities

**Runtime and memory scalability.** To evaluate the performance of `GraphFLA`, we generated synthetic fitness landscapes of varying sizes using the *NK* model by Kauffman. We then compared the scalability of landscape construction with that of the `MAGELLAN` package and the community implementation used in [4]. The benchmarks were conducted using a single core of an Intel Xeon Platinum 8260 CPU with 256GB RAM. Our findings in Fig. 2a show that `GraphFLA` is significantly faster in landscape construction compared to existing implementations. For example, the two baselines took more than 5h and 3h respectively to construct the landscape at the scale of 1 million mutants, whereas `GraphFLA` used only 20s. Regarding memory efficiency, `GraphFLA` stood out by requiring only 2GB memory to process 1 million mutants. In contrast, the community implementation encountered out-of-memory errors when handling over 100,000 mutants. In sum, `GraphFLA` scales almost linearly with landscape size, allowing it to process even the largest empirical fitness landscapes with millions of mutants.

**Application to diverse real-world data.** To demonstrate the versatility of `GraphFLA` in processing real-world data for fitness predictions, we applied it to 4 large-scale benchmarks and databases to construct 5,300+ landscapes in different modalities and sizes for different species. We then calculated the 20 landscape features introduced in Table 1 for each of them (Table A3, Table A4, Table A5).

- **ProteinGym.** We used its 217 DMS substitution tasks. Since the data is highly curated, `GraphFLA` can directly construct landscape for each task using the "`sequences`" and "`DMS_score`" columns, with the only preprocessing being removing the common genetic backgrounds. This results in 217 landscapes (2.2M total mutants) describing protein activity, binding, expression, stability, etc. We note that 168 of these landscapes only contain single mutants, and were excluded from subsequent analysis as there is little information beyond the local neighborhood of the WT.

- **RNAGym.** We used the 33 fitness prediction tasks from it, following the same procedure above. This leads to 31 landscapes (358k total mutants) for mRNAs, tRNAs, aptamers, and ribozymes.

- **Combinatorially complete landscapes.** We constructed 155 combinatorially complete using the datasets introduced in Table 2, which contain a mixture of DNA, RNA, and protein landscapes.

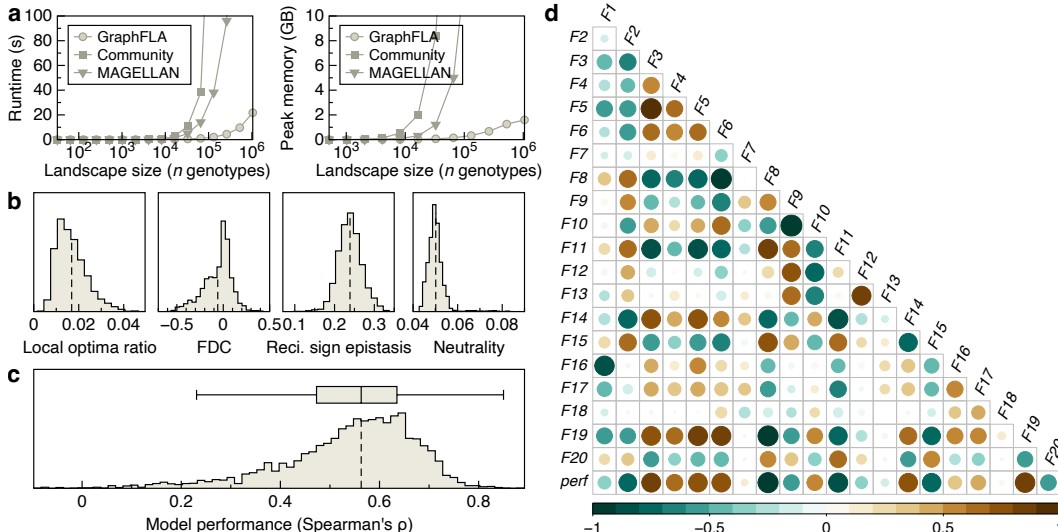

Figure 2: **GraphFLA scales efficiently and captures influential landscape features for model performance.** (a) Runtime (*left*) and peak memory usage (*right*) during fitness landscape construction for `GraphFLA`, `MAGELLAN`, and a community implementation [4], as a function of landscape size. Landscapes were generated using the *NK* model [70] by varying the number of loci $N$ from 5 to 20 ($\rightarrow$ landscape sizes from $2^5$ to $2^{20}$). Results shown are averages across 10 replicates. (b) Distribution of 3 representative landscape features across 155 combinatorially complete landscapes collected in `GraphFLA`. (c) Distribution of model performance, measured by Spearman's $\rho$, for Evo2 predictions across the same 155 landscapes. (d) Correlation matrix showing Spearman's $\rho$ between 20 landscape features derived from `GraphFLA` and Evo2 performance across all 155 combinatorial landscapes.

- **CIS-BP database.** It harbors protein-binding microarray (PBM) data for 5,016 TFs of 329 eukaryotic species and 162 DNA-binding domain structural classes from 78 studies [46]. For each TF, the fitness is its binding affinity to all 32,896 possible 8-nucleotide, double-stranded DNA sequences. `GraphFLA` constructed these 5,016 TF binding landscapes with 174M total mutants.

**Validation with existing literature.** We proceeded to assess the precision of `GraphFLA`'s landscape analysis by performing a large-scale replication study across the 61 papers from which our 155 combinatorially complete datasets originate. For each publication, we identified its reported qualitative (e.g., "highly navigable" [4]) and quantitative (e.g., "514 peaks" [4]) landscape characteristics. We then utilized the relevant metrics within `GraphFLA` to re-analyze these landscapes, and compared our outputs against the original findings. `GraphFLA` successfully replicated the qualitative conclusions from all 61 studies (full results in Table A3). Notably, for features with unique definitions such as $\phi_{lo}$ and $\epsilon_{reci}$, `GraphFLA` precisely reproduced the published values if data processing details were sufficiently described. For features with more generalized definitions for which implementations can vary (e.g., $\epsilon_{DR}$), `GraphFLA`'s analysis consistently supported the conclusions drawn in the original studies. These results demonstrated that `GraphFLA` is a reliable framework for landscape analysis.

## 4.2   GraphFLA is Robust to Incomplete, Noisy, and Biasedly Sampled Data

A crucial validation for any analysis framework is quantifying its robustness to imperfect data. Real-world empirical landscapes often suffer from (a) missing variants, (b) noise in fitness measurement, or (c) biased sampling in generating the mutant library. In order to validate `GraphFLA`'s rosbutness to these, we conducted experiments on a *complete NK* landscape with moderate size ($n = 15$, thus $2^{15}$ total variants) and ruggedness ($k = 7$), which serves as a reference for the most ideal data.

**Robustness to incomplete data.** We created *incomplete* landscapes by randomly removing a fraction $\alpha = \{10\%, 20\%, 50\%\}$ variants from the reference landscape. We then calculated four representative landscape features for these and the reference landscape in Table 3. The results shows that most key features are highly robust to data incompleteness. The one exception is global optima accessibility, which, as expected, decreases as more data is removed, since this will destroy paths leading to the global optima regardless of their evolutionary accessibility. Yet, this effect is predictable and can be partially corrected by scaling the measured accessibility by the fraction of remaining data $(1 - \alpha)$.

Table 3: Essential landscape metrics for reference, incomplete, noisy, and biased $NK$ landscapes.

| Setting | Reciprocal sign epistasis | Global optima accessibility | Autocorrelation | FDC |
|---|---|---|---|---|
| **reference (complete)** | **0.1885** | **0.6729** | **0.1151** | **-0.0313** |
| incomplete (10%) | 0.1883 | 0.6103 | 0.0927 | -0.0420 |
| incomplete (20%) | 0.1884 | 0.5276 | 0.0771 | -0.0337 |
| incomplete (50%) | 0.1774 | 0.3223 | 0.0553 | -0.0313 |
| noisy ($0.01\sigma$) | 0.1889 | 0.6492 | 0.0965 | -0.0314 |
| noisy ($0.05\sigma$) | 0.1896 | 0.6542 | 0.0927 | -0.0317 |
| noisy ($0.1\sigma$) | 0.1921 | 0.6362 | 0.0966 | -0.0319 |
| noisy ($0.2\sigma$) | 0.1984 | 0.6339 | 0.0867 | -0.0414 |
| biased (random mutagenesis) | 0.1823 | 0.7246 | 0.1208 | -0.0837 |

**Robustness to noisy data.** To simulate experimental noise, we added random noise drawn from a Gaussian distribution $\mathcal{N}(0, (\beta\sigma)^2)$ to the variant fitness in the reference landscape, where $\sigma$ is the standard deviation of the original fitness values and the noise level $\beta$ was set to $\{0.01, 0.05, 0.1, 02\}$. The results in Table 3 demonstrate that all four landscape features remain remarkably stable. Even with noise equivalent to $0.2\sigma$, the calculated values are consistent with the reference. This highlights that GraphFLA's feature calculations are resilient to typical levels of experimental noise.

**Robustness to biased sampling.** We then simulated a more realistic scenario of random mutagenesis, which often creates a library that is densely sampled near a wild-type sequence but sparse elsewhere. We created a sparse, *biased* library of 1,804 variants (from 32,768 total) by applying a 10% per-site mutation rate to the global optimum. As shown in Table 3, the key landscape features remain highly consistent with the reference landscape, even when calculated on this much smaller, non-uniform subset. This implies that GraphFLA can still provide reliable approximation of the overall landscape topography even with only biasedly and sparsely sampled data.

## 4.3 GraphFLA Identifies Key Influence Factors and Bottlenecks in Fitness Prediction

The $5,300$ empirical landscapes we constructed in Section 4.1 exhibit significant variation in their topography. For instance, across the 155 combinatorially complete landscapes, the percentage of local optima ranges from around $0\%$ to $5\%$ (Fig. 2b). While the lower bound corresponds to a fairly smooth, unimodal landscape, the upper bound is on par with that of the most rugged *NK* landscapes [71]. Similar observations can be made for other landscape features (Fig. 2b; Fig. A1) and for ProteinGym (Fig. A2) as well as RNAGym (Fig. A3). From a performance benchmarking perspective, this is a good sign since it implies that the included tasks are diverse enough to "stress-test" models [72].

To illustrate how landscape features can shed light on fitness prediction performance, and thus our **Q1** in Section 1: *"Why did one model perform well on one set of tasks but poorly on another?"*, we used Evo2-7b [27], the successor of Evo [63], as an example. It is trained on 9.3 trillion DNA base pairs and applicable to diverse modalities including DNA, RNA, and protein. We applied Evo2 to each of our 155 combinatorial landscapes, and assessed its performance in fitness prediction using Spearman's $\rho$ as in prior works [73, 60, 74, 61, 75]. We found that Evo2's performance varies significantly across landscapes (Fig. 2c), and is highly dependent on landscape features (Fig. 2d). Specifically, half of the landscape features yielded Spearman's $|\rho|$ higher than $0.6$ with Evo2's performance, and 6 revealed moderate correlation ($0.3 < |\rho| < 0.6$). Similar results can be obtained by considering partial correlations controlling for landscape size. By taking a closer look at how Evo2's performance varies with landscape features Fig. 3a, we found that it struggles on landscapes that are:

- **More rugged** and **more epistatic**. In such landscapes, fitness values often fluctuate dramatically even in local regions within a small genetic distance (low $\rho_a$ and NFC). Extrapolation of such landscapes, even across only a single mutation, may fail due to the existence of local epistatic hotspots (often local optima) resulting from high-order (indicated by low $\epsilon_{(2)}$), non-magnitude epistasis (indicated by high $\epsilon_{\text{reci}}$) between sites that are unique to the current landscape.

- **Less navigable.** Benign landscapes that are easy to navigate and predict have strongly negative fitness distance correlation (FDC)—variants closer to the global peak in genetic distance tend to have higher fitness. In contrast, when FDC becomes closer to zero or positive, this information diminishes and even becomes "deceptive" (e.g., fitness declines as approaching global peak).

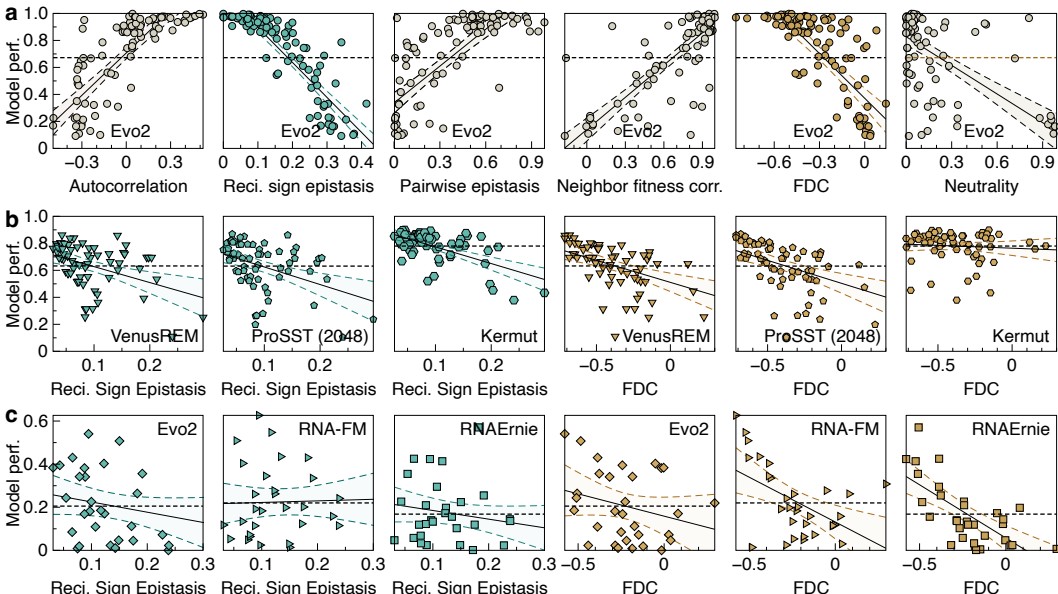

Figure 3: **GraphFLA identifies influencing factors for model performance.** For **(a)** our 155 combinatorial landscapes, **(b)** ProteinGym, and **(c)** RNAGym, we plot the distribution of model (name specified in each plot) performance ($y$-axis; measured as Spearman's $\rho$) against landscape features ($x$-axis). Straight lines show a fit of the linear regression model, and shaded regions depict the 95% confidence intervals. Dashed horizontal lines indicate the average performance across all landscapes.

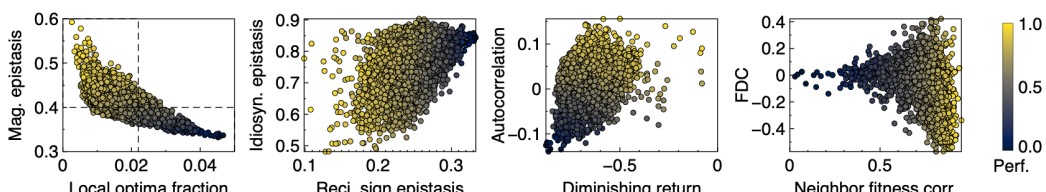

Figure 4: **Visualizing the distribution of model performance in landscape feature space.** We map each of the 5,016 landscapes constructed from the CIS-BP data in the space composed of landscape features and color-coded with the performance (Spearman's $\rho$) of Evo2-7b to visualize its distribution in the feature space.

- **Highly neutral.** These landscapes feature abundant "plateau" regions where mutations have zero fitness effects (i.e., neutral), which can hardly be predicted by models trained on non-neutral data.

While such findings were drawn from landscapes with heterogeneous modalities and a general model, similar patterns can be observed in more specific settings. For example, current leading zero-shot models on ProteinGym's 217 DMS substitution tasks, VenusREM [24], ProSST ($k = 2,048$) [25], and the leading supervised model, Kermut [26], tend to excel at fitness prediction for benign protein landscapes with FDC $< -0.5$ and $\epsilon_{\text{reci}} < 0.1$, yet still struggles for more complex ones (Fig. 3b; more models and features in Fig. A8, Fig. A9). Established models for RNA fitness prediction like RNA-FM [28] and RNAErine [29] exhibited the same behavior (Fig. 3c; more in Fig. A10, Fig. A11).

To see this at a larger scale, we evaluated the performance of Evo2-7b on the 5,016 CIS-BP TF binding landscapes described in Section 4.1. We plotted each landscape instance in the feature space shown in Fig. 4, and mapped Evo2's performance to this space. From the results, we can observe a clear trend that landscapes located in certain regions of the feature space are in general harder to predict. For example, in the first panel, Quadrant II is occupied by landscapes featuring both a large number of local optima ($\phi_{\text{lo}}$) and abundant non-magnitude epistasis ($\epsilon_{\text{reci}}$), and Evo2 can hardly identify the true fitness rank in them despite extensive pre-training. On the other hand, landscapes belonging to Quadrant IV are much more benign with low $\phi_{\text{lo}}$ and $\epsilon_{\text{reci}}$. For such landscapes, Evo2 typically achieved Spearman's $\rho > 0.5$. As for Quadrant I and III, landscapes in these regions contain

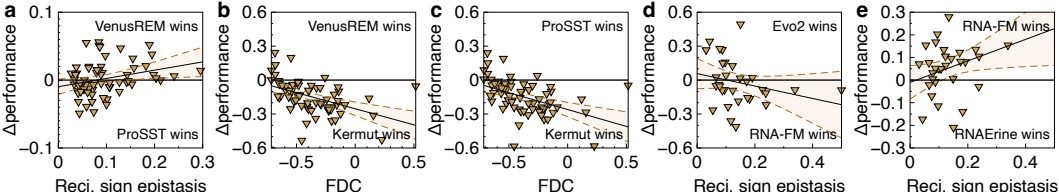

Figure 5: **GraphFLA facilitates landscape-aware model comparison.** Difference in performance ($y$-axis) between 5 pairs of baselines in ProteinGym (**a, b, c**) and RNAGym (**d, e**) is plotted against landscape features on the $x$-axis. Line regression fit lines and 95% confidence intervals are depicted.

a combination of complex and benign landscapes, which gave rise to mixed performance outcomes. The same trend can be observed for other combinations of features in other panels of Fig. 4.

## 4.4 GraphFLA Facilitates Landscape-aware Model Comparison

Another important question we asked is **Q2:** "*why does one model outperform the baseline on one task, but not on the other?*" For example, though VenusREM and ProSST ($k = 2,048$) have similar (Spearman's $\rho$: 0.518 vs 0.507) zero-shot performance on ProteinGym, they lead on 53% and 47% of tasks, respectively. Without further information on each task it is hard to strictly distinguish them.

Here we demonstrate that `GraphFLA`'s landscape features can shed light on their respective advantages. Fig. 5a plots the $\Delta$performance between VenusREM and ProSST on each task against the $\epsilon_{\text{reci}}$ of the corresponding landscape. We found that on benign landscapes with little reciprocal sign epistasis ($\epsilon_{\text{reci}} < 0.1$), ProSST tends to outperform VenusREM (Wilcoxon signed-rank test, $w = 123, p = 0.003$). Yet this advantage diminishes as $\epsilon_{\text{reci}}$ increases, as indicated by a positive slope of the linear regression line in Fig. 5a. Eventually, for landscapes with $\epsilon_{\text{reci}} > 0.15$, VenusREM consistently outperforms ProSST, which implies it is better at capturing complex epistatic interactions.

More intriguingly, both these zero-shot models can only outperform Kermut, the leading supervised baseline, on highly navigable landscapes (FDC $\approx -0.7$; Fig. 5b, c). As landscapes become less navigable (i.e., FDC increases), the performance gap between VenusREM (or ProSST) and Kermut increases drastically (Fig. 5b, c). Notably, for the ODP2 landscape from [76], which has an FDC = 0.23, Kermut outperforms VenusREM by a Spearman's $\rho$ of 0.53. The same pattern can be observed if we replace Kermut with other supervised baselines like ProteinNPT [77] (Fig. A12). These highlight that supervised training is still necessary to better extrapolate on complex landscapes.

As for RNAGym, though Evo2 and RNA-FM achieved comparable prediction performance across all analyzed landscapes (Fig. 3c), the former falls short on landscapes with high incidence of reciprocal sign epistasis ($\epsilon_{\text{reci}} > 0.2$; Fig. 5d). Also, the performance gap between RNA-FM and RNAErine increases as the landscape becomes more epistatic ($\epsilon_{\text{reci}}$ increases; Fig. 5e). These results shed new light on the respective of different models beyond simple averaged scores.

## 4.5 GraphFLA is Applicable to a Broader Range of Tasks and Data

**Application to directed evolution.** Beyond fitness prediction, `GraphFLA` can also be employed to shed light on other tasks related to fitness landscapes. For example, as directed evolution (DE) [78–81], a central technique in protein engineering, is essentially an adaptive walk on the protein fitness landscape to find high fitness variants, landscape topography can have fundamental impact on its success. For instance, DE on rugged landscapes is notoriously difficult [82]. However, a comprehensive understanding of the impact of ruggedness, and other landscape features, on DE, has been missing due to the lack of (1) holistic landscape analysis frameworks like `GraphFLA`, and (2) large collection of combinatorially complete empirical landscapes like in Table 2.

To demonstrate how `GraphFLA` can provide insights into DE, we used 20 protein landscapes from our combinatorial library that are 3- or 4-site-saturated (i.e., total variants being $20^3$ or $20^4$). For each of them, we evaluated the performance of 5 DE classic approaches: (1) the simplest DE, implemented as a greedy adaptive walk; (2) ML-guided DE (MLDE) [82, 83]; (3) MLDE warm-started with a zero short model [80]; (4) Active learning-guided directed evolution (ALDE) [84, 85]; (5) ALDE with zero-shot warm start. Detailed implementations are available in Appendix F. For each approach, we

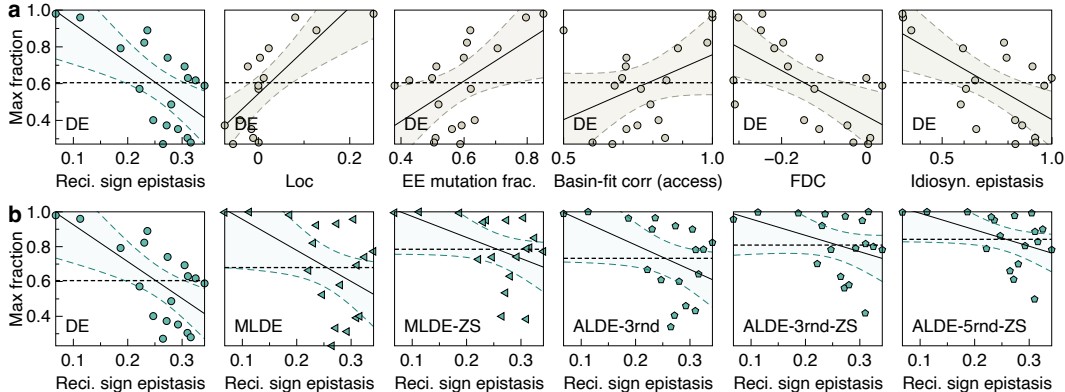

Figure 6: **GraphFLA can be employed to interpret directed evolution (DE) outcomes. (a)** The maximum fitness achieved (in percentile) by DE on each selected landscape against landscape features. **(b)** We fixed the $x$-axis to be reciprocal sign epistasis and replaced DE with 5 other ML-guided methods. Dashed horizontal lines indicate the average performance across all landscapes.

measured performance using the fitness percentile (where 1 indicates finding the global optimum) of the best variant it found, and aggregated across 100 randomly initialized runs. For the two ALDE methods, we additionally set the number of iterations to 3 or 5 rounds.

From Fig. 6a, we can see that while the basic DE method can easily find variants with fitness close to the global optimum on benign landscapes, it struggles on ones that are more rugged and epistatic, and less navigable. The 5 ML- or active learning-guided approaches are also susceptible to the incidence of epistasis, but more advanced approaches—such as MLDE with a zero-shot warm start and the ALDE variants—demonstrated greater robustness and are less adversely affected (Fig. 6b).

**Application to other data.** Beyond molecular sequences such as DNA, RNA, and proteins, `GraphFLA` can be applied to fitness landscapes at various other biological scales. For instance, it can be utilized to analyze evolutionary landscapes of gene regulatory networks [48, 49] or metabolic landscapes [86] at the cellular level. It can also analyze how alterations in community composition impact collective functions [50–54]. As a demo, we analyzed 6 microbial community-function landscapes in Table A2. In addition to traditional fitness landscapes, `GraphFLA` can be adapted to study a broad array of phenotype landscapes, a.k.a, genotype-phenotype (GP) maps, for RNA secondary structure [87], protein tertiary structure [88, 89], and protein complexes [90], etc. We provide demonstrations for these using computational models and 3 phenotype landscape features in Appendix E.

## 5   Conclusion

`GraphFLA` addresses the critical lack of meaningful features for interpreting performance benchmarking results in sequence fitness prediction. Using its comprehensive suite of 20 features describing the underlying landscape topography, we are now able to answer questions like *"why model performance varies across tasks?"*, *"when and why will a model outperform the other?"*. In this way, `GraphFLA` augments current benchmarks like ProteinGym and RNAGym to fully take advantage of their impressive scales, and assists in obtaining granular understanding of the capabilities and limitations of existing genomic models that were previously impossible. In addition, since `GraphFLA` itself is designed for arbitrary combinatorial landscapes, we expect it will be a useful resource for advancing our understanding on a broader range of tasks and data that is related to combinatorial optimization.

Future work could be done to see how landscape features can enable more principled selection or development of models, or how they may enable insights regarding other intriguing aspects that are unexplored in this paper (e.g., model scaling [91]).

**Acknowledgements.** This work was supported by the UKRI Future Leaders Fellowship under Grant MR/S017062/1 and MR/X011135/1; in part by NSFC under Grant 62376056 and 62076056; in part by the Royal Society Faraday Discovery Fellowship (FDF/S2/251014), BBSRC Transformative Research Technologies (UKRI1875), Royal Society International Exchanges Award (IES/R3/243136), Kan Tong Po Fellowship (KTP/R1/231017); and the Alan Turing Fellowship.

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

# A    Limitations

While `GraphFLA` provides extensive quantitative features for characterizing fitness landscapes, effectively visualizing their topography remains challenging due to the inherent high dimensionality and associated curse of dimensionality [92]. Dimensionality reduction methods, such as PCA [93], t-SNE [94], and UMAP [95], partially address this issue and have been effectively applied to diverse biological data [96–99]. However, these methods risk generating misleading visualizations of fitness landscapes. Specifically, compressing data into fewer dimensions inevitably leads to loss of information, potentially distorting spatial relationships among variants. Although this might be acceptable for general visualization purposes—where overall data trends remain intact—such distortions can result in the incorrect identification of local optima that do not exist in the original high-dimensional space. Consequently, despite the intuitive appeal of visualizing fitness landscape topography, `GraphFLA`, along with much of the landscape analysis literature, emphasizes quantitative metrics that inherently capture patterns within high-dimensional spaces.

Additionally, despite extensive optimization efforts, the scale of landscapes analyzable by `GraphFLA` within practical computational times remains small compared to the entire genotype space. For instance, the number of potential RNA sequences of length $n = 100$ is $4^{100}$, vastly exceeding the number of atoms in the observable universe. Nonetheless, by efficiently handling landscapes containing millions of variants, `GraphFLA` aligns well with current experimental capabilities, and can comfortably accommodate even the largest empirically measured fitness landscapes.

# B    Data-driven Literature Survey on Landscape Analysis

Building on our prior success in leveraging data-driven methods and large language models (LLMs) to enhance literature comprehension [100–102], we adopted a similar strategy in the development of `GraphFLA`. This involved conducting an extensive literature survey designed to: (1) identify key landscape features that characterize their topography, and (2) gather empirical data for combinatorially complete fitness landscapes. The methodology encompassed several distinct stages:

## B.1    Initial Literature Collection and Filtering

**Search Query Formulation.** The initial step involved crafting a targeted search query to retrieve literature pertinent to fitness landscapes. This topic is central to research where landscape features and combinatorially complete datasets are developed or utilized. The formulated query was:

> *(("fitness landscape*" OR "adaptive landscape*" OR "genotype network" OR "genotype-phenotype map*") OR ("epistasis" OR "diminishing return*" OR "increasing cost*" OR "NK landscape*"))*

This query is structured with two primary components:

1. The first component utilizes established terminology for fitness landscapes (e.g., "fitness landscape", "adaptive landscape") and associated concepts (e.g., "genotype network", "genotype-phenotype mapping") to ensure a broad capture of relevant studies.
2. The second component augments the search by incorporating specific terms frequently used in biological landscape analysis, such as "epistasis" and "NK landscape".

While other relevant concepts exist (e.g., local optima, $r/s$ ratio, see Appendix C), terms like "local optima" are prevalent across diverse optimization fields, making them less specific. Similarly, terms like "$r/s$ ratio" can be challenging for effective textual matching. Consequently, these were excluded from the initial query. This search strategy deliberately prioritized high recall, acknowledging that it might retrieve studies from related domains (e.g., energy landscapes in physics/chemistry [103], optimization landscapes in evolutionary computation [104]). Distinguishing these fields solely through keywords is often infeasible due to their broad scopes; therefore, subsequent filtering steps were planned to refine the selection for biological relevance.

**Database Search.** The formulated search query was executed on the Web of Science database[1]. This platform was chosen for its comprehensive coverage of peer-reviewed literature and high-quality

---

[1]https://www.webofscience.com

metadata. To enhance the precision of the initial retrieval and minimize noise from full-text searches (such as incidental mentions of keywords), the search scope was restricted to the title, abstract, and author keywords fields. These sections typically encapsulate the core subject matter of a publication.

**LLM-based Filtering.** This initial search yielded a substantial corpus of $31,784$ potentially relevant publications. To manage this volume and efficiently identify studies most pertinent to our research scope, we employed an LLM—specifically GPT-4o-mini—for automated initial screening. The title and abstract of each publication were processed by the LLM, which was prompted to classify the study based on two primary criteria:

1. Does the publication investigate fitness landscapes or closely related concepts specifically within biological systems?

2. If the answer to the first criterion is affirmative, does the publication report on empirical data, as opposed to being a purely theoretical analysis, *in-silico* simulation study, or review article?

This LLM-driven filtering process significantly narrowed the candidate pool. After applying the first criterion, the number of papers was reduced to $11,098$. The second criterion further refined this set to $1,673$ publications. This curated collection of papers, focusing on analysis of empirically measured fitness landscapes in biological systems, formed the basis for subsequent landscape feature set construction and the collection of combinatorially complete landscape data.

## B.2 Landscape Feature Set Construction

Following the identification of $1,673$ core publications relevant to empirical fitness landscape analysis, we proceeded to construct a comprehensive and representative set of landscape features. The objective was to distill a manageable yet informative collection of quantitative indicators that capture the fundamental topographical aspects of fitness landscapes.

**Initial Feature Candidate Identification.** The full texts of the $1,673$ curated papers were systematically reviewed to identify all quantitative measures used to describe landscape topography. This extensive survey, augmented by an LLM (GPT-4o) to scan for mentions and definitions of landscape metrics, initially yielded a broad list of over 100 candidate features. These candidates encompassed a wide range of mathematical formulations, statistical measures, and network-based properties that researchers have employed to characterize landscapes.

**LLM-assisted Feature Filtering and Selection.** To refine this extensive list into a practical and impactful feature set, we devised a set of carefully crafted criteria to guide selection:

1. **Empirical prevalence in literature:** How frequently is the feature used or discussed in the surveyed $1,673$ papers? Features with high prevalence were prioritized as they represent established and widely accepted indicators. The number of local optima is employed in $45\%$ of our analyzed literature.

2. **Biological significance:** Does the feature provide meaningful insights into evolutionary processes or other biological phenomena? Features with clear connections to biological interpretations were favored. For instance, features quantifying aspects like diminishing returns epistasis can shed light on the rate of adaptation [38], while measures related to neutrality can have great biological implications for understanding mutational robustness [43].

3. **Coverage across different topographical aspects:** Does the feature contribute to characterizing one of the 4 fundamental topographical aspects: ruggedness, navigability, epistasis, and neutrality? We aimed for a balanced set that provides a holistic view of the landscape.

4. **Computational feasibility:** Can the feature be computed efficiently for landscapes of varying sizes and complexities, such as those included in `GraphFLA`? Features requiring prohibitive computational resources for typical dataset sizes were deprioritized. For example, while the Walsh-Hadamard transform [105] can be used to calculate a full spectrum of epistatic interactions, its computational demand can be prohibitive for large landscapes.

5. **Compatibility with data modalities, Sizes, and Structures:** Is the feature applicable to the types of data commonly found in empirical fitness landscapes? Is it robust to missing data or variations in landscape size? Features with broad applicability and robustness were preferred.

**Final feature set.** By manually reviewing the initial set of features along with expert consultations, we arrived at the final set of 20 essential landscape features presented in Table 1, which cover all 4 fundamental aspects of landscape topography and are extensively used in landscape analysis literature to offer different biological insights. They can also be efficiently computed for empirical landscapes with different modalities and sizes, and are applicable to non-complete landscapes. A full introduction to each of these 20 features, including their definitions, interpretations, and computational considerations, is available in Appendix C.

## B.3 Combinatorially Complete Landscape Collection

In GraphFLA, we focused on collecting combinatorially complete datasets derived from extensive mutagenesis studies. Unlike datasets generated by randomly sampling mutants around a wild-type sequence, combinatorially complete landscapes encompass measurements for *all* possible genotypes within a defined genotype space. We identified and collected such datasets from our focused set of $1,673$ papers via manual scrutiny of the full-texts with the aid of GPT-4o. Specifically, for each paper, we asked:

- Does the publication publish new empirical landscape data?
    - $\rightarrow$ If yes, was the published data combinatorially complete? This criterion specifically excludes datasets focusing only on evolutionary trajectories (i.e., monitoring changes in population mean fitness and genotypic composition) or deep mutational scanning data (i.e., sampling only mutants closely related to a wild-type sequence).
    - $\rightarrow$ If no, did it mention or use combinatorially complete datasets from previous works?

After this final review, we arrived at a total of 155 datasets as listed in Table A3, sourced from more than 67 studies. This number is much smaller compared to the initial corpus of over $30,000$ papers. The main reason for this is such combinatorially complete landscapes are extremely costly to construct. Consequently, they are regarded as extremely valuable resources and are extensively utilized in subsequent works for both deriving biological insights [106, 107] and testing ML systems [85, 108, 58].

## B.4 Datasets Processing.

Here we describe a few standards we applied when preparing these data.

**Naming convention.** We established a systematic naming convention to uniquely and informatively identify each dataset within the collection, following these guidelines:

- **Base identifier:** The core of the name typically consists of the first author's last name concatenated with the four-digit publication year (e.g., "Papkou2024").

- **Common name suffix:** If a dataset is widely recognized by a common identifier, often related to the specific biological system or molecule studied, this identifier may be appended as a suffix for easier recognition. For instance, the study by Wu et al. [59] investigated the fitness landscape across $20^4 = 160,000$ variants at four sites (V39, D40, G41, V54) within protein G domain B1 (GB1). This landscape is commonly referred to as "GB1," and thus the dataset might be named incorporating this suffix (e.g., "Wu2016_GB1").

- **Disambiguation suffixes:** Additional suffixes are employed when a single publication or study system yields multiple distinct datasets. These suffixes serve to differentiate datasets based on key experimental variables, such as:
    - Different subjects and fitness measures (e.g., Phillips et al. [109] studied binding affinities of different variants of antibodies CR9114 and CR6261 against various influenza HA antigens like H1, H3, etc., which results in separate datasets per antibody-antigen pair).
    - Variations in experimental conditions or environments (e.g., Soo et al. [110] measured the self-splicing activity of $4^8 = 65,536$ Tetrahymena intron variants at two different temperatures, 30°C and 37°C, leading to two distinct datasets).
    - Exploration of different mutation sites or regions (e.g., Johnston et al. [7] generated the "TrpB3D" landscape from $20^3 = 8,000$ variants at sites {T117, A118, A119}

of the thermostable tryptophan synthase $\beta$-subunit (TrpB), and the distinct "TrpB3E" landscape based on sites {F184, G185, S186}).

– A combination of the above factors.

These distinguishing characteristics are systematically incorporated into the dataset name as further suffixes to ensure clarity and uniqueness.

**Search space representation.** Genotypes within each dataset are presented using two formats: a sequence-based representation (e.g., "ATTA") and a vector explicitly listing the allele at each locus (e.g., ["A", "T", "T", "A"]). The combinatorial nature of these representations leads to vast theoretical search spaces. For a sequence of length $L$, the total number of possible genotypes is $4^L$ for DNA or RNA, $20^L$ for proteins, or $2^L$ for binary representations. Binary representations typically indicate the presence or absence of specific mutations and can be applied to DNA, RNA, proteins, or other biological systems, such as microbial communities [52, 111]. Although the theoretical search spaces for our collected landscapes are combinatorially complete by design, the experimentally generated data often exhibit incomplete coverage due to experimental constraints or subsequent filtering steps. For example, the DHFR landscape from Papkou et al. [4] measured fitness for $261,382$ variants, which constitutes $99.7\%$ of the total $4^9 = 262,144$ possible genotypes. Similarly, the GB1 protein landscape reported by Wu et al. [59] includes $149,361$ variants, covering $93.4\%$ of the theoretical space of $20^4 = 160,000$ sequences.

**Fitness measure.** Following our naming convention, each of the $155$ dataset listed in Table A3 focuses on a single fitness measure under a given condition (e.g., environment). To ensure the potential for accurate landscape reconstruction and the replication of published analysis results, we have retained the original fitness values as reported in the source publications. No transformations were applied to these values unless such transformations were already part of the published dataset. For datasets where fitness variance across replications is available, this is amended as an additional column.

## C Fitness Landscape Features

### Core Definitions

We begin by defining fundamental concepts used throughout the fitness landscape analysis.

**Definition C.1** (Alleles and Loci). *Let $n$ be the number of polymorphic loci under consideration. For each locus $i \in \{1, 2, \ldots, n\}$, let $\mathcal{A}_i$ denote the set of distinct alleles present. The number of alleles at locus $i$ is $m_i = |\mathcal{A}_i| \geq 2$.*

**Example 1.** *For example, in complete DNA or RNA landscapes, $\mathcal{A}_i$ typically contains four nucleotide bases ($\{A, C, G, T\}$ for DNA, $\{A, C, G, U\}$ for RNA), thus $m_i = |\mathcal{A}_i| = 4$. For proteins, where loci usually represent amino acid positions, $\mathcal{A}_i$ comprises the 20 standard amino acids, resulting in $m_i = |\mathcal{A}_i| = 20$.*

**Definition C.2** (Genotype Space). *A genotype $g$ is a specific combination of alleles across all $n$ loci, represented as a sequence $(a_1, a_2, \ldots, a_n)$ where $a_i \in \mathcal{A}_i$ for each $i$. The set of all possible genotypes constitutes the genotype space $\mathcal{G}$. The total number of genotypes is $|\mathcal{G}| = \prod_{i=1}^{n} m_i$.*

**Definition C.3** (Hamming Distance). *The Hamming (genetic) distance $d_H(g, g')$ between two genotypes $g = (a_1, \ldots, a_n)$ and $g' = (a'_1, \ldots, a'_n)$ is the number of loci at which their alleles differ:*

$$d_H(g, g') = \sum_{i=1}^{n} \mathbb{I}(a_i \neq a'_i),$$

*where $\mathbb{I}(\cdot)$ is the indicator function ($\mathbb{I}(\text{condition}) = 1$ if the condition is true, $0$ otherwise).*

**Definition C.4** (Mutational Neighborhood). *The neighborhood $\mathcal{N}(g)$ of a genotype $g \in \mathcal{G}$ is the set of all genotypes reachable from $g$ by a single point mutation, i.e., genotypes $g'$ differing from $g$ at exactly one locus:*

$$\mathcal{N}(g) = \{g' \in \mathcal{G} \mid d_H(g, g') = 1\}.$$

*The size of the neighborhood is $|\mathcal{N}(g)| = \sum_{i=1}^{n} (m_i - 1)$.*

**Definition C.5** (Fitness Function). *The fitness function $f : \mathcal{G} \to \mathbb{R}$ assigns a scalar fitness value $f(g)$ to each genotype $g \in \mathcal{G}$.*

**Definition C.6** (Mutant Genotype Notation). *Let $g = (a_1, \ldots, a_j, \ldots, a_n)$. We denote the specific single-mutant neighbor resulting from changing the allele $a_j$ at locus $j$ to a different allele $a'_j \in \mathcal{A}_j \setminus \{a_j\}$ as $g_{[j \leftarrow a'_j]}$. When the specific mutated allele $a'_j$ is not critical or is clear from context (e.g., in biallelic systems where the alternative allele is unique), we may use the shorthand $g_{[j]}$ to denote any single mutant differing from $g$ only at locus $j$.*

**Definition C.7** (Selection Coefficient). *Let $g$ be a genotype with allele $a_j$ at locus $j$. The selection coefficient $s_{j, a_j \to a'_j}(g)$ measures the fitness effect of mutating allele $a_j$ to $a'_j$ (where $a'_j \in \mathcal{A}_j \setminus \{a_j\}$) at locus $j$ within the genetic background of $g$:*

$$s_{j, a_j \to a'_j}(g) = f(g_{[j \leftarrow a'_j]}) - f(g).$$

*When the specific mutation $a_j \to a'_j$ is unambiguous or when referring generally to the effect of a mutation at locus $j$, we may use the simpler notation $s_j(g) = f(g_{[j]}) - f(g)$.*

## C.1 Ruggedness

### C.1.1 Number of Local Optima

**Definition C.8** (Local and Global Optima). *A genotype $g^\ell \in \mathcal{G}$ is a **local optimum** if its fitness is greater than or equal to that of all its neighbors: $f(g^\ell) \geq f(g')$ for all $g' \in \mathcal{N}(g^\ell)$. A **global optimum** $g^*$ is a genotype with the maximum fitness across the entire landscape: $g^* \in \arg\max_{g \in \mathcal{G}} f(g)$. We generally assume a unique global optimum for simplicity, although the concepts readily extend to scenarios with multiple global optima.*

The *number of local optima*, $n_{\text{lo}}$, serves as a primary indicator of landscape ruggedness [4]. A smooth landscape possesses only a single local optimum, which coincides with the global optimum. In contrast, highly rugged landscapes, such as Kauffman's $NK$ model [112] with $k = n-1$ interactions, can feature a large number of local optima, potentially scaling exponentially with system size (e.g., approximately $\frac{2^n}{n+1}$ local optima for certain $NK$ parameters [71]).

In the graph-based representation employed by `GraphFLA`, where directed edges implicitly encode fitness comparisons between neighboring genotypes, local optima correspond directly to *sink* nodes (i.e., nodes with no outgoing edges). These sinks can be efficiently identified using standard graph algorithms (e.g., via the `igraph` library). To facilitate comparisons between landscapes of varying sizes, the absolute number of local optima is often normalized by the total number of genotypes, $|\mathcal{G}| = \prod_{i=1}^n m_i$. This yields the dimensionless quantity *fraction of local optima*.

### C.1.2 Autocorrelation

Autocorrelation assesses the smoothness or ruggedness of a fitness landscape by measuring the correlation between the fitness values of genotypes encountered along random walks [113, 114]. Specifically, consider a random walk $g_0, g_1, \ldots, g_L$ of length $L$ through the genotype space $\mathcal{G}$, where each step $g_{t+1}$ is typically chosen uniformly at random from the neighbors $\mathcal{N}(g_t)$. The lag-$k$ autocorrelation $\rho_a(k)$ is defined as:

$$\rho_a(k) = \frac{\mathbb{E}[(f(g_t) - \langle f \rangle)(f(g_{t+k}) - \langle f \rangle)]}{\text{Var}[f(g_t)]} \tag{A1}$$

where the expectation $\mathbb{E}[\cdot]$ and variance $\text{Var}[\cdot]$ are taken over all valid time steps $t$ (and potentially multiple walks), and $\langle f \rangle$ represents the average fitness value across all considered steps $g_t$.

The primary focus is often on the lag-1 autocorrelation, $\rho_a(1)$ (simplified as $\rho_a$), which measures the fitness correlation between genotypes separated by a single mutation (Hamming distance 1).

In practice, `GraphFLA` estimates $\rho_a$ by simulating a large number ($N_{walks}$, e.g., $1,000$) of independent random walks, each of length $L$. A common choice for the walk length is $L = n$, where $n$ is the number of loci. The covariance term (numerator) and variance term (denominator) in Eq. A1 are estimated by averaging over all adjacent pairs $(g_t, g_{t+1})$ across all simulated walks. Averaging over multiple walks provides a robust estimate.

A value of $\rho_a$ close to 1 indicates a smooth landscape where fitness changes gradually between neighboring genotypes. Conversely, $\rho_a$ values close to 0 suggest a rugged landscape where the fitness of neighbors is largely uncorrelated, implying rapid and unpredictable fitness changes.

### C.1.3 Neighbor Fitness Correlation (NFC)

Another intuitive measure of landscape ruggedness assesses the relationship between a genotype's fitness and the average fitness of its mutational neighbors. This evaluates the tendency for high-fitness genotypes to be surrounded by neighbors that also have high fitness, and conversely for low-fitness genotypes.

Specifically, we compute the Pearson correlation coefficient, denoted $\rho_{f,\langle f\rangle_{\mathcal{N}}}$, between the fitness $f(g)$ of each genotype $g \in \mathcal{G}$ and the average fitness of its neighborhood $\langle f\rangle_{\mathcal{N}(g)}$:

$$\text{NFC} = \text{Cor}\left[f(g), \langle f\rangle_{\mathcal{N}(g)}\right], \tag{A2}$$

where $\langle f\rangle_{\mathcal{N}(g)}$ is the mean fitness over all single-mutant neighbors of $g$:

$$\langle f\rangle_{\mathcal{N}(g)} = \frac{1}{|\mathcal{N}(g)|} \sum_{g' \in \mathcal{N}(g)} f(g'). \tag{A3}$$

The correlation in Eq. (A2) is calculated across all genotypes $g \in \mathcal{G}$ in the landscape.

This neighbor fitness correlation provides insight into the local smoothness or ruggedness of the landscape structure:

- A value of $\text{NFC} \approx 1$ indicates a relatively smooth landscape where fitness changes tend to be gradual; high-fitness genotypes are typically surrounded by other high-fitness genotypes.

- A value of $\text{NFC} \approx 0$ suggests a rugged landscape where a genotype's fitness provides little predictive power for its neighbors' fitness, indicating abrupt changes.

- A value of $\text{NFC} \approx -1$, though less common, would imply an anticorrelated or oscillatory landscape structure where high-fitness genotypes are predominantly surrounded by low-fitness neighbors, and vice versa.

This measure focuses specifically on the fitness relationship between immediate neighbors, and complements measures like autocorrelation (C.1.2) which consider correlations along walks.

### C.1.4 Roughness-Slope Ratio ($r/s$)

The roughness-slope ratio ($r/s$) quantifies the deviation of a fitness landscape from a purely additive model, thereby measuring the relative contribution of epistasis to the landscape structure [115, 116]. It is defined as the ratio of the landscape's 'roughness' ($r$), representing the magnitude of non-additive effects (residuals from an additive fit), to its 'slope' ($s$), representing the average magnitude of additive effects. A higher $r/s$ value indicates greater ruggedness and stronger relative epistasis, while $r/s = 0$ corresponds to a perfectly additive (non-epistatic) landscape.

To compute $r/s$, an additive fitness model $f^{\text{add}}(g)$ is fitted to the observed fitness data $f(g)$ using ordinary least squares (OLS) regression. For multi-allelic landscapes, genotypes are typically represented using one-hot encoding. The additive model is generally specified as:

$$f^{\text{add}}(g) = \beta_0 + \sum_{i=1}^{n} \sum_{a \in \mathcal{A}'_i} \beta_{i,a} X_{i,a}(g), \tag{A4}$$

where $X_{i,a}(g)$ is a binary indicator variable (1 if genotype $g$ has allele $a$ at locus $i$, 0 otherwise), $\beta_0$ is the intercept, and $\beta_{i,a}$ are the fitted coefficients representing the additive effect of allele $a$ at locus $i$ relative to a reference allele. The inner sum $\sum_{a \in \mathcal{A}'_i}$ typically runs over $m_i - 1$ alleles for each locus $i$ (excluding one reference allele per locus, denoted implicitly by $\mathcal{A}'_i$) to ensure model identifiability.

The roughness $r$ is defined as the root-mean-square error (RMSE) between the true fitness values $f(g)$ and the fitness predicted by the additive model $f^{\text{add}}(g)$:

$$r = \sqrt{\frac{1}{|\mathcal{G}|}\sum_{g\in\mathcal{G}}(f(g) - f^{\text{add}}(g))^2}. \tag{A5}$$

The slope $s$ is calculated as the average absolute value of the estimated additive coefficients (excluding the intercept and reference allele coefficients, which are implicitly zero or absorbed):

$$s = \frac{1}{\sum_{k=1}^{n}(m_k - 1)}\sum_{i=1}^{n}\sum_{a\in\mathcal{A}_i'}|\beta_{i,a}|. \tag{A6}$$

Here, the denominator $\sum_{k=1}^{n}(m_k-1)$ represents the total number of independent additive coefficients fitted in the model (one for each non-reference allele across all loci), and the summation $\sum_{a\in\mathcal{A}_i'}$ covers these specific fitted coefficients for locus $i$.

The $r/s$ ratio thus provides a scale-independent measure comparing the magnitude of epistatic deviations to the average strength of individual additive allelic effects.

### C.1.5 Gamma statistic.

This measure is initially introduced in [117] for di-allelic data (i.e., $m_i = 2$, e.g., genotypes encoded with "0"s and "1"s as is common in mutation data) and is extended to multi-allelic data (i.e., $m_i \geq 3$, DNA sequences) in [106]. It is defined as the single-step correlation of fitness effects for mutations between neighboring genotypes. It quantifies how the fitness effect of a focal mutation is altered when it occurs in a different genetic background, averaged over all genotypes of the fitness landscape. Geometrically, $\gamma$ measures the correlation between "slopes" (i.e., direction and magnitude) of the same mutation put into different genetic backgrounds. Thus, if the fitness effect of a mutation is independent of its genetic background (i.e., if there is no epistasis), the correlation in slopes will be perfect ($\gamma = 1$), whereas it will be zero if the fitness slopes of each genotype are independent of the fitnesses of other genotypes. Depending on the scale $\gamma$ can either be used to quantify the strength of gene×gene interactions between specific mutations or as an overall measure for the entire landscape.

Then the matrix of epistatic effects between loci $i$ and $j$ carrying alleles $a_i, b_i \in \mathcal{A}_i$ and $A_j, B_j \in \mathcal{A}_j$ is given by

$$\gamma_{(a_i,b_i)\to(A_j,B_j)} = \text{Cor}\left[s_{(A_j,B_j)}(g), s_{(A_j,B_j)}\left(g_{[(a_i,b_i)]}\right)\right] = \frac{\sum_g s_j(g)s_j(g_{[i]})}{\sum_g(s_j(g))^2}. \tag{A7}$$

where $g := \{x \in \mathcal{G} | x_i = a_i \text{ or } x_i = b_i \text{ and } x_j = A_j \text{ or } x_j = B_j\} \subseteq \mathcal{G}$ such that the sum is only calculated over the subset of genotypes carrying one of the two focal alleles at each focal locus. Thus, $\gamma_{(a_i,b_i)\to(A_j,B_j)}$ is a quadratic matrix of dimension $\left(\sum_{i=1}^{n}\frac{|\mathcal{A}_i|(|\mathcal{A}_i|-1)}{2}\right)$.

Likewise, the epistatic effect of a mutation in locus $i$ with alleles $(a_i, b_i)$ on other loci (and pairs of alleles) can be calculated as

$$\gamma_{(a_i,b_i)\to} = \text{Cor}\left[s(g), s\left(g_{[(a_i,b_i)]}\right)\right] = \frac{\sum_{j\neq i}\sum_{a_j}\sum_g s_j(g)s_j(g_{[i]})}{\sum_{j\neq i}\sum_{a_j}\sum_g(s_j(g))^2}, \tag{A8}$$

where the summation index $\mathfrak{a}_j = \{(A_j, B_j) \mid A_j, B_j \in \mathcal{A}_j \text{ and } A_j \neq B_j\}$ is over the set of subsets of size two that can be constructed from all alleles found at locus $j$. Note that the third summation index $g$ changes depending on $\mathfrak{a}_j$.

An additional summation allows calculation of the epistatic effect of a mutation in locus $i$ carrying allele $(a_i)$ on other loci (and pairs of alleles) can be calculated as:

$$\gamma_{a_i \to} = \text{Cor}[s(g), s(g_{[a_i]})] = \frac{\sum_{j \neq i} \sum_{\mathfrak{f}_i} \sum_{\mathfrak{a}_j} \sum_g s_j(g) s_j(g_{[i]})}{\sum_{j \neq i} \sum_{\mathfrak{f}_i} \sum_{\mathfrak{a}_j} \sum_g (s_j(g))^2}, \tag{A9}$$

where $\mathfrak{f}_i = \{(a_i, b_i) \mid b_i \in \mathcal{A}_i \text{ and } a_i \neq b_i\}$ such that the sum is only calculated over the elements of the set of subsets of size two that can be constructed from all alleles found at locus $i$ that contain allele $a_i$.

Then, summing over $l_i = \{(a_i, b_i) \mid a_i, b_i \in \mathcal{A}_i \text{ and } a_i \neq b_i\}$, i.e., the elements of the set of subsets of size two that can be constructed from all alleles found at locus $i$, gives the epistatic effect of a mutation in locus $i$:

$$\gamma_{i \to} = \text{Cor}[s(g), s(g_{[i]})] = \frac{\sum_{j \neq i} \sum_{l_i} \sum_{\mathfrak{a}_j} \sum_g s_j(g) s_j(g_{[i]})}{\sum_{j \neq i} \sum_{l_i} \sum_{\mathfrak{a}_j} \sum_g (s_j(g))^2}. \tag{A10}$$

Similarly, the epistatic effect of other mutations (again considering pairs of alleles first) on locus $j$ with alleles $(a_i, b_i)$ can be calculated as

$$\gamma_{\to (A_j, B_j)} = \text{Cor}\left[s_{(A_j, B_j)}(g), s_{(A_j, B_j)}(g_1)\right] = \frac{\sum_{i \neq j} \sum_{\mathfrak{a}_i} \sum_g s_j(g) s_j(g_{[i]})}{\sum_{i \neq j} \sum_{\mathfrak{a}_i} \sum_g (s_j(g))^2}, \tag{A11}$$

the epistatic effect of other mutations on locus $j$ carrying allele $A_j$ is given by

$$\gamma_{\to A_j} = \text{Cor}\left[s_{(A_j)}(g), s_{(A_j)}(g_1)\right] = \frac{\sum_{i \neq j} \sum_{\mathfrak{f}_j} \sum_{\mathfrak{a}_i} \sum_g s_j(g) s_j(g_{[i]})}{\sum_{i \neq j} \sum_{\mathfrak{f}_j} \sum_{\mathfrak{a}_i} \sum_g (s_j(g))^2}, \tag{A12}$$

and the epistatic effect of other mutations on locus $j$ becomes

$$\gamma_{\to j} = \text{Cor}[s_j(g), s_j(g_1)] = \frac{\sum_{i \neq j} \sum_{l_j} \sum_{\mathfrak{a}_i} \sum_g s_j(g) s_j(g_{[i]})}{\sum_{i \neq j} \sum_{l_j} \sum_{\mathfrak{a}_i} \sum_g (s_j(g))^2}, \tag{A13}$$

Finally, $\gamma_d$, that is the decay of correlation of fitness effects with Hamming distance $d$ (i.e., the cumulative epistatic effect of $d$ mutations averaged over the entire fitness landscape) is calculated as

$$\gamma_d = \text{Cor}[s(g), s_j(g_d)] = \frac{\sum_g \sum_{g_d} \sum_{j \neq i_1, i_2, \ldots, i_d} \sum_{\mathcal{A}_j \setminus \{A_j\}} s_j(g) s_j(g_{[i_1 i_2 \ldots i_d]})}{\sum_g \sum_{\mathcal{A}_j \setminus \{A_j\}} (s_j(g))^2}, \tag{A14}$$

where the last summation is over all different alleles present at locus $j$ except the one carried by genotype $g$ at locus $j$. We provide implementation of $\gamma_1$ in `GraphFLA` since increasing $d$ would significantly increase the amount of calculations required (e.g., $\gamma_d$ would require $\binom{n}{d}$ times more calculations compared to $\gamma_1$). In doing this, we utilize vector operation from `pandas` and `numpy` whenever possible to avoid nested loops as would appear with a brute-force implementation of (A14).

## C.2 Navigability

Navigability refers to the ease with which an evolving population can traverse the fitness landscape, typically towards genotypes of higher fitness, under the influence of mutation and natural selection. A highly navigable landscape allows populations to readily find high-fitness peaks, potentially the global optimum, through series of fitness-increasing mutations. Conversely, low navigability implies that evolutionary trajectories might be hindered, often getting trapped on suboptimal peaks due to fitness valleys or complex landscape structures [18, 4, 34].

### C.2.1 Global Optima Accessibility

**Definition C.9** (Adaptive Walk). *An adaptive walk is a sequence of genotypes $g_0, g_1, \ldots, g_k$ such that each genotype $g_{t+1}$ is a neighbor of $g_t$ ($g_{t+1} \in \mathcal{N}(g_t)$) and has strictly higher fitness ($f(g_{t+1}) > f(g_t)$) for all steps $t \in \{0, \ldots, k-1\}$. The walk terminates at step $k$ when $g_k$ is a local optimum (Definition C.8). Under the strong selection weak mutation (SSWM) regime [118], evolution often proceeds along such paths, as natural selection favors higher-fitness genotypes and prevents populations from crossing fitness valleys.*

**Definition C.10** (Evolutionary Accessibility). *A target genotype $g_T \in \mathcal{G}$ is considered **accessible** from a starting genotype $g_S \in \mathcal{G}$ if there exists at least one adaptive walk (Definition C.9) connecting $g_S$ to $g_T$. Specifically, a sequence of single mutations $g_S = g_0, g_1, \ldots, g_k = g_T$ exists such that $g_{t+1} \in \mathcal{N}(g_t)$ and $f(g_{t+1}) > f(g_t)$ for all $t \in \{0, \ldots, k-1\}$.*

Global optimum accessibility quantifies the extent to which the global optimum ($g^*$) is accessible (Definition C.10) from other genotypes in the landscape via adaptive walks (Definition C.9). Fitness landscape theory posits that rugged landscapes can hinder adaptation by trapping evolving populations on suboptimal local optima, thereby limiting access to the global optimum [114, 56].

Following [4], we measure global optimum accessibility as the fraction of genotypes residing within the basin of attraction of $g^*$, denoted $\mathcal{B}(g^*)$ (Definition C.11). This fraction represents the probability that an adaptive walk initiated from a random genotype will eventually reach $g^*$:

$$\alpha_{\mathrm{go}} = \frac{|\mathcal{B}(g^*)|}{|\mathcal{G}|} = \frac{|\mathcal{B}(g^*)|}{\prod_{i=1}^{n} m_i}. \tag{A15}$$

An accessibility value $\alpha_{\mathrm{go}} \approx 1$ suggests a highly navigable landscape where the global optimum is readily reachable via simple hill-climbing dynamics from most starting points. Conversely, a low $\alpha_{\mathrm{go}}$ indicates that reaching the global optimum through adaptive walks alone is improbable for populations starting from random initial genotypes, suggesting they are likely to become trapped on local optima.

It is noteworthy that recent findings suggest some rugged landscapes can still be highly navigable, with the global optimum accessible from a large fraction of genotypes [4]. Furthermore, even if $g^*$ is inaccessible via adaptive walks from certain regions, this does not necessarily prevent its discovery through methods used in directed evolution. Many computational optimizers (e.g., simulated annealing [119], machine learning-guided directed evolution) employ mechanisms to traverse fitness valleys and escape local optima. Nevertheless, highly rugged landscape topographies generally pose greater challenges for locating global fitness peaks [18, 34].

### C.2.2 Basin of Attraction and BFC

**Definition C.11** (Basin of Attraction). *The basin of attraction $\mathcal{B}(g^\ell)$ of a local optimum $g^\ell$ is the set of all genotypes $g \in \mathcal{G}$ from which at least one adaptive walk starting at $g$ can reach $g^\ell$:*

$$\mathcal{B}(g^\ell) = \{g \in \mathcal{G} \mid \exists \text{ an adaptive walk } g_0 = g, \ldots, g_k = g^\ell\}.$$

*The size of the basin, $|\mathcal{B}(g^\ell)|$, reflects the accessibility of the local optimum $g^\ell$ within the genotype space. Larger basins are often associated with higher-fitness local optima [4]. We use the basin size-fitness correlation to quantify this relationship.*

`GraphFLA` estimates basin sizes by analyzing adaptive walks. It supports two distinct methods, yielding different definitions and computational properties for basin size:

- **Stochastic adaptive walks:** Also known as *first-improvement hill climbing* [120]. At each step, one neighbor $g' \in \mathcal{N}(g)$ with $f(g') > f(g)$ is chosen uniformly at random from all such improving neighbors. Because of this stochasticity, walks from the same starting genotype can terminate at different local optima, leading to overlapping basins [4]. `GraphFLA` calculates the basin size deterministically by identifying the set of *all* genotypes from which *at least one* such adaptive walk can reach the local optimum $g^\ell$. In the graph representation, this corresponds to finding all ancestors of the node $g^\ell$ reachable via directed paths representing fitness increases, typically using functions like those available in `igraph`.

This computation can be expensive for large landscapes (e.g., $> 300,000$ genotypes) with numerous local optima.

- **Greedy adaptive walks:** Also known as *best-improvement hill climbing*. At each step, this walk deterministically selects a neighbor $g' \in \mathcal{N}(g)$ that maximizes the fitness increase, $\Delta f = f(g') - f(g)$. If multiple neighbors offer the same maximal increase, a consistent tie-breaking rule (e.g., random choice implemented once or lexicographical order) ensures determinism. The path from any starting genotype $g$ is unique, meaning each genotype belongs to the basin of exactly one local optimum. Consequently, these basins partition the genotype space $\mathcal{G}$, and their sizes sum to $|\mathcal{G}| = \prod_{i=1}^{n} m_i$. `GraphFLA` calculates these basin sizes by simulating a greedy walk starting from every genotype $g \in \mathcal{G}$ and recording the local optimum reached.

Using these two measures of basin size, `GraphFLA` provides two features, $\text{BFC}_{\text{acc}}$ and $\text{BFC}_{\text{greedy}}$ that assess whether local optima with higher fitness tend to have larger basin of attraction:

$$\text{BFC}_{\text{acc}} = \text{Cor}\left[ f(g^{\ell}), |\mathcal{B}_{\text{acc}}(g^{\ell})| \right] \tag{A16}$$

$$\text{BFC}_{\text{greedy}} = \text{Cor}\left[ f(g^{\ell}), |\mathcal{B}_{\text{greedy}}(g^{\ell})| \right] \tag{A17}$$

A higher value of BFC indicates that fitter local optima would have larger basin of attraction compared to those have lower fitness, which results in their higher evolutionary accessibility and increased navigability of the whole landscape [4].

### C.2.3 Fitness Distance Correlation (FDC)

Fitness Distance Correlation (FDC) is a measure used to assess the global structure of a fitness landscape and its potential navigability by an evolutionary process [121]. Specifically, it quantifies the relationship between the fitness of genotypes and their distance to a known global optimum $g^*$. In biological fitness landscapes, this distance is typically the Hamming distance $d_H(g, g^*)$ (Definition C.3) between a genotype $g \in \mathcal{G}$ and the global optimum $g^* \in \mathcal{G}$ (Definition C.8).

The FDC is calculated as the Pearson correlation coefficient between the fitness values $f(g)$ of all genotypes in the landscape (or a representative sample) and their respective Hamming distances to the global optimum $g^*$:

$$\text{FDC} = \text{Cor}\left[ f(g), d_H(g, g^*) \right], \tag{A18}$$

The interpretation of the FDC value provides insights into the navigability of the fitness landscape:

- **FDC $\approx -1$**: A strong negative correlation indicates that genotypes with higher fitness tend to be closer (i.e., have a smaller Hamming distance) to the global optimum $g^*$. This signifies a relatively smooth, "funnel-like" landscape structure where fitness gradients consistently guide an evolutionary search towards $g^*$. Such landscapes are considered highly navigable by processes like adaptive walks (Definition C.9), as selection for increased fitness generally directs the population towards the global peak.

- **FDC $\approx 0$**: A correlation close to zero suggests that there is no clear relationship between a genotype's fitness and its distance to the global optimum. This is characteristic of rugged or random landscapes, where fitness values can change erratically and provide little information about the direction towards $g^*$. In such landscapes, navigability is low, and evolutionary processes are more likely to become trapped on local optima (Definition C.8) far from $g^*$.

- **FDC $\approx +1$**: A strong positive correlation implies that genotypes with higher fitness tend to be further away from the global optimum $g^*$. This indicates a "deceptive" landscape, where selection for immediate fitness gains would systematically lead an evolving population away from the global optimum. Such landscapes are exceptionally difficult to navigate towards $g^*$ using simple hill-climbing strategies.

### C.2.4 Evolvability-Enhancing Mutations

*Evolvability* [122] refers to the capacity of a biological system to generate adaptive heritable variation. Unlike measures focusing solely on the direct fitness impact of a mutation (first-order selection), evolvability emphasizes the potential for future adaptive change (second-order selection) [122]. On top of this, Wagner introduced the concept of an evolvability-enhancing (EE) mutation, defined as one that modifies the genetic background such that subsequent mutations at other loci tend to be, on average, more beneficial or less deleterious, irrespective of the initial mutation's own fitness effect.

**Definition C.12** (Evolvability-Enhancing (EE) Mutation). *Consider a mutation at locus $j$ that converts genotype $g$ to $g_{[j]}$. Let $\mathcal{N}_{-j}(g)$ denote the set of single-mutant neighbors of $g$ resulting from mutations at any locus $k \neq j$. The size of this set is $|\mathcal{N}_{-j}(g)| = \sum_{k \neq j, k=1}^{n}(m_k - 1)$. Let $\langle f \rangle_{\mathcal{N}_{-j}(g)}$ be the average fitness over the genotypes in $\mathcal{N}_{-j}(g)$. Similarly, let $\mathcal{N}_{-j}(g_{[j]})$ be the set of single-mutant neighbors of $g_{[j]}$ resulting from mutations at loci $k \neq j$, and let $\langle f \rangle_{\mathcal{N}_{-j}(g_{[j]})}$ be their average fitness.*

*The mutation $g \to g_{[j]}$ is defined as evolvability-enhancing (EE) if:*

- *It is $\sigma$-neutral ($|s_j(g)| < \sigma$, see Def. C.14) and increases the average fitness of subsequent mutants:*

$$\langle f \rangle_{\mathcal{N}_{-j}(g_{[j]})} - \langle f \rangle_{\mathcal{N}_{-j}(g)} > 0. \tag{A19}$$

- *It is beneficial ($s_j(g) > 0$) and enhances the fitness prospects of subsequent mutations beyond its own additive contribution:*

$$\langle f \rangle_{\mathcal{N}_{-j}(g_{[j]})} - \langle f \rangle_{\mathcal{N}_{-j}(g)} > s_j(g). \tag{A20}$$

*This condition is equivalent to requiring that the average fitness effect of subsequent mutations (relative to the background they arise in) is greater in the $g_{[j]}$ background than in the $g$ background:*

$$\left( \langle f \rangle_{\mathcal{N}_{-j}(g_{[j]})} - f(g_{[j]}) \right) > \left( \langle f \rangle_{\mathcal{N}_{-j}(g)} - f(g) \right). \tag{A21}$$

This definition implies that an EE mutation at locus $j$ exhibits, on average, positive epistasis with mutations occurring at other loci $k \neq j$ [107]. Following [107], we primarily consider beneficial EE mutations. Such mutations can spread through populations via direct (first-order) selection due to their immediate fitness advantage, potentially increasing future evolvability as a byproduct without needing selection to act directly on evolvability itself (second-order selection). Beneficial EE mutations are expected to shift the distribution of fitness effects (DFE) of subsequent mutations favourably, for instance, by reducing the impact of deleterious mutations or increasing the frequency and/or magnitude of beneficial ones.

### C.2.5 Mean Accessible Path Length

The minimum number of single mutations required to transition between a genotype $g$ and the global optimum $g^*$ is their Hamming distance, $d_H(g, g^*)$ (Definition C.3). However, not all paths realizing this minimum distance are necessarily evolutionarily accessible; that is, they may not consist solely of fitness-increasing steps (Section C.2.1). Consequently, the shortest accessible path, composed entirely of fitness-increasing mutations (an adaptive walk, Definition C.9), can be longer than the Hamming distance, potentially requiring detours to navigate around fitness valleys [59, 123].

We measure the typical length of such paths using the mean shortest accessible path length to the global optimum, averaged over all genotypes from which $g^*$ is reachable.

**Definition C.13** (Shortest Accessible Path Length). *For a genotype $g$ within the basin of attraction of the global optimum $g^*$ (i.e., $g \in \mathcal{B}(g^*)$), the shortest accessible path length, $d_{acc}(g, g^*)$, is the minimum number of steps $k$ in an adaptive walk $g_0 = g, \ldots, g_k = g^*$ terminating at $g^*$. If $g$ is not in the basin of $g^*$ ($g \notin \mathcal{B}(g^*)$), then by definition $d_{acc}(g, g^*) = \infty$.*

The mean accessible path length to the global optimum, $\langle d_{\mathrm{acc}} \rangle_{g^*}$, is calculated as:

$$\langle d_{\mathrm{acc}} \rangle_{g^*} = \frac{1}{|\mathcal{B}(g^*)|} \sum_{g \in \mathcal{B}(g^*)} d_{\mathrm{acc}}(g, g^*).$$

The average is taken over all genotypes $g$ belonging to the basin of attraction $\mathcal{B}(g^*)$ of the global optimum (Definition C.11).

In `GraphFLA`, $d_{\text{acc}}(g, g^*)$ is computed for all $g \in \mathcal{B}(g^*)$ using shortest path algorithms on the subgraph containing only fitness-increasing transitions, implemented via `igraph`'s `distances` targeting $g^*$. While the absolute value of $\langle d_{\text{acc}} \rangle_{g^*}$ is context-dependent (influenced by landscape size and dimensionality), comparing it to the mean Hamming distance between genotypes in the basin and the optimum, $\langle d_H(g, g^*) \rangle_{g \in \mathcal{B}(g^*)}$, can provide valuable insights into landscape navigability. In a perfectly smooth landscape with a single peak, $d_{\text{acc}}(g, g^*) = d_H(g, g^*)$ for all $g$. In rugged landscapes, accessible paths often meander, leading to $\langle d_{\text{acc}} \rangle_{g^*} > \langle d_H(g, g^*) \rangle_{g \in \mathcal{B}(g^*)}$ [4, 36]. A larger difference signifies greater path indirectness imposed by the landscape's rugged structure.

## C.3 Epistasis

### C.3.1 Classification of Epistasis

This section defines different types of pairwise epistatic interactions between mutations at two distinct loci, say $i$ and $j$. Epistasis occurs when the fitness effect of a mutation at one locus depends on the allele present at the other locus. We classify epistasis based on how the fitness effects change across genetic backgrounds. Consider a reference genotype $g$, the single mutants $g_{[i]}$ and $g_{[j]}$, and the double mutant $g_{[ij]}$ (assuming specific mutations $a_i \to a_i'$ at locus $i$ and $a_j \to a_j'$ at locus $j$ are implied or defined). The interaction epistasis term $\epsilon_{ij}$ measures the deviation from additivity:

$$\epsilon_{ij} = f(g_{[ij]}) - f(g) - [f(g_{[i]}) - f(g)] - [f(g_{[j]}) - f(g)] \tag{A22}$$
$$= f(g_{[ij]}) - f(g_{[i]}) - f(g_{[j]}) + f(g) \tag{A23}$$

Using the selection coefficient notation from Definition C.7, where $s_i(g) = f(g_{[i]}) - f(g)$ is the effect of the mutation at locus $i$ in the background $g$, and $s_i(g_{[j]}) = f(g_{[ij]}) - f(g_{[j]})$ is the effect of the same mutation at locus $i$ but in the background $g_{[j]}$, the epistasis term can be equivalently written as:

$$\epsilon_{ij} = s_i(g_{[j]}) - s_i(g) = s_j(g_{[i]}) - s_j(g)$$

Based on the sign and magnitude of the selection coefficients involved, we can classify the interaction:

- **No epistasis ($\epsilon_{ij} = 0$):** The effects of the mutations are additive. The effect of mutation $i$ is the same regardless of the allele at locus $j$, i.e., $s_i(g_{[j]}) = s_i(g)$.
- **Magnitude epistasis ($\epsilon_{ij} \neq 0$, no sign changes):** The fitness effects are non-additive ($\epsilon_{ij} \neq 0$), but the sign of each mutation's effect remains consistent across the backgrounds considered. That is, $s_i(g)$ and $s_i(g_{[j]})$ have the same sign (or zero), and $s_j(g)$ and $s_j(g_{[i]})$ also have the same sign (or zero). This occurs when the combined effect deviates from the sum of individual effects.
    - **Positive epistasis ($\epsilon_{ij} > 0$):** The combined effect is greater than expected from additivity ($f(g_{[ij]}) - f(g) > s_i(g) + s_j(g)$). This includes synergistic interactions where, for example, two beneficial mutations together yield a larger benefit than their sum, or two deleterious mutations are less harmful together than expected (antagonistic interaction between deleterious mutations).
    - **Negative epistasis ($\epsilon_{ij} < 0$):** The combined effect is less than expected from additivity ($f(g_{[ij]}) - f(g) < s_i(g) + s_j(g)$). This includes antagonistic interactions like diminishing returns, where two beneficial mutations yield a smaller benefit together than their sum [38] (see Section C.3.2), or synergistic interactions where two deleterious mutations are more harmful together than expected. Negative epistasis can decelerate adaptation [41, 38, 124], create concave fitness peaks [17], and increase mutational robustness near peaks [17, 11].
- **Sign epistasis ($\epsilon_{ij} \neq 0$, one sign change):** The sign of the fitness effect of one mutation (e.g., beneficial vs. deleterious) flips depending on the background provided by the other mutation, while the second mutation's sign remains consistent. For instance, mutation $i$ might be

beneficial in background $g$ ($s_i(g) > 0$) but deleterious in background $g_{[j]}$ ($s_i(g_{[j]}) < 0$), while mutation $j$'s sign remains the same ($s_j(g)$ and $s_j(g_{[i]})$ have the same sign). Sign epistasis restricts accessible mutational trajectories [5, 36, 18] and contributes to landscape ruggedness [37, 125].

- **Reciprocal sign epistasis** ($\epsilon_{ij} \neq 0$, **two sign changes**): A symmetric form where the sign of the effect of both mutations changes depending on the background provided by the other. For example, both single mutations might be deleterious ($s_i(g) < 0, s_j(g) < 0$), but each becomes beneficial in the background containing the other mutation ($s_i(g_{[j]}) > 0, s_j(g_{[i]}) > 0$), often leading to a beneficial double mutant ($f(g_{[ij]}) > f(g)$).

Formally, the type of epistatic interaction $\mathfrak{e}(g, i, j)$ between specific mutations at loci $i$ and $j$ relative to a reference genotype $g$ can be classified based on the selection coefficients:

$$\mathfrak{e}(g,i,j) = \begin{cases} \text{None} & \text{if } \epsilon_{ij} = 0 \\ \text{Magnitude} & \text{if } \epsilon_{ij} \neq 0 \text{ and } [s_i(g) \cdot s_i(g_{[j]}) \geq 0 \text{ and } s_j(g) \cdot s_j(g_{[i]}) \geq 0] \\ \text{Reciprocal Sign} & \text{if } s_i(g) \cdot s_i(g_{[j]}) < 0 \text{ and } s_j(g) \cdot s_j(g_{[i]}) < 0 \\ \text{Sign} & \text{otherwise (i.e., if } \epsilon_{ij} \neq 0 \text{ and exactly one sign product is negative)} \end{cases}$$
$$\text{(A24)}$$

where the condition $\epsilon_{ij} = s_i(g_{[j]}) - s_i(g) = 0$ defines the non-epistatic case. The product conditions check for sign changes (a negative product indicates a sign change, assuming neither term is zero).

The prevalence of each epistasis type across the entire landscape can be estimated by enumerating all pairs of single mutations originating from all possible reference genotypes $g$. Computationally, this involves analyzing local structures corresponding to double mutants relative to a reference genotype. For graph-based representations, these structures correspond to specific 4-node motifs. As noted by [4], specific non-isomorphic directed motifs identifiable using graph libraries like `igraph` correspond to these epistasis types (e.g., motifs 66, 52, and 19 were identified as potentially representing magnitude, sign, and reciprocal sign epistasis, respectively).

### C.3.2 Global epistasis.

Global epistasis refers to systematic trends where the fitness effect of a mutation exhibits a predictable relationship with the overall fitness of the genetic background it occurs in [39, 124, 41]. Global epistasis often manifests in two particular forms:

- **Diminishing returns epistasis:** This pattern describes scenarios where the fitness benefit ($f(g') - f(g) > 0$) conferred by a beneficial mutation decreases as the fitness of the genetic background, $f(g)$, increases. In other words, the positive impact of a beneficial mutation diminishes in already fit genotypes [38]. To quantify this, we identify all beneficial single-step mutations (where the selection coefficient $f(g') - f(g)$ is positive) across the landscape. Diminishing returns is then measured by calculating the Pearson correlation coefficient between the fitness of the background genotype, $f(g)$, and the corresponding positive selection coefficient, $s(g \to g')$, across all such beneficial mutations. A negative correlation value indicates the presence of diminishing returns epistasis.

- **Increasing costs epistasis:** This describes a pattern where the fitness cost (negative effect) of a deleterious mutation ($f(g') - f(g) < 0$) becomes larger (more negative) as the fitness of the genetic background, $f(g)$, increases. This implies that fitter genotypes are less tolerant to deleterious mutations [126]. To quantify this, we identify all deleterious single-step mutations (where the selection coefficient $s(g \to g')$ is negative). Increasing costs epistasis is measured by calculating the Pearson correlation coefficient between the fitness of the background genotype, $f(g)$, and the magnitude (absolute value) of the corresponding negative selection coefficient, $|s(g \to g')|$, across all such deleterious mutations. A positive correlation value indicates the presence of increasing costs epistasis, meaning the fitness cost tends to be larger in higher-fitness backgrounds.

These global epistatic trends suggest a general "coupling" of mutations through overall fitness, potentially leading to predictable macro-evolutionary dynamics like decelerating rates of adapta-

tion [41, 124]. This contrasts with idiosyncratic epistasis (see the next section), where interactions depend more specifically on the identities and combination of the mutations involved [39].

### C.3.3 Idiosyncratic Epistasis

Idiosyncratic epistasis describes genetic interactions where the fitness effect of a mutation depends strongly and often unpredictably on the specific genetic background [127, 39, 128]. The term "idiosyncratic" highlights that these interactions are specific to the identities of the involved loci and alleles, resulting in context-dependent effects that can vary significantly even across similar genetic backgrounds. This contrasts with global epistasis models in the previous section (e.g., diminishing returns, increasing costs), where a mutation's effect is predicted to vary systematically based mainly on the background genotype's fitness.

To quantify the extent of idiosyncratic epistasis in a landscape, we measure the variability of individual mutation effects across different genetic backgrounds. Following Lyons et al., an idiosyncrasy index can be calculated for each specific mutational transition.

Consider a specific mutation at locus $i$ changing allele $a_i \in \mathcal{A}_i$ to allele $b_i \in \mathcal{A}_i$ ($a_i \neq b_i$). Let $\mathcal{G}_{i,a_i} = \{g \in \mathcal{G} \mid \text{allele at locus } i \text{ in } g \text{ is } a_i\}$ denote the set of all genotypes (backgrounds) carrying allele $a_i$ at locus $i$. The selection coefficient for this specific mutation $a_i \rightarrow b_i$ occurring in a background $g \in \mathcal{G}_{i,a_i}$ is defined according to Definition C.7:

$$s_{i,a_i \rightarrow b_i}(g) = f(g_{[i \leftarrow b_i]}) - f(g), \tag{A25}$$

where $g_{[i \leftarrow b_i]}$ is the genotype identical to $g$ but with allele $b_i$ at locus $i$.

The variability of this specific mutation's effect across all possible backgrounds is captured by its variance:

$$V_{i,a_i \rightarrow b_i} = \text{Var}_{g \in \mathcal{G}_{i,a_i}} \left[ s_{i,a_i \rightarrow b_i}(g) \right] = \frac{1}{|\mathcal{G}_{i,a_i}|} \sum_{g \in \mathcal{G}_{i,a_i}} \left( s_{i,a_i \rightarrow b_i}(g) - \bar{s}_{i,a_i \rightarrow b_i} \right)^2, \tag{A26}$$

where $\bar{s}_{i,a_i \rightarrow b_i}$ is the mean selection coefficient of the mutation $a_i \rightarrow b_i$ averaged over all backgrounds $g \in \mathcal{G}_{i,a_i}$.

To normalize this measure, the variability is compared to the overall variability of selection coefficients across the entire landscape. Let $\mathcal{S}$ represent the set of all possible single-mutation selection coefficients:

$$\mathcal{S} = \{s_{k,C_k \rightarrow D_k}(h) \mid k \in \{1, \ldots, n\}, C_k, D_k \in \mathcal{A}_k, C_k \neq D_k, h \in \mathcal{G}_{k,C_k}\}. \tag{A27}$$

The total variance of selection coefficients across the landscape is:

$$V_s = \text{Var}[s \in \mathcal{S}] = \frac{1}{|\mathcal{S}|} \sum_{s \in \mathcal{S}} (s - \bar{s})^2, \tag{A28}$$

where $\bar{s}$ is the global mean selection coefficient averaged over all single mutations in all backgrounds.

The idiosyncrasy index for the specific mutation $a_i \rightarrow b_i$ is the ratio of the standard deviation of its effect across backgrounds to the standard deviation of all selection coefficients in the landscape:

$$I_{\text{id}}(i, a_i \rightarrow b_i) = \frac{\sqrt{V_{i,a_i \rightarrow b_i}}}{\sqrt{V_s}}. \tag{A29}$$

An index value $I_{\text{id}}(i, a_i \rightarrow b_i) \approx 1$ signifies high idiosyncrasy, implying that the effect of this mutation is highly context-dependent, varying across backgrounds almost as much as selection coefficients vary globally. Conversely, $I_{\text{id}}(i, a_i \rightarrow b_i) \approx 0$ indicates low idiosyncrasy, suggesting the mutation has a relatively consistent effect regardless of the genetic background.

To derive a single measure representing the overall level of idiosyncrasy for the entire landscape, we average the index across all possible single mutations. Let $\mathcal{M}$ be the set of all possible directed single mutations, $\mathcal{M} = \{(i, a_i \to b_i) \mid i \in \{1, \ldots, n\}, a_i, b_i \in \mathcal{A}_i, a_i \neq b_i\}$. The total number of such mutations is $|\mathcal{M}| = \sum_{k=1}^{n} m_k(m_k - 1)$. The average idiosyncrasy index for the landscape is:

$$I_{\text{id}} = \frac{1}{|\mathcal{M}|} \sum_{(i, a_i \to b_i) \in \mathcal{M}} I_{\text{id}}(i, a_i \to b_i). \tag{A30}$$

A high average idiosyncrasy $I_{\text{id}}$ suggests that, overall, predicting a mutation's fitness effect requires detailed knowledge of the specific genetic background, and highlights the prevalence of complex, context-specific interactions within the landscape.

### C.3.4 Pairwise and Higher-order Epistasis

Epistatic interactions can occur between pairs of mutations (pairwise epistasis) or among multiple ones (high-order epistasis). Ideally, to exactly determine the fraction of each pair of epistasis would require decomposing the landscape into products of the single-locus variables according to the following expansion [129, 130]:

$$f(g) = a^{(0)} + \sum_i a_i^{(1)} a_i + \sum_{ij} a_{ij}^{(2)} a_i a_j + \sum_{ijk} a_{ijk}^{(3)} a_i a_j a_k + \ldots + a_{12\ldots n}^{(n)} a_1 a_2 \cdots a_n \tag{A31}$$

There are $\binom{n}{k}$ coefficients of type $a^{(k)}$ in this expansion, one for each subset of $k$ of $n$ loci. According to the binomial theorem, the total number of coefficients equals $2^n$, which makes it evident that the mapping between fitness values and expansion coefficients is one-to-one. The first-order coefficient $a^{(1)}$ describes the linear, non-epistatic effects, the second-order coefficient $a^{(2)}$ denotes pairwise epistatic interactions and so on.

However, such expansion is often computationally prohibitive because of the exponential growth in epistasis terms. As an alternative, researchers often specifically distinguish between pairwise and higher-order epistasis because the latter describes complex dependencies that are not reducible to combinations of pairwise interactions and can have profound consequences for the fitness landscape [130]. For example, higher-order epistasis can cause the effects of mutations, and even the nature of pairwise interactions, to change depending on the broader genetic background, sometimes leading to mutations switching between being beneficial and detrimental—outcomes that cannot be predicted by considering only pairwise effects [42].

To measure the prevalence of higher-order epistasis in the landscape, `GraphFLA` provides a measure, $\epsilon_{(2)}$, which assesses how much variance in fitness distributions can be explained by pairwise epistasis alone. This is performed by fitting a polynomial linear regression model with interaction terms up to the second order. $\epsilon_{(2)}$ is then derived as the $R^2$ score of this model in fitting the data.

### C.4 Neutrality

### C.4.1 Neutrality

*Mutational robustness* [43, 44, 131] measures the extent to which a genotype's fitness remains unchanged by mutations, reflecting its ability to buffer genetic perturbations. Related is the concept of *neutral mutations*, which cause little to no change in fitness. Neutral mutations allow populations to explore the genotype space without incurring significant fitness costs. Sets of genotypes connected by neutral mutations and sharing approximately the same fitness level form *neutral networks* [17, 16, 43, 13].

**Definition C.14** (Neutral Mutation and Neighbors). *A mutation converting genotype g into a neighbor $g' \in \mathcal{N}(g)$ is considered $\sigma$-neutral if the absolute fitness change is below a predefined tolerance $\sigma \geq 0$: $|f(g') - f(g)| < \sigma$. The set of $\sigma$-neutral neighbors of g is:*

$$\mathcal{N}_\sigma(g) = \{g' \in \mathcal{N}(g) \mid |f(g') - f(g)| < \sigma\}.$$

Empirically, $\sigma$ is often set based on the variance in the measured fitness across replications and act as a noise threshold. With the above definition, we can now define the mutational robustness for a specific genotype:

**Definition C.15** (Mutational Robustness). *The mutational robustness $R(g)$ of a genotype $g$ is the fraction of its neighbors that are $\sigma$-neutral [131, 122]:*

$$R(g) = \frac{|\mathcal{N}_\sigma(g)|}{|\mathcal{N}(g)|}.$$

To characterize the neutrality of the entire fitness landscape, we average the mutational robustness across all genotypes:

$$\eta = \frac{1}{|\mathcal{G}|} \sum_{g \in \mathcal{G}} R(g). \tag{A32}$$

This quantity, referred to as *landscape neutrality*, measures the overall prevalence of neutrality. A high landscape neutrality ($\langle R \rangle_\mathcal{G}$) indicates the presence of extensive neutral networks [17, 16, 43, 13]. These networks can facilitate exploration of the genotype space via neutral drift without much fitness costs, which could potentially enhance *evolvability* [122], as discussed in the following section (C.2.4).

# D  Landscape Models

In the following, we briefly introduce the various fitness landscape models implemented in `GraphFLA`. For an excellent general overview, we refer the reader to a review by Szendro et al..

**The additive model.** In the additive model, the fitness of each genotype is given by the sum/product over the individual per locus fitness effects. Thus, the fitness effect of a specific allele, drawn from a Normal distribution with mean $\mu_a$ and variance $\sigma_a^2$ [117] (see also [132]), is independent of its genetic background (i.e., it is constant across all genetic backgrounds), such that all mutations are non-interacting (i.e., there is no epistasis) and the resulting unimodal fitness landscape is (maximally) smooth. In particular, the roughness-to-slope ratio is 0 and $\mathbb{E}[\gamma_d] = 1$ for the entire range of mutational distances $d$. Note that when fitnesses are given by the product over the individual fitness effects, this model is also referred to as the multiplicative model.

**The House-of-cards model.** On the other extreme, in the House-of-cards (HoC) model [133] the fitness of each genotype is an i.i.d. normally distributed random variable with zero mean and variance $\sigma_{\text{HoC}}^2$ resulting in an uncorrelated, maximally rugged fitness landscape that is characterized by multiple local optima [56, 70]. In particular, the fitness effect of an allele entirely depends on its genetic background such that there is complete interaction between all loci (i.e., full epistasis) which is also reflected in the roughness-to-slope ratio and $\mathbb{E}[\gamma_d]$ that become infinity and zero (for $d > 0$), respectively.

**The Rough Mount Fuji model.** Introduced by [134], the Rough-Mount-Fuji (RMF) model, named after the eponymous mountain in Japan, which was initially formulated in the context of protein evolution [see also [132], for a simplified version], interpolates between the former two extremes. The fitness of a genotype is composed by an additive component (parametrized by $\mu_a$ and $\sigma_a^2$; see above) and a HoC component (parametrized by $\sigma_{\text{HoC}}^2$) such that the extent of epistatic interactions ranges between none (additive model) to complete (HoC model) depending on the relative sizes of these three parameters. In particular, when $\sigma_{\text{HoC}}^2 \ll \mu_a^2 + \sigma_a^2$ the RMF model becomes an additive model whereas for $\sigma_{\text{HoC}}^2 \gg \mu_a^2, \sigma_a^2$ it essentially behaves like a HoC model [117]. Accordingly, $0 \leq \mathbb{E}[\gamma_d] = \text{const} \leq 1$ for $d > 0$ and the roughness-to-slope ratio ranges from zero to infinity.

**The Kauffman NK model.** Another frequently used fitness landscape model that, as the RMF model, also interpolates between the additive and the HoC model [56, 70] is the Kauffman NK model, where $N$ di-allelic loci interact with $K \in \{0, 1, \ldots, L-1\}$ randomly assigned other loci. In particular, for $K = 0$ the NK model collapses to an additive model whereas for $K = L - 1$ it approaches the HoC model. Although there are different ways how groups of interacting loci can be chosen [135], properties such as the mean number and height of local optima tend only to be weakly dependent on

the exact choice being made. For the NK model, $\mathbb{E}[\gamma_d]$ is a non-negative monotonically decreasing function in $d$, and the roughness-to-slope ratio can again range from zero to infinity.

**The eggbox model.** Introduced by [117], the eggbox model is a maximally epistatic, anticorrelated fitness landscape model (i.e., all loci interact with each other up to interactions of order $L$), in which the fitness effect of an allele switches from the highest to lowest value (or vice versa) between genetic backgrounds one step apart. Accordingly, depending on whether two genotypes are separated by an odd or even Hamming distance, their absolute fitness difference is either twice the mean allelic fitness effect or zero. Thus, this model generates an extreme case of reciprocal sign epistasis in which each mutation is either deleterious or compensatory, multiple local optima exist, and $\gamma_d$ accordingly oscillates between -1 and 1.

# E   Analysis for Phenotype Landscapes

## E.1   Phenotype Landscape Models

To demonstrate the applicability of `GraphFLA` to analyzing phenotype landscapes, we consider several well-known systems, including ▶ the RNA secondary structure phenotype landscape for lengths $n = 12$ and $n = 15$ (RNA12, RNA15) representing the RNA sequence's minimum free energy folded secondary structure [136, 87, 137], ▶ the Polyomino lattice self-assembly maps ($S_{2,8}$ and $S_{3,8}$) modelling the topology of protein quaternary structure assembled from interacting constituent tiles [138, 90, 139], and ▶ several hydrophobic-polar (HP) lattice protein models for folding of a sequence into a tertiary structure (two compact models, $HP_{5x5}$ and $HP_{3x3x3}$, and two non-compact ones, $HP_{20}$ and $HP_{25}$) [88, 140, 141]. These phenotype landscapes have been thoroughly studied and compared in [138, 13]

**RNA secondary structure:** The search space $\mathcal{G}$ is made of RNA sequences $g$ where each position can take 4 RNA nucleotide bases ($\mathcal{A}_i = \{A, C, G, U\}$). Phenotypes $\mathcal{P}$ are the secondary structure bonding pattern of the minimum free energy fold of the genotype, represented with the dot-bracket notation [87]. We use the Vienna package [87] with default parameters to convert RNA sequences $g \in \mathcal{G}$ to dot-bracket secondary structures $p \in \mathcal{P}$. Phenotype landscapes is the mapping from the sequence space $\mathcal{G}$ to their phenotypes $\mathcal{P}$, and are represented as RNA-$n$ with sequences of length $n$. As illustration, we consider $n = 12, 15$, resulting in the RNA12 and RNA15 phenotype landscapes.

**HP lattice model:** In this model, genotypes $\mathcal{G}$ comprise sequences of hydrophobic (H) or polar (P) amino acids (alphabet $\mathcal{A}_i = \{H, P\}$) [88, 89]. Phenotypes $\mathcal{P}$ correspond to the unique minimum energy conformation of a genotype when folded onto a 2D (square) or 3D (cubic) lattice. Folds are represented as strings of directional moves (2D: "Up", "Down", "Left", "Right"; 3D additionally "Forward", "Back"). Following [140, 141], only non-adjacent H-H pairs contribute to energy ($E_{HH} = -1$), while $E_{HP} = E_{PP} = 0$. Sequences lacking a unique minimum energy structure are considered undefined. We investigate both non-compact ($HP_L$) and compact phenotype landscapes. In $HP_L$, the phenotype is the minimum energy fold among all possible folds of a specific length. In compact maps (e.g., 2D $HP_{lxw}$ like $HP_{5x5}$; 3D $HP_{lxwxh}$ like $HP_{3x3x3}$), folds are confined to a prescribed grid. These compact models aim to better emulate the globular nature of native proteins [142], significantly reducing the conformational space while enhancing fidelity to observed protein topologies. We analyzed compact ($HP_{3x3x3}$ and $HP_{5x5}$) and non-compact ($HP_{20}$ and $HP_{25}$) landscapes for illustration.

**Polyomino model:** This model represents protein quaternary structure on a 2D square lattice using an assembly kit of tiles. Genotypes $g \in \mathcal{G}$ define this kit of $n_t$ tiles, where each tile edge has one of $n_c$ colors (interface types) denoted by integers. We follow [138, 90], focusing on phenotype landscapes $S_{n_t, n_c}$, specifically $S_{2,8}$. We use $n_c = 8$ colors; tile edges are assigned bases from the alphabet $\mathcal{A}_i = \{0, 1, 2, 3, 4, 6, 7\}$. Interactions are restricted to $1 \leftrightarrow 2$, $3 \leftrightarrow 4$, and $5 \leftrightarrow 6$, while colors 0 and 7 are neutral. The genotype sequence, consisting of bases from $A$, is encoded clockwise onto the four edges of each tile in the kit. Phenotype construction begins by "seeding" the lattice with the first tile. Subsequent tiles from the kit are stochastically placed at complementary interaction sites on the lattice. Assembly halts if no further placements possible or if the structure grows unboundedly. This assembly process is repeated $k = 200$ times. The phenotype is the unique, rotationally invariant, bounded polyomino shape observed across the ensemble of assemblies.

### E.2 Phenotype Landscape Features

Following [138], we consider the following 3 landscape features that are dedicated to characterizing phenotype landscape topography. The analysis results for the previously described landscapes are summarized in Table A1 below.

Table A1: Features of different phenotype landscapes analyzed

| Phenotype landscape | $|\mathcal{A}|$ | $n$ | $|\mathcal{G}|$ | $|\mathcal{P}|$ | $\phi_{\text{del}}$ | $\log_{10} R$ | $\eta_p$ |
|---|---|---|---|---|---|---|---|
| RNA12 | 4 | 12 | $4^{12}$ | 58 | 0.854 | 4.6 | 0.465 |
| RNA15 | 4 | 15 | $4^{15}$ | 432 | 0.650 | 5.9 | 0.482 |
| $S_{2,8}$ | 8 | 8 | $8^8$ | 14 | 0.537 | 5.8 | 0.487 |
| HP5x5 | 2 | 25 | $2^{25}$ | 550 | 0.816 | 4.1 | 0.285 |
| HP3x3x3 | 2 | 27 | $2^{27}$ | 49,808 | 0.939 | 2.2 | 0.115 |
| HP20 | 2 | 20 | $2^{20}$ | 5,311 | 0.976 | 0.7 | 0.102 |
| HP25 | 2 | 25 | $2^{25}$ | 107,337 | 0.977 | 0.9 | 0.099 |

**Redundancy.** Denoted by $R$, it is defined as the average number of distinct genotypes $g \in \mathcal{G}$ that map to each non-deleterious phenotype $p \in \mathcal{P}$. Redundancy is intrinsically linked to the average size of phenotypically neutral networks, which consist of sets of genotypes that share the same phenotype and are often connected by single mutations.

**Deleterious frequency.** Denoted as $\phi_{\text{del}}$, this metric represents the fraction of the entire genotype space $\mathcal{G}$ that is occupied by genotypes failing to map to a well-defined, functional phenotype. The nature of a deleterious phenotype is model-specific:

- In RNA secondary structure landscapes, such as RNA12 and RNA15, a deleterious phenotype corresponds to an unfolded RNA sequence that lacks any defined secondary structure.
- For Hydrophobic-Polar (HP) lattice protein models, like $HP_{5x5}$ (a compact 2D model) or $HP_{20}$ (a non-compact model), a deleterious outcome signifies an amino acid sequence that does not fold into a unique minimum energy conformation.
- In the context of Polyomino lattice self-assembly models, for example $S_{2,8}$ or $S_{3,8}$ which model protein quaternary structure, a deleterious genotype is one that results in an unbounded or non-deterministic assembly process.

**Phenotypic neutrality.** Denoted as $\eta_p$, it is the average proportion of mutational neighbors of a genotype $g$ (i.e., genotypes $g' \in \mathcal{N}(g)$) that exhibit the same phenotype as $g$. This average is computed over all genotypes in $\mathcal{G}$ that correspond to non-deleterious phenotypes. The value of $\eta_p$ provides a measure of local neutral connectivity within the phenotypic landscape, indicating the extent to which mutations can occur without altering the observable phenotype. This concept of phenotypic neutrality is distinct from fitness-based neutrality ($\eta$) defined in C.15, as it specifically pertains to the preservation of phenotype rather than fitness.

## F Directed Evolution

This section provides details regarding how each directed evolution (DE) approach in Section 4.5 is implemented. Specifically, we considered 5 DE variants. For each approach, the results are evaluated by the highest fitness variant they identified. To enable comparison across tasks, this is reported as percentiles from 0 to 1, where 1 represents the fitness of the global optimum. Each approach is run with random initialization for 100 repetitions, and we report the average in Section 4.5.

**Basic DE.** In this simplest form, DE is implemented via a greedy adaptive walk algorithm starting from a random variant $g \in \mathcal{G}$. For each step, it exhaustively searches within its neighborhood for the single-point mutation $g \to g', (g' \in \mathcal{N}(g))$ that yields the highest fitness increase (i.e, $\Delta f$), until a local optimum is reached.

**MLDE.** In this paradigm, a supervised ML model (TabPFN [27])[2] is trained on a set of $N$ randomly sampled protein variants from $\mathcal{G}$ along with their fitness. Following [144, 145, 85], we set $N = 384$

---

[2]We also experimented with other common models including XGBoost [143] and a convolutional neural network (CNN) [108]. We report results for TabPFN in Section 4.5 as it yielded the highest performance.

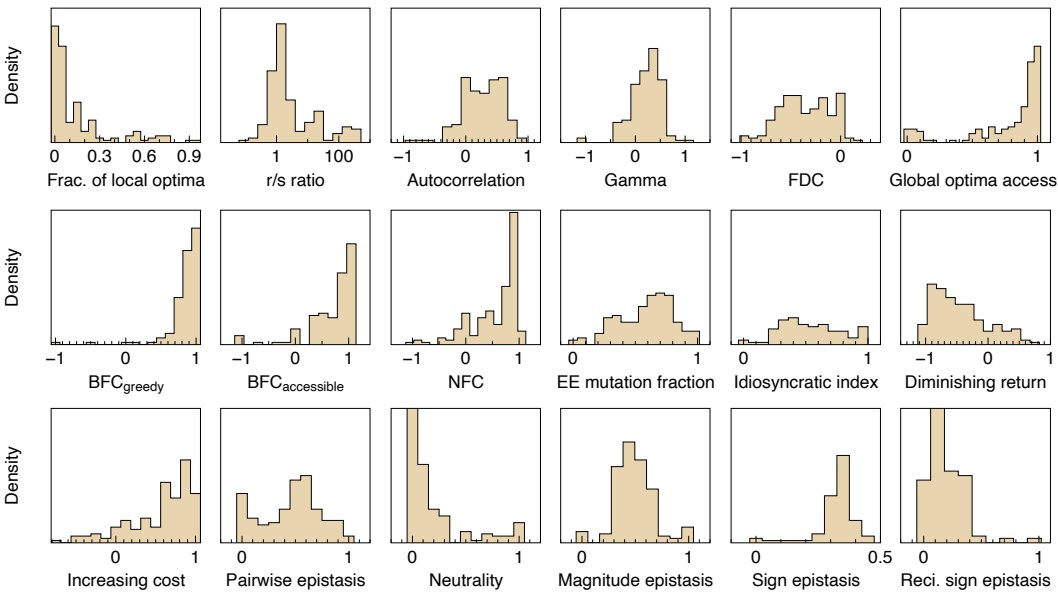

Figure A1: **Distribution of landscape features across our combinatorially complete datasets.**

as performance typically plateaus at this value. Increasing the sample size beyond this point yields only marginal gains while incurring higher costs. During training, protein sequences were represented using one-hot encoding flattened over the mutated sites. The trained model was then employed to predict fitness values for the entire library, with the top 96 predicted variants selected for evaluation.

**MLDE with zero-shot warm start.** Instead of randomly selecting the training set from the entire search space $\mathcal{G}$, this approach first uses a zero-shot predictor (ESM [146] here) to identify prominent regions composed of the top ranked 10% variants. The initial training samples are then sampled from these regions to bias the learning towards them. The subsequent steps are the same as in MLDE.

**ALDE.** This active learning-assisted DE implements an iterative learning strategy with 3 or 5 rounds. In each round, the ML model (TabPFN) is trained on all data acquired up to that point. The initial round involves the same random sampling as in MLDE with $N = 96$. For subsequent rounds, the ML model serves as an acquisition function to rank all variants in the library, thus guiding the selection of the next batch of variants for fitness evaluation. After the final round, the trained model predicts fitness values, and the top 96 variants are selected for analysis for evaluation.

**ALDE with zero-shot warm start.** This approach is almost identical to ALDE, except that instead of randomly sampling the initial 96 variants, it employs the same zero-shot ranking as in MLDE with zero-shot warm start.

## G    Additional Results

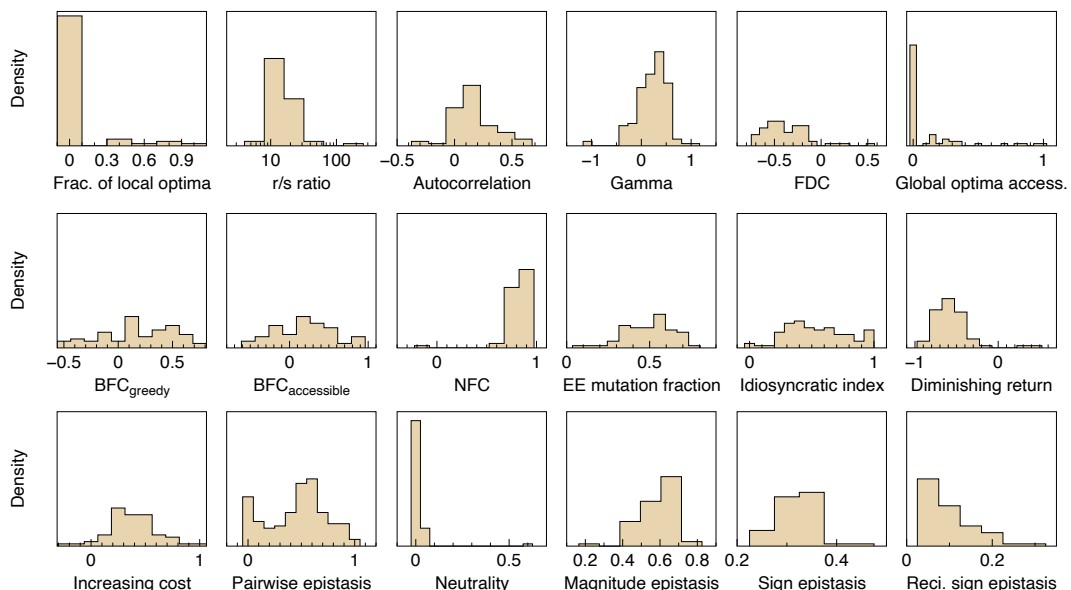

Figure A2: **Distribution of landscape features across ProteinGym tasks.**

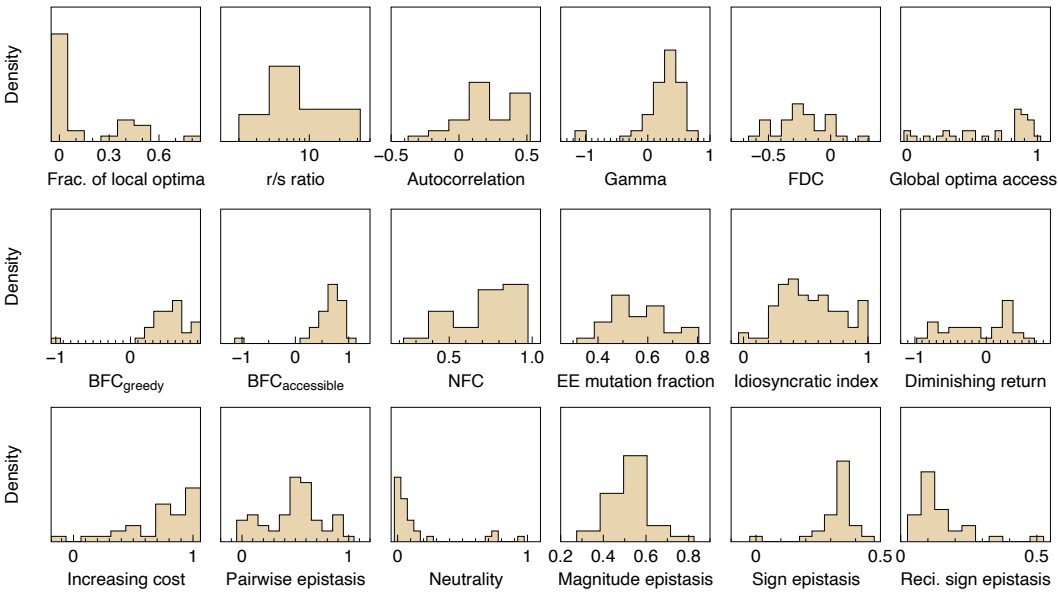

Figure A3: **Distribution of landscape features across ProteinGym tasks.**

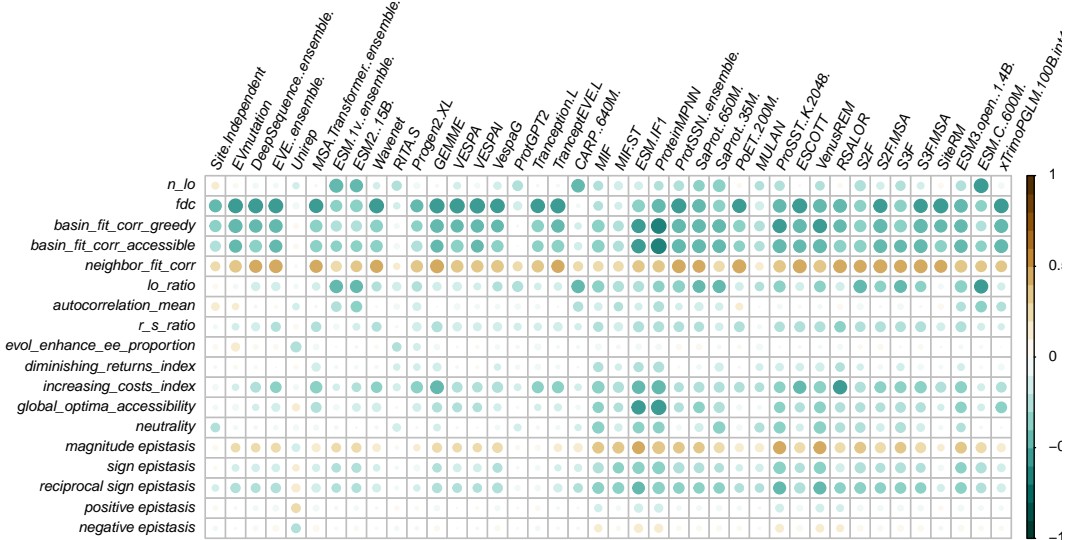

Figure A4: **Spearman correlation between fitness landscape features and the performance of zero-shot protein fitness models on ProteinGym tasks.** Due to the vast number of baselines on ProteinGym leaderboard, we only display representative ones here for each model series. Model performances are reported as Spearman's $\rho$ between predicted and true fitness.

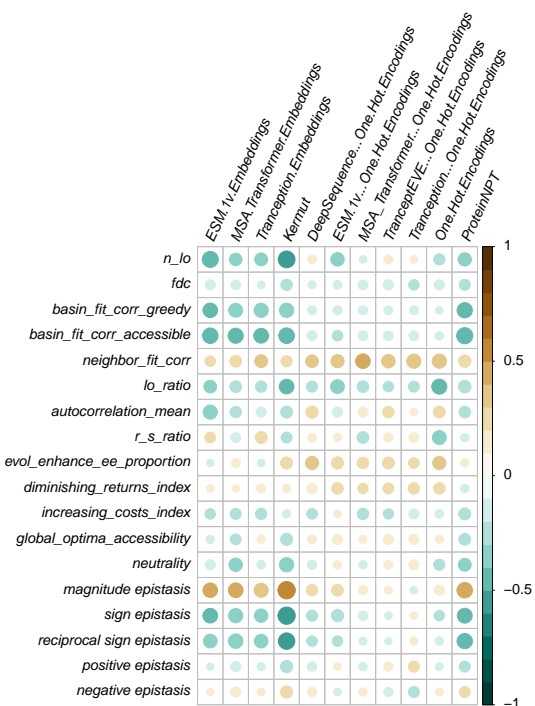

Figure A5: **Spearman correlation between fitness landscape features and the performance of supervised protein fitness models on ProteinGym tasks.** Model performances are reported as Spearman's $\rho$ between predicted and true fitness.

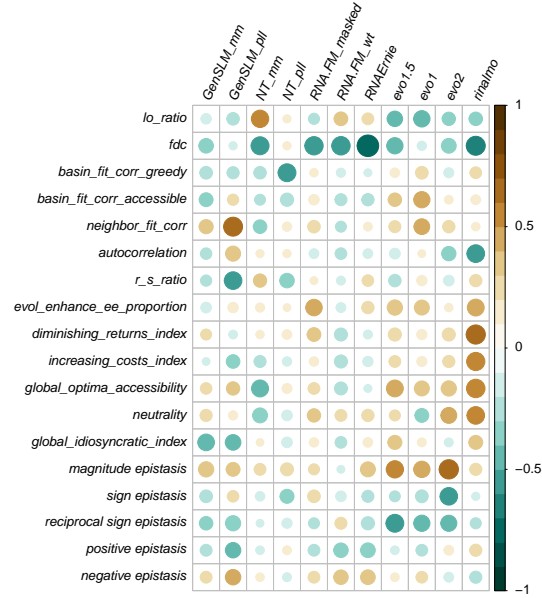

Figure A6: **Spearman correlation between fitness landscape features and the performance of supervised RNA fitness models on RNAGym tasks.** Model performances are reported as Spearman's $\rho$ between predicted and true fitness.

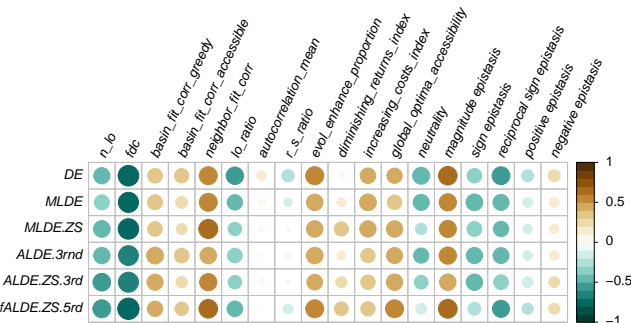

Figure A7: **Spearman correlation between fitness landscape features and the performance of 5 directed evolution (DE) approaches on 20 combinatorially complete protein fitness landscapes.** Model performances are reported as the maximum fitness (normalized to [0,1] by taking percentiles) reached, averaged across 100 randomly initialized repeatations.

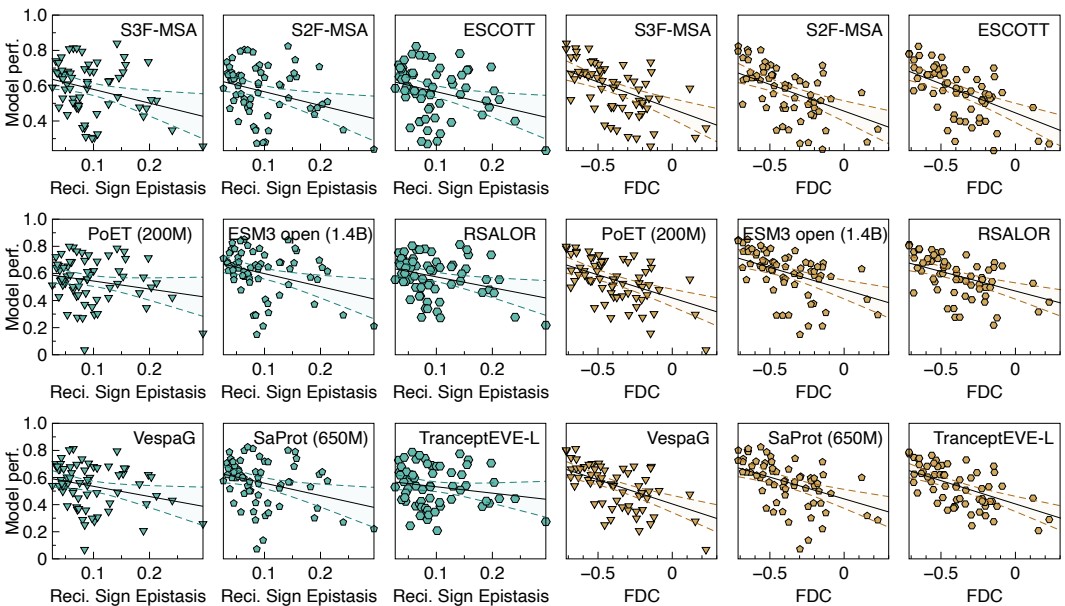

Figure A8: **Influence of landscape features on protein fitness model performance on ProteinGym.** We plot the distribution of model (name specified in each plot) performance ($y$-axis; measured as Spearman's $\rho$) against landscape features ($x$-axis). Straight lines show a fit of the linear regression model, and shaded regions depict the 95% confidence intervals. References: S3F-MSA [147], ESCOTT [148], PoET [149], ESM3 [150], RSALOR [151], VespaG [152], SaProt [153], and TranceptEVE-L [154].

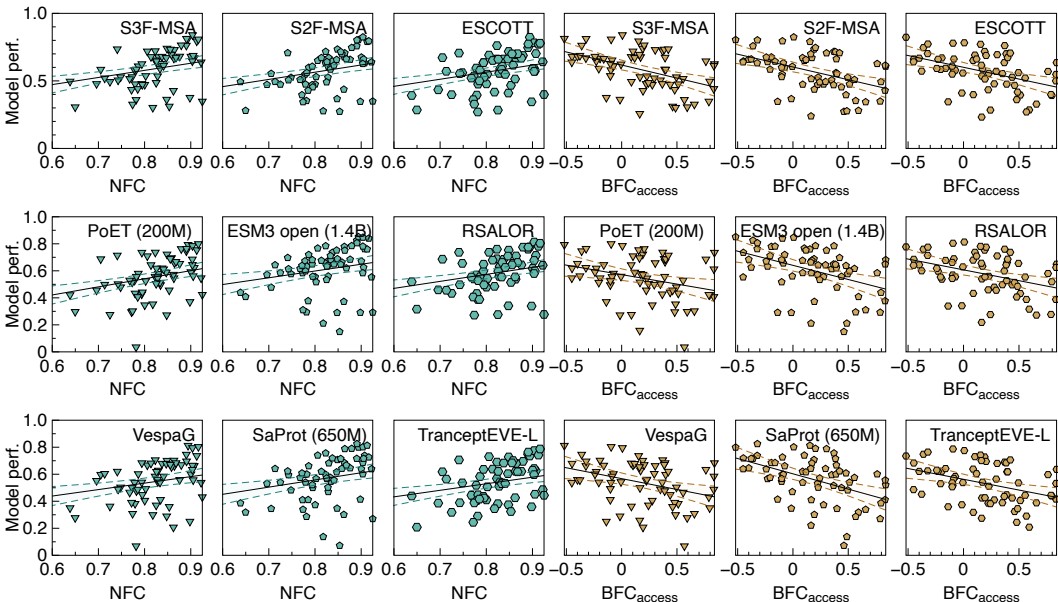

Figure A9: **Influence of landscape features on protein fitness model performance on ProteinGym.** We plot the distribution of model (name specified in each plot) performance ($y$-axis; measured as Spearman's $\rho$) against landscape features ($x$-axis). Straight lines show a fit of the linear regression model, and shaded regions depict the 95% confidence intervals.

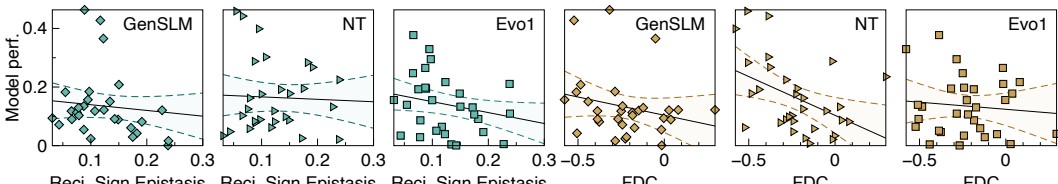

Figure A10: **Influence of landscape features on RNA fitness model performance on RNAGym.** We plot the distribution of model (name specified in each plot) performance ($y$-axis; measured as Spearman's $\rho$) against landscape features ($x$-axis). Straight lines show a fit of the linear regression model, and shaded regions depict the 95% confidence intervals. References: GenSLM [155], NT [156], and Evo1 [63].

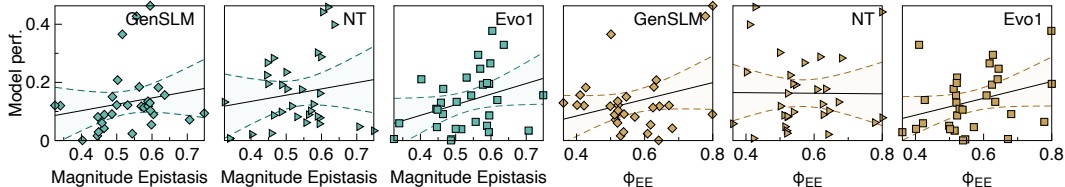

Figure A11: **Influence of landscape features on RNA fitness model performance on RNAGym.** We plot the distribution of model (name specified in each plot) performance ($y$-axis; measured as Spearman's $\rho$) against landscape features ($x$-axis). Straight lines show a fit of the linear regression model, and shaded regions depict the 95% confidence intervals.

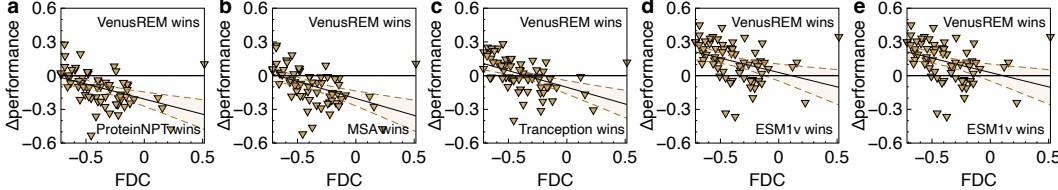

Figure A12: Difference in performance ($y$-axis) between 5 supervised baselines and VenusREM in ProteinGym is plotted against landscape features on the $x$-axis. Straight lines show a fit of the linear regression model, and shaded regions depict the 95% confidence intervals. References: VenusREM [24], ProteinNP [77], MSA [157], and Tranception [158].

Table A2: Summary of landscape features for different microbial community-function landscapes.

| Feature | [159] | [52] | [54] | [53] | [50] | [111] |
|---|---|---|---|---|---|---|
| $n$ | 63 | 54 | 279 | 614 | 21198 | 1561 |
| $\phi_{\mathrm{lo}}$ | 0.032 | 0.074 | 0.075 | 0.040 | 0.158 | 0.584 |
| $r/s$ | 0.944 | 0.863 | 1.417 | 1.778 | 3.087 | 2.294 |
| $\rho_a$ | 0.528 | 0.491 | 0.556 | 0.490 | 0.559 | 0.196 |
| $\gamma$ | 0.384 | 0.345 | 0.449 | 0.330 | 0.146 | 0.161 |
| FDC | -0.559 | -0.780 | -0.587 | -0.161 | -0.378 | -0.487 |
| $\alpha_{\mathrm{go}}$ | 0.984 | 0.944 | 0.888 | 0.853 | 0.003 | 0.003 |
| $\mathrm{BFC}_{\mathrm{greedy}}$ | 1.000 | 0.800 | 0.861 | 0.724 | 0.502 | 0.156 |
| $\mathrm{BFC}_{\mathrm{acc}}$ | 1.000 | 1.000 | 0.861 | 0.950 | 0.409 | 0.025 |
| NFC | 0.919 | 0.902 | 0.873 | 0.820 | 0.790 | 0.635 |
| $\phi_{\mathrm{EE}}$ | 0.683 | 0.709 | 0.644 | 0.625 | 0.535 | 0.402 |
| $I_{\mathrm{id}}$ | 0.398 | 0.395 | 0.539 | 0.639 | 0.552 | 0.605 |
| $\epsilon_{\mathrm{DR}}$ | -0.741 | -0.314 | -0.603 | -0.194 | 0.043 | -0.319 |
| $\epsilon_{\mathrm{IC}}$ | 0.648 | 0.871 | 0.574 | 0.728 | 0.816 | 0.583 |
| $\epsilon_{(2)}$ | 0.861 | 0.869 | 0.633 | 0.361 | 0.547 | 0.675 |
| $\eta$ | 0.038 | 0.000 | 0.527 | 0.183 | 0.000 | 0.000 |
| $\epsilon_{\mathrm{mag}}$ | 0.489 | 0.543 | 0.496 | 0.447 | 0.404 | 0.457 |
| $\epsilon_{\mathrm{sign}}$ | 0.431 | 0.413 | 0.396 | 0.375 | 0.395 | 0.354 |
| $\epsilon_{\mathrm{reci}}$ | 0.080 | 0.043 | 0.108 | 0.178 | 0.201 | 0.189 |
| $\epsilon_{\mathrm{pos}}$ | 0.764 | 0.768 | 0.742 | 0.840 | 0.841 | 0.774 |
| $\epsilon_{\mathrm{neg}}$ | 0.236 | 0.232 | 0.258 | 0.160 | 0.159 | 0.226 |

Table A3: Selected landscape features for combinatorially complete fitness landscapes

| Author | Ref. | SubID | Space | Size | $\phi_{\text{lo}}(\%)$ | $\rho_a$ (%) | NFC | $\gamma$ | $\epsilon_{\text{reci}}$ | $\epsilon_{\text{DR}}$ | FDC | $\phi_{\text{EE}}$ | $\eta$ |
|---|---|---|---|---|---|---|---|---|---|---|---|---|---|
| Kuo2020 | [160] | - | Nucleotide | $4^9 = 262,144$ | 0.012 | 0.573 | 0.917 | 0.454 | 0.206 | 0.391 | -0.235 | 0.669 | 0.074 |
| Papkou2023 | [4] | DHFR | Nucleotide | $4^9 = 262,144$ | 0.004 | 0.641 | 0.954 | 0.399 | 0.255 | 0.065 | -0.140 | 0.553 | 0.076 |
| PodgornaiaL15 | [47] | PhoQ | Protein | $20^4 = 160,000$ | 0.013 | 0.241 | 0.540 | NaN | 0.234 | 0.469 | -0.196 | 0.609 | 0.282 |
| Wu2016 | [59] | GB1 | Protein | $20^4 = 160,000$ | 0.001 | 0.406 | 0.872 | 0.440 | 0.221 | 0.499 | -0.123 | 0.621 | 0.717 |
| Jalal2020 | [161] | ParB | Protein | $20^4 = 160,000$ | 0.005 | 0.450 | 0.594 | 0.120 | 0.284 | -0.107 | -0.310 | 0.519 | 0.002 |
| Jalal2020 | [161] | Noc | Protein | $20^4 = 160,000$ | 0.004 | 0.346 | 0.388 | 0.131 | 0.331 | -0.164 | -0.313 | 0.426 | 0.003 |
| Tu2022 | [162] | TEV | Protein | $20^4 = 160,000$ | 0.007 | 0.053 | 0.242 | 0.031 | 0.309 | -0.655 | 0.008 | 0.510 | 0.141 |
| Tu2022 | [162] | T7 | Protein | $20^3 = 8,000$ | 0.018 | -0.025 | -0.172 | NaN | 0.341 | -0.855 | 0.036 | 0.380 | 0.018 |
| Johnston2024 | [7] | TrpB4 | Protein | $20^4 = 160,000$ | 0.005 | 0.399 | 0.831 | 0.195 | 0.294 | 0.393 | -0.112 | 0.562 | 0.343 |
| Johnston2024 | [7] | TrpB3A | Protein | $20^3 = 8,000$ | 0.008 | 0.161 | 0.485 | 0.036 | 0.256 | -0.239 | -0.178 | 0.621 | 0.167 |
| Johnston2024 | [7] | TrpB3B | Protein | $20^3 = 8,000$ | 0.009 | 0.074 | 0.306 | 0.025 | 0.295 | -0.326 | -0.021 | 0.555 | 0.205 |
| Johnston2024 | [7] | TrpB3C | Protein | $20^3 = 8,000$ | 0.008 | 0.115 | 0.395 | 0.020 | 0.284 | -0.585 | -0.065 | 0.572 | 0.089 |
| Johnston2024 | [7] | TrpB3D | Protein | $20^3 = 8,000$ | 0.006 | 0.251 | 0.676 | 0.145 | 0.264 | 0.325 | -0.125 | 0.611 | 0.175 |
| Johnston2024 | [7] | TrpB3E | Protein | $20^3 = 8,000$ | 0.012 | 0.113 | 0.498 | 0.030 | 0.316 | 0.153 | -0.036 | 0.500 | 0.339 |
| Johnston2024 | [7] | TrpB3F | Protein | $20^3 = 8,000$ | 0.011 | 0.105 | 0.390 | 0.026 | 0.309 | -0.236 | -0.197 | 0.500 | 0.179 |
| Johnston2024 | [7] | TrpB3G | Protein | $20^3 = 8,000$ | 0.010 | 0.162 | 0.543 | 0.043 | 0.305 | 0.184 | -0.176 | 0.514 | 0.313 |
| Johnston2024 | [7] | TrpB3H | Protein | $20^3 = 8,000$ | 0.015 | 0.046 | 0.253 | 0.011 | 0.322 | 0.110 | -0.012 | 0.475 | 0.346 |
| Johnston2024 | [7] | TrpB3I | Protein | $20^3 = 8,000$ | 0.002 | 0.433 | 0.856 | 0.275 | 0.171 | 0.373 | -0.240 | 0.726 | 0.145 |
| Domingo2018 | [42] | - | Nucleotide | $2^6 \times 3^4 = 5,184$ | 0.021 | 0.517 | 0.777 | 0.114 | 0.182 | -0.757 | -0.506 | 0.681 | 0.076 |
| Phillips2021 | [109] | CR6261-h1 | Mutation | $2^{11} = 2,048$ | 0.011 | 0.561 | 0.820 | 0.538 | 0.044 | -0.956 | -0.379 | 0.794 | 0.075 |
| Phillips2021 | [109] | CR6261-h9 | Mutation | $2^{11} = 2,048$ | 0.019 | 0.547 | 0.794 | 0.420 | 0.026 | -0.901 | -0.575 | 0.794 | 0.037 |
| Phillips2021 | [109] | CR9114-h1 | Mutation | $2^{16} = 65,536$ | 0.013 | 0.765 | 0.956 | 0.391 | 0.118 | -0.945 | -0.219 | 0.673 | 0.093 |
| Phillips2021 | [109] | CR9114-h3 | Mutation | $2^{16} = 65,536$ | 0.563 | 0.411 | 0.198 | NaN | 0.071 | -0.400 | -0.354 | 0.586 | 0.019 |
| Phillips2021 | [109] | CR9114-flueB | Mutation | $2^{16} = 65,536$ | 0.972 | -0.062 | -0.162 | NaN | 0.118 | -0.304 | -0.089 | 0.278 | 0.018 |
| Phillips2023 | [163] | CH65-SI06 | Mutation | $2^{16} = 65,536$ | 0.481 | 0.655 | 0.741 | 0.442 | 0.082 | -0.474 | -0.498 | 0.672 | 0.026 |
| Phillips2023 | [163] | CH65-MA90 | Mutation | $2^{16} = 65,536$ | 0.000 | 0.831 | 0.991 | 0.481 | 0.078 | -0.320 | -0.534 | 0.798 | 0.070 |
| Phillips2023 | [163] | CH65-G189E | Mutation | $2^{16} = 65,536$ | 0.144 | 0.664 | 0.860 | 0.408 | 0.072 | 0.569 | -0.554 | 0.763 | 0.034 |
| Westmann24 | [34] | - | Nucleotide | $4^8 = 65,536$ | 0.118 | 0.202 | 0.362 | 0.072 | 0.305 | -0.170 | -0.111 | 0.457 | 0.053 |
| Soo2021 | [110] | 30C | Nucleotide | $4^8 = 65,536$ | 0.015 | 0.445 | 0.773 | 0.219 | 0.231 | -0.743 | -0.260 | 0.648 | 0.007 |
| Soo2021 | [110] | 37C | Nucleotide | $4^8 = 65,536$ | 0.021 | 0.419 | 0.748 | 0.196 | 0.240 | -0.785 | -0.004 | 0.639 | 0.008 |
| Wong2018 | [164] | BRCA2 | Nucleotide | $32,768$ | 0.022 | 0.535 | 0.897 | 0.444 | 0.290 | 0.247 | 0.019 | 0.529 | 0.143 |

**Table A3 – continued from previous page**

| Author | Ref. | SubID | Space | Size | $\phi_{\text{lo}}$ | $\rho_a$ | NFC | $\gamma$ | $\epsilon_{\text{reci}}$ | $\epsilon_{\text{DR}}$ | FDC | $\phi_{\text{EE}}$ | $\eta$ |
|---|---|---|---|---|---|---|---|---|---|---|---|---|---|
| Wong2018 | [164] | SMN1 | Nucleotide | $32,768$ | 0.009 | 0.502 | 0.861 | 0.421 | 0.183 | 0.290 | 0.011 | 0.668 | 0.096 |
| Wong2018 | [164] | IKBKAP | Nucleotide | $32,768$ | 0.036 | 0.392 | 0.808 | 0.380 | 0.375 | 0.216 | 0.004 | 0.372 | 0.102 |
| Moulana2022 | [165] | ACE | Mutation | $2^{15} = 32,768$ | 0.001 | 0.833 | 0.993 | 0.536 | 0.058 | -0.875 | -0.485 | 0.848 | 0.050 |
| Moulana2023 | [166] | CB6 | Mutation | $2^{15} = 32,768$ | 0.006 | 0.811 | 0.965 | 0.368 | 0.107 | -0.593 | -0.394 | 0.774 | 0.040 |
| Moulana2023 | [166] | CoV555 | Mutation | $2^{15} = 32,768$ | 0.020 | 0.770 | 0.957 | 0.180 | 0.201 | -0.707 | -0.150 | 0.598 | 0.040 |
| Moulana2023 | [166] | REGN10987 | Mutation | $2^{15} = 32,768$ | 0.007 | 0.716 | 0.924 | 0.222 | 0.160 | -0.817 | -0.449 | 0.680 | 0.055 |
| Moulana2023 | [166] | S309 | Mutation | $2^{15} = 32,768$ | 0.006 | 0.775 | 0.962 | 0.348 | 0.115 | -0.535 | -0.339 | 0.696 | 0.071 |
| Bendixsen2019 | [167] | HDV | Mutation | $2^{14} = 16,384$ | 0.071 | 0.239 | 0.240 | 0.218 | 0.410 | 0.192 | 0.140 | 0.271 | 0.232 |
| Bendixsen2019 | [167] | Ligase | Mutation | $2^{14} = 16,384$ | 0.004 | 0.347 | 0.803 | 0.510 | 0.137 | 0.418 | -0.437 | 0.705 | 0.810 |
| Poelwijk2019 | [168] | eqFP611 | Mutation | $2^{13} = 8,192$ | 0.009 | 0.675 | 0.979 | 0.452 | 0.187 | -0.281 | 0.077 | 0.623 | 0.146 |
| Lite2020 | [169] | ParD2 | Protein | $20^3 = 8,000$ | 0.001 | 0.577 | 0.960 | 0.319 | 0.113 | -0.348 | -0.302 | 0.796 | 0.058 |
| Lite2020 | [169] | ParD3 | Protein | $20^3 = 8,000$ | 0.001 | 0.579 | 0.957 | 0.287 | 0.068 | -0.560 | -0.245 | 0.851 | 0.052 |
| Centurion2019 | [170] | - | Mutation | $2^{10} \times 3 = 3,072$ | 0.032 | 0.657 | 0.919 | 0.366 | 0.155 | -0.850 | -0.050 | 0.751 | 0.053 |
| Schulz2025 | [171] | - | Mutation | $2^{10} = 1,024$ | 0.036 | 0.653 | 0.911 | 0.215 | 0.153 | -0.820 | -0.223 | 0.689 | 0.016 |
| Bakerlee2022 | [39] | hap-4NQ0 | Mutation | $2^{10} = 1,024$ | 0.479 | 0.029 | 0.025 | 0.175 | 0.273 | -0.536 | -0.037 | 0.351 | 0.953 |
| Bakerlee2022 | [39] | hap-37C | Mutation | $2^{10} = 1,024$ | 0.630 | 0.013 | 0.039 | 0.081 | 0.422 | 0.044 | 0.043 | 0.278 | 0.977 |
| Bakerlee2022 | [39] | hap-gu | Mutation | $2^{10} = 1,024$ | 0.681 | -0.009 | -0.004 | -0.039 | 0.377 | -0.377 | 0.034 | 0.205 | 0.964 |
| Bakerlee2022 | [39] | hap-salt | Mutation | $2^{10} = 1,024$ | 0.774 | -0.101 | 0.045 | 0.299 | 0.282 | 0.408 | 0.015 | 0.245 | 0.959 |
| Bakerlee2022 | [39] | hap-suloc | Mutation | $2^{10} = 1,024$ | 0.510 | -0.024 | -0.013 | -0.036 | 0.434 | 0.114 | -0.046 | 0.224 | 0.992 |
| Bakerlee2022 | [39] | hap-YPDA | Mutation | $2^{10} = 1,024$ | 0.633 | 0.007 | 0.012 | 0.083 | 0.378 | -0.943 | 0.039 | 0.271 | 0.972 |
| Bakerlee2022 | [39] | hom-4NQO | Mutation | $2^{10} = 1,024$ | 0.570 | 0.065 | 0.048 | 0.185 | 0.305 | -0.417 | -0.033 | 0.326 | 0.936 |
| Bakerlee2022 | [39] | hom-37C | Mutation | $2^{10} = 1,024$ | 0.535 | 0.022 | 0.024 | 0.052 | 0.412 | -0.163 | -0.015 | 0.235 | 0.992 |
| Bakerlee2022 | [39] | hom-gu | Mutation | $2^{10} = 1,024$ | 0.740 | -0.043 | 0.012 | 0.086 | 0.357 | -0.452 | 0.000 | 0.209 | 0.925 |
| Bakerlee2022 | [39] | hom-salt | Mutation | $2^{10} = 1,024$ | 0.770 | -0.129 | 0.027 | 0.190 | 0.250 | -0.005 | -0.032 | 0.277 | 0.881 |
| Bakerlee2022 | [39] | hom-suloc | Mutation | $2^{10} = 1,024$ | 0.549 | 0.005 | -0.004 | 0.001 | 0.320 | -1.000 | 0.026 | 0.325 | 0.982 |
| Bakerlee2022 | [39] | hom-YPDA | Mutation | $2^{10} = 1,024$ | 0.569 | 0.031 | 0.020 | 0.103 | 0.355 | -0.707 | -0.051 | 0.268 | 0.958 |
| Bank2016 | [106] | - | Mutation | $640$ | 0.027 | 0.476 | 0.807 | 0.210 | 0.117 | 0.190 | -0.411 | 0.780 | 0.000 |
| Bank2016 | [106] | - | Mutation | $2^6 = 64$ | 0.062 | 0.403 | 0.886 | 0.606 | 0.125 | -0.965 | -0.126 | 0.719 | 0.250 |
| Wu2020 | [172] | Bei89 | Mutation | $576$ | 0.005 | 0.651 | 0.980 | 0.475 | 0.024 | -0.556 | -0.428 | 0.916 | 0.008 |
| Wu2020 | [172] | Bk79 | Mutation | $576$ | 0.002 | 0.627 | 0.967 | 0.475 | 0.055 | -0.354 | -0.519 | 0.856 | 0.006 |
| Wu2020 | [172] | Bris07L194 | Mutation | $576$ | 0.007 | 0.680 | 0.951 | 0.466 | 0.141 | -0.087 | -0.463 | 0.726 | 0.009 |
| Wu2020 | [172] | Bris07P194 | Mutation | $576$ | 0.052 | 0.359 | 0.719 | 0.191 | 0.272 | 0.288 | -0.405 | 0.540 | 0.006 |
| Wu2020 | [172] | HK68 | Mutation | $576$ | 0.016 | 0.595 | 0.946 | 0.522 | 0.070 | -0.711 | -0.260 | 0.846 | 0.007 |

| Author | Ref. | SubID | Space | Size | $\phi_{\mathrm{lo}}$ | $\rho_a$ | NFC | $\gamma$ | $\epsilon_{\mathrm{reci}}$ | $\epsilon_{\mathrm{DR}}$ | FDC | $\phi_{\mathrm{EE}}$ | $\eta$ |
|---|---|---|---|---|---|---|---|---|---|---|---|---|---|
| Wu2020 | [172] | Mos99 | Mutation | 576 | 0.012 | 0.632 | 0.941 | 0.399 | 0.089 | -0.621 | -0.498 | 0.809 | 0.006 |
| Wu2020 | [172] | NDako16 | Mutation | 576 | 0.003 | 0.700 | 0.971 | 0.523 | 0.040 | -0.251 | -0.644 | 0.875 | 0.011 |
| Lunzer2005 | [6] | fitness | Protein | 512 | 0.002 | 0.496 | 0.918 | 0.639 | 0.019 | -0.978 | -0.586 | 0.939 | 0.003 |
| Lunzer2005 | [6] | NAD | Protein | 512 | 0.002 | 0.561 | 0.953 | -0.024 | 0.000 | -0.762 | -0.503 | 1.000 | 0.036 |
| Lunzer2005 | [6] | NADP | Protein | 512 | 0.002 | 0.685 | 0.972 | -0.004 | 0.000 | -0.671 | -0.665 | 1.000 | 0.000 |
| Doud2024 | [173] | base | Mutation | $2^9 = 512$ | 0.016 | 0.568 | 0.921 | 0.549 | 0.105 | -0.462 | -0.346 | 0.740 | 0.005 |
| Doud2024 | [173] | LamB | Mutation | $2^9 = 512$ | 0.012 | 0.625 | 0.939 | 0.511 | 0.133 | -0.057 | -0.548 | 0.729 | 0.011 |
| Doud2024 | [173] | Lspec | Mutation | $2^9 = 512$ | 0.026 | 0.586 | 0.932 | 0.628 | 0.107 | -0.594 | -0.270 | 0.722 | 0.008 |
| Doud2024 | [173] | OmpF | Mutation | $2^9 = 512$ | 0.040 | 0.558 | 0.901 | 0.538 | 0.124 | -0.460 | -0.265 | 0.681 | 0.009 |
| Doud2024 | [173] | Ospec | Mutation | $2^9 = 512$ | 0.020 | 0.541 | 0.931 | 0.619 | 0.097 | -0.859 | -0.108 | 0.753 | 0.011 |
| Colunga2024 | [51] | colorants | Mutation | $2^8 = 256$ | 0.004 | 0.730 | 0.996 | 0.162 | 0.001 | -0.507 | -0.854 | 0.981 | 0.015 |
| Colunga2024 | [51] | pseudo | Mutation | $2^8 = 256$ | 0.035 | 0.396 | 0.782 | 0.290 | 0.198 | -0.838 | -0.306 | 0.615 | 0.055 |
| Hall2020 | [174] | NfsA-2039 | Mutation | $2^7 = 128$ | 0.016 | 0.586 | 0.945 | 0.404 | 0.074 | -0.461 | -0.363 | 0.738 | 0.518 |
| Hall2020 | [174] | NfsA-3637 | Mutation | $2^7 = 128$ | 0.031 | 0.601 | 0.946 | 0.392 | 0.071 | -0.496 | -0.509 | 0.770 | 0.107 |
| Frohlich2021 | [175] | CAZtraj1 | Mutation | $2^4 = 16$ | 0.062 | 0.228 | 0.867 | 0.322 | 0.083 | -0.333 | -0.714 | 0.719 | 0.156 |
| Frohlich2021 | [175] | CAZtraj2 | Mutation | $2^6 = 64$ | 0.031 | 0.523 | 0.942 | 0.600 | 0.026 | -0.149 | -0.446 | 0.880 | 0.047 |
| Frohlich2021 | [175] | CAZtraj3 | Mutation | $2^6 = 64$ | 0.016 | 0.513 | 0.962 | 0.320 | 0.100 | -0.367 | -0.433 | 0.647 | 0.258 |
| Frohlich2021 | [175] | PIPtraj1 | Mutation | $2^4 = 16$ | 0.125 | 0.182 | 0.745 | 0.044 | 0.125 | -0.621 | 0.000 | 0.625 | 0.688 |
| Frohlich2021 | [175] | PIPtraj2 | Mutation | $2^6 = 64$ | 0.016 | 0.418 | 0.862 | 0.423 | 0.106 | -0.171 | -0.373 | 0.649 | 0.455 |
| Frohlich2021 | [175] | PIPtraj3 | Mutation | $2^6 = 64$ | 0.062 | 0.581 | 0.958 | 0.030 | 0.151 | -0.719 | -0.441 | 0.587 | 0.032 |
| Hall2010 | [176] | Haploid | Mutation | $2^6 = 64$ | 0.141 | 0.239 | 0.483 | -0.150 | 0.292 | -0.814 | -0.263 | 0.469 | 0.245 |
| Hall2010 | [176] | Diploid | Mutation | $2^6 = 64$ | 0.125 | 0.214 | 0.451 | -0.235 | 0.342 | -0.746 | -0.316 | 0.443 | 0.130 |
| Tamer2019 | [177] | kcat-trajr | Mutation | $2^5 = 32$ | 0.125 | 0.222 | 0.706 | 0.253 | 0.163 | -0.941 | -0.214 | 0.575 | 0.163 |
| Tamer2019 | [177] | kcat-trajg | Mutation | $2^5 = 32$ | 0.125 | 0.256 | 0.800 | 0.480 | 0.150 | -0.961 | -0.357 | 0.662 | 0.075 |
| Tamer2019 | [177] | ki-trajr | Mutation | $2^5 = 32$ | 0.031 | 0.471 | 0.966 | 0.192 | 0.000 | -0.066 | -0.952 | 1.000 | 0.000 |
| Tamer2019 | [177] | ki-trajg | Mutation | $2^5 = 32$ | 0.031 | 0.472 | 0.971 | 0.510 | 0.000 | 0.333 | -0.959 | 1.000 | 0.000 |
| Lozovsky2021 | [178] | ic50-c57 | Mutation | $2^4 = 16$ | 0.125 | 0.006 | 0.320 | -0.059 | 0.133 | -0.271 | -0.594 | 0.414 | 0.000 |
| Lozovsky2021 | [178] | ic50-c58 | Mutation | $2^4 = 16$ | 0.188 | 0.034 | 0.342 | -0.330 | 0.125 | -0.233 | -0.539 | 0.483 | 0.000 |
| Lozovsky2021 | [178] | ic50-c59 | Mutation | $2^4 = 16$ | 0.125 | -0.039 | 0.132 | -0.169 | 0.167 | 0.594 | -0.606 | 0.407 | 0.000 |
| Lozovsky2021 | [178] | ic50-c60 | Mutation | $2^4 = 16$ | 0.250 | -0.167 | -0.064 | -0.232 | 0.222 | -0.477 | -0.184 | 0.400 | 0.240 |
| Lozovsky2021 | [178] | ic50-c61 | Mutation | $2^4 = 16$ | 0.312 | -0.139 | -0.010 | 0.015 | 0.167 | -0.394 | -0.536 | 0.273 | 0.000 |
| Hall2019 | [179] | Acetate | Mutation | $2^5 = 32$ | 0.094 | 0.171 | 0.494 | -0.060 | 0.200 | -0.757 | -0.250 | 0.500 | 0.100 |
| Hall2019 | [179] | Beef | Mutation | $2^5 = 32$ | 0.062 | 0.365 | 0.878 | 0.558 | 0.125 | -0.477 | -0.561 | 0.738 | 0.075 |

| Author | Ref. | SubID | Space | Size | $\phi_{lo}$ | $\rho_a$ | NFC | $\gamma$ | $\epsilon_{reci}$ | $\epsilon_{DR}$ | FDC | $\phi_{EE}$ | $\eta$ |
|---|---|---|---|---|---|---|---|---|---|---|---|---|---|
| Hall2019 | [179] | Casamino | Mutation | $2^5 = 32$ | 0.062 | 0.303 | 0.774 | 0.309 | 0.125 | -0.735 | -0.493 | 0.625 | 0.050 |
| Hall2019 | [179] | Glucose | Mutation | $2^5 = 32$ | 0.031 | 0.421 | 0.904 | 0.162 | 0.050 | -0.775 | -0.843 | 0.850 | 0.050 |
| Hall2019 | [179] | Milk | Mutation | $2^5 = 32$ | 0.062 | 0.392 | 0.876 | 0.276 | 0.050 | 0.089 | -0.625 | 0.775 | 0.062 |
| Hall2019 | [179] | NAG | Mutation | $2^5 = 32$ | 0.062 | 0.411 | 0.911 | 0.420 | 0.075 | -0.742 | -0.601 | 0.713 | 0.100 |
| Hall2019 | [179] | Rhamnose | Mutation | $2^5 = 32$ | 0.062 | 0.329 | 0.858 | 0.461 | 0.087 | -0.303 | -0.779 | 0.800 | 0.037 |
| Hall2019 | [179] | Trypsin | Mutation | $2^5 = 32$ | 0.094 | 0.261 | 0.753 | 0.432 | 0.163 | -0.785 | -0.640 | 0.650 | 0.113 |
| Whitlock2000 | [180] | - | Mutation | $2^5 = 32$ | 0.094 | 0.267 | 0.734 | 0.391 | 0.138 | -0.897 | -0.739 | 0.750 | 0.000 |
| deVisser2009 | [181] | - | Mutation | $2^5 = 32$ | 0.156 | 0.156 | 0.416 | -0.401 | 0.300 | -0.617 | -0.568 | 0.388 | 0.200 |
| daSilva2010 | [182] | CCR5 | Mutation | $2^5 = 32$ | 0.094 | 0.240 | 0.680 | 0.336 | 0.200 | -0.615 | -0.569 | 0.636 | 0.013 |
| daSilva2010 | [182] | CXCR5 | Mutation | $2^5 = 32$ | 0.094 | 0.137 | 0.469 | 0.215 | 0.197 | -0.476 | 0.112 | 0.544 | 0.000 |
| Sunden2015 | [183] | AP | Mutation | $2^5 = 32$ | 0.031 | 0.456 | 0.960 | 0.529 | 0.037 | -0.242 | -0.714 | 0.863 | 0.000 |
| Anderson2021 | [184] | MPH-CaPTM | Mutation | $2^5 = 32$ | 0.036 | 0.269 | 0.844 | 0.710 | 0.038 | -0.511 | -0.507 | 0.810 | 0.016 |
| Anderson2021 | [184] | MPH-CdPTM | Mutation | $2^5 = 32$ | 0.031 | 0.407 | 0.905 | 0.538 | 0.075 | -0.464 | -0.470 | 0.738 | 0.025 |
| Anderson2021 | [184] | MPH-CoPTM | Mutation | $2^5 = 32$ | 0.031 | 0.373 | 0.887 | 0.494 | 0.025 | -0.273 | -0.445 | 0.887 | 0.000 |
| Anderson2021 | [184] | MPH-CuPTM | Mutation | $2^5 = 32$ | 0.031 | 0.326 | 0.834 | 0.441 | 0.087 | -0.204 | -0.476 | 0.750 | 0.013 |
| Anderson2021 | [184] | MPH-MgPTM | Mutation | $2^5 = 32$ | 0.062 | 0.362 | 0.893 | 0.770 | 0.050 | -0.376 | -0.465 | 0.863 | 0.000 |
| Anderson2021 | [184] | MPH-MnPTM | Mutation | $2^5 = 32$ | 0.062 | 0.411 | 0.931 | 0.831 | 0.037 | -0.548 | -0.440 | 0.863 | 0.025 |
| Mira2015 | [185] | TEM-AMP | Mutation | $2^4 = 16$ | 0.688 | -0.672 | -0.723 | -0.222 | 1.000 | -1.000 | -0.368 | 0.067 | 0.200 |
| Mira2015 | [185] | TEM-AM | Mutation | $2^4 = 16$ | 0.125 | -0.008 | 0.382 | 0.351 | 0.083 | -0.989 | -0.169 | 0.594 | 0.719 |
| Mira2015 | [185] | TEM-CEC | Mutation | $2^4 = 16$ | 0.188 | 0.065 | 0.407 | 0.259 | 0.333 | -0.848 | 0.038 | 0.406 | 0.000 |
| Mira2015 | [185] | TEM-CTX | Mutation | $2^4 = 16$ | 0.250 | 0.002 | 0.271 | -0.260 | 0.375 | -0.544 | -0.184 | 0.312 | 0.000 |
| Mira2015 | [185] | TEM-ZOX | Mutation | $2^4 = 16$ | 0.125 | 0.015 | 0.313 | -0.401 | 0.333 | -0.715 | -0.614 | 0.344 | 0.344 |
| Mira2015 | [185] | TEM-CXM | Mutation | $2^4 = 16$ | 0.125 | 0.090 | 0.537 | 0.304 | 0.167 | -0.721 | -0.683 | 0.594 | 0.000 |
| Mira2015 | [185] | TEM-CRO | Mutation | $2^4 = 16$ | 0.250 | -0.015 | 0.192 | -0.306 | 0.250 | -0.599 | -0.161 | 0.406 | 0.031 |
| Mira2015 | [185] | TEM-AMC | Mutation | $2^4 = 16$ | 0.875 | -0.938 | -1.000 | NaN | 0.000 | NaN | -0.214 | 0.000 | 0.000 |
| Mira2015 | [185] | TEM-CAZ | Mutation | $2^4 = 16$ | 0.688 | -0.743 | -0.953 | NaN | 1.000 | 0.408 | -0.160 | 0.000 | 0.143 |
| Mira2015 | [185] | TEM-CTT | Mutation | $2^4 = 16$ | 0.312 | -0.142 | -0.212 | 0.228 | 0.375 | -0.949 | -0.107 | 0.250 | 0.000 |
| Mira2015 | [185] | TEM-SAM | Mutation | $2^4 = 16$ | 0.062 | 0.101 | 0.701 | 0.292 | 0.048 | -0.953 | -0.430 | 0.677 | 0.645 |
| Mira2015 | [185] | TEM-CPR | Mutation | $2^4 = 16$ | 0.188 | 0.018 | 0.259 | -0.242 | 0.292 | -0.875 | -0.268 | 0.375 | 0.000 |
| Mira2015 | [185] | TEM-CPD | Mutation | $2^4 = 16$ | 0.125 | 0.116 | 0.515 | -0.245 | 0.125 | -0.617 | -0.445 | 0.562 | 0.000 |
| Mira2015 | [185] | TEM-TZP | Mutation | $2^4 = 16$ | 0.125 | 0.180 | 0.745 | 0.468 | 0.125 | -0.903 | -0.545 | 0.719 | 0.688 |
| Mira2015 | [185] | TEM-FSP | Mutation | $2^4 = 16$ | 0.250 | -0.202 | -0.437 | -0.241 | 0.458 | -0.963 | -0.499 | 0.219 | 0.000 |
| Meini2015 | [186] | - | Mutation | $2^4 = 16$ | 0.062 | 0.209 | 0.798 | 0.501 | 0.000 | -0.682 | -0.627 | 0.741 | 0.000 |

**Table A3 – continued from previous page**

| Author | Ref. | SubID | Space | Size | $\phi_{lo}$ | $\rho_a$ | NFC | $\gamma$ | $\epsilon_{reci}$ | $\epsilon_{DR}$ | FDC | $\phi_{EE}$ | $\eta$ |
|---|---|---|---|---|---|---|---|---|---|---|---|---|---|
| Lozovsky2009 | [187] | P. falciparum | Mutation | $2^4 = 16$ | 0.125 | -0.009 | 0.111 | -0.103 | 0.167 | -0.875 | -0.736 | 0.562 | 0.031 |
| Jiang2013 | [188] | P. vivax | Mutation | $2^4 = 16$ | 0.062 | 0.212 | 0.762 | 0.013 | 0.125 | -0.927 | -0.652 | 0.625 | 0.000 |
| Ogbunugafor2022 | [189] | Pyrimethamine | Mutation | $(2^4) \times 12$ | 0.200 | -0.027 | 0.235 | 0.251 | 0.333 | -0.884 | -0.187 | 0.321 | 0.071 |
| Ogbunugafor2022 | [189] | Cycloguanil | Mutation | $(2^4) \times 12$ | 0.200 | -0.022 | 0.235 | 0.250 | 0.333 | -0.884 | -0.187 | 0.321 | 0.071 |
| Weinreich20016 | [5] | Cefotaxime | Mutation | $2^5 = 32$ | 0.031 | 0.363 | 0.821 | 0.149 | 0.049 | -0.484 | -0.728 | 0.682 | 0.000 |
| Khan2011 | [41] | DM25 | Mutation | $2^5 = 32$ | 0.062 | 0.253 | 0.652 | -0.057 | 0.087 | -0.614 | -0.724 | 0.700 | 0.287 |
| Flynn2013 | [190] | DM25-EGTA | Mutation | $2^5 = 32$ | 0.094 | 0.214 | 0.597 | 0.202 | 0.087 | -0.156 | -0.608 | 0.662 | 0.175 |
| Flynn2013 | [190] | DM25-guanazole | Mutation | $2^5 = 32$ | 0.094 | 0.212 | 0.597 | 0.202 | 0.087 | -0.156 | -0.608 | 0.662 | 0.175 |
| Chou2011 | [38] | - | Mutation | $2^4 = 16$ | 0.062 | 0.310 | 0.993 | 0.570 | 0.000 | -0.742 | -0.836 | 1.000 | 0.000 |
| Malcolm1990 | [191] | Diploid | Mutation | $2^3 = 8$ | 0.125 | 0.013 | 0.961 | 0.333 | 0.000 | 0.085 | -0.951 | 0.917 | 0.000 |
| Guerrero2019 | [192] | C-muri-GroEL | Mutation | $2^3 = 8$ | 0.250 | -0.062 | 0.571 | 0.333 | 0.167 | -0.992 | -0.676 | 0.583 | 0.750 |
| Guerrero2019 | [192] | C-muri-LON | Mutation | $2^3 = 8$ | 0.250 | -0.227 | -0.150 | -0.333 | 0.333 | -0.976 | -0.025 | 0.333 | 0.500 |
| Guerrero2019 | [192] | C-muri-WT | Mutation | $2^3 = 8$ | 0.250 | -0.373 | -0.864 | -0.333 | 0.333 | -0.733 | -0.150 | 0.250 | 0.083 |
| Guerrero2019 | [192] | E.coli-GroEL | Mutation | $2^3 = 8$ | 0.250 | -0.266 | -0.218 | -0.333 | 0.333 | -0.964 | -0.401 | 0.333 | 0.250 |
| Guerrero2019 | [192] | E.coli-LON | Mutation | $2^3 = 8$ | 0.125 | -0.180 | -0.286 | -0.333 | 0.000 | -0.943 | -0.476 | 0.417 | 0.333 |
| Guerrero2019 | [192] | E.coli-WT | Mutation | $2^3 = 8$ | 0.375 | -0.358 | -0.403 | -1.000 | 0.667 | -0.744 | -0.175 | 0.083 | 0.250 |
| Guerrero2019 | [192] | L-grayi-GroEL | Mutation | $2^3 = 8$ | 0.250 | -0.337 | -0.505 | -0.333 | 0.333 | -0.978 | -0.601 | 0.333 | 0.500 |
| Guerrero2019 | [192] | L-grayi-LON | Mutation | $2^3 = 8$ | 0.250 | -0.263 | -0.110 | 1.000 | 0.333 | -0.999 | -0.200 | 0.333 | 0.167 |
| Guerrero2019 | [192] | L-grayi-WT | Mutation | $2^3 = 8$ | 0.375 | -0.281 | -0.173 | -1.000 | 0.667 | -0.981 | -0.300 | 0.083 | 0.833 |

Table A4: Selected landscape features for ProteinGym tasks with mean mutation depth > 1.

| Dataset | Size | Avg. Mutation | $\phi_{\text{lo}}$ | $\rho_a$ | NFC | $I_d$ | $\epsilon$_reci | FDC | $\phi_{\text{EE}}$ | $\eta$ |
|---|---|---|---|---|---|---|---|---|---|---|
| PIN1_HUMAN_Tsuboyama_2023_1I6C | 802 | 1.145 | 0.050 | -0.047 | 0.806 | 0.157 | -0.664 | -0.330 | 0.116 | 0.022 |
| RAD_ANTMA_Tsuboyama_2023_2CJJ | 912 | 1.151 | 0.064 | -0.054 | 0.800 | 0.195 | -0.595 | -0.524 | 0.064 | 0.017 |
| RCD1_ARATH_Tsuboyama_2023_5OAO | 1261 | 1.216 | 0.048 | -0.026 | 0.696 | 0.182 | -0.777 | -0.411 | 0.204 | 0.016 |
| RD23A_HUMAN_Tsuboyama_2023_1IFY | 1019 | 1.217 | 0.044 | 0.026 | 0.748 | 0.064 | -0.806 | -0.089 | 0.261 | 0.015 |
| SRBS1_HUMAN_Tsuboyama_2023_2O2W | 1556 | 1.222 | 0.042 | 0.038 | 0.849 | 0.033 | -0.731 | -0.532 | 0.357 | 0.016 |
| PSAE_PICP2_Tsuboyama_2023_1PSE | 1579 | 1.228 | 0.042 | 0.028 | 0.830 | 0.153 | -0.514 | 0.515 | 0.303 | 0.022 |
| RPC1_BP434_Tsuboyama_2023_1R69 | 1459 | 1.230 | 0.045 | 0.010 | 0.823 | 0.091 | -0.697 | -0.495 | 0.316 | 0.011 |
| RL20_AQUAE_Tsuboyama_2023_1GYZ | 1461 | 1.233 | 0.041 | 0.036 | 0.907 | 0.040 | -0.871 | -0.680 | 0.347 | 0.022 |
| TNKS2_HUMAN_Tsuboyama_2023_5JRT | 1479 | 1.244 | 0.041 | 0.050 | 0.778 | 0.049 | -0.783 | -0.226 | 0.371 | 0.015 |
| UBR5_HUMAN_Tsuboyama_2023_1I2T | 1453 | 1.247 | 0.039 | 0.051 | 0.786 | 0.053 | -0.789 | -0.374 | 0.367 | 0.018 |
| NUSG_MYCTU_Tsuboyama_2023_2MI6 | 1380 | 1.262 | 0.040 | 0.063 | 0.773 | 0.045 | -0.768 | -0.157 | 0.409 | 0.015 |
| RBP1_HUMAN_Tsuboyama_2023_2KWH | 1332 | 1.268 | 0.041 | 0.052 | 0.725 | 0.071 | -0.697 | -0.200 | 0.395 | 0.015 |
| RFAH_ECOLI_Tsuboyama_2023_2LCL | 1326 | 1.269 | 0.041 | 0.045 | 0.650 | 0.102 | -0.351 | 0.155 | 0.380 | 0.017 |
| SPG2_STRSG_Tsuboyama_2023_5UBS | 1451 | 1.291 | 0.040 | 0.049 | 0.823 | 0.042 | -0.589 | -0.566 | 0.404 | 0.013 |
| CATR_CHLRE_Tsuboyama_2023_2AMI | 1903 | 1.296 | 0.040 | 0.019 | 0.814 | 0.034 | -0.765 | -0.141 | 0.324 | 0.012 |
| SAV1_MOUSE_Tsuboyama_2023_2YSB | 965 | 1.296 | 0.048 | -0.016 | 0.768 | 0.212 | -0.659 | -0.223 | 0.326 | 0.023 |
| CBPA2_HUMAN_Tsuboyama_2023_1O6X | 2068 | 1.344 | 0.036 | 0.090 | 0.890 | 0.043 | -0.660 | -0.681 | 0.455 | 0.015 |
| FECA_ECOLI_Tsuboyama_2023_2D1U | 1886 | 1.354 | 0.038 | 0.092 | 0.743 | 0.063 | -0.476 | -0.202 | 0.478 | 0.012 |
| NUSA_ECOLI_Tsuboyama_2023_1WCL | 2028 | 1.356 | 0.035 | 0.104 | 0.863 | 0.041 | -0.474 | -0.541 | 0.497 | 0.018 |
| EPHB2_HUMAN_Tsuboyama_2023_1F0M | 1960 | 1.368 | 0.035 | 0.086 | 0.894 | 0.064 | -0.600 | -0.651 | 0.485 | 0.012 |
| CUE1_YEAST_Tsuboyama_2023_2MYX | 1580 | 1.396 | 0.035 | 0.088 | 0.782 | 0.089 | -0.641 | 0.118 | 0.478 | 0.013 |
| ODP2_GEOSE_Tsuboyama_2023_1W4G | 1134 | 1.410 | 0.043 | 0.096 | 0.782 | 0.084 | -0.808 | 0.227 | 0.461 | 0.022 |
| TCRG1_MOUSE_Tsuboyama_2023_1E0L | 1058 | 1.413 | 0.036 | 0.069 | 0.879 | 0.155 | -0.487 | -0.646 | 0.441 | 0.021 |
| PR40A_HUMAN_Tsuboyama_2023_1UZC | 2033 | 1.428 | 0.032 | 0.132 | 0.904 | 0.142 | -0.527 | -0.713 | 0.443 | 0.021 |
| BCHB_CHLTE_Tsuboyama_2023_2KRU | 1572 | 1.434 | 0.033 | 0.120 | 0.794 | 0.038 | -0.821 | -0.395 | 0.586 | 0.012 |
| SR43C_ARATH_Tsuboyama_2023_2N88 | 1583 | 1.438 | 0.031 | 0.127 | 0.865 | 0.037 | -0.600 | -0.522 | 0.572 | 0.011 |
| MBD11_ARATH_Tsuboyama_2023_6ACV | 2116 | 1.454 | 0.031 | 0.121 | 0.913 | 0.079 | -0.517 | -0.646 | 0.518 | 0.017 |
| DNJA1_HUMAN_Tsuboyama_2023_2LO1 | 2264 | 1.463 | 0.029 | 0.149 | 0.916 | 0.057 | -0.508 | -0.714 | 0.571 | 0.015 |
| MAFG_MOUSE_Tsuboyama_2023_1K1V | 1429 | 1.467 | 0.030 | 0.102 | 0.838 | 0.100 | -0.558 | -0.593 | 0.536 | 0.019 |
| RCRO_LAMBD_Tsuboyama_2023_1ORC | 2278 | 1.475 | 0.028 | 0.150 | 0.876 | 0.054 | -0.608 | -0.481 | 0.586 | 0.019 |
| BBC1_YEAST_Tsuboyama_2023_1TG0 | 2069 | 1.476 | 0.031 | 0.127 | 0.779 | 0.070 | -0.510 | -0.336 | 0.563 | 0.010 |

**Table A4 – continued from previous page**

| Dataset | Size | Avg. Mutation | $\phi_{\text{lo}}$ | $\rho_a$ | NFC | $I_d$ | $\epsilon\_\text{reci}$ | FDC | $\phi_{\text{EE}}$ | $\eta$ |
|---|---|---|---|---|---|---|---|---|---|---|
| PITX2_HUMAN_Tsuboyama_2023_2L7M | 1824 | 1.486 | 0.033 | 0.121 | 0.829 | 0.089 | -0.654 | -0.397 | 0.529 | 0.013 |
| THO1_YEAST_Tsuboyama_2023_2WQG | 1279 | 1.487 | 0.033 | 0.109 | 0.828 | 0.075 | -0.513 | -0.551 | 0.583 | 0.012 |
| SPA_STAAU_Tsuboyama_2023_1LP1 | 2105 | 1.508 | 0.025 | 0.149 | 0.834 | 0.129 | -0.343 | -0.471 | 0.546 | 0.024 |
| YAIA_ECOLI_Tsuboyama_2023_2KVT | 1890 | 1.509 | 0.026 | 0.143 | 0.847 | 0.059 | -0.739 | -0.589 | 0.569 | 0.010 |
| ISDH_STAAW_Tsuboyama_2023_2LHR | 1944 | 1.516 | 0.030 | 0.153 | 0.772 | 0.053 | -0.791 | -0.292 | 0.614 | 0.014 |
| VILI_CHICK_Tsuboyama_2023_1YU5 | 2568 | 1.532 | 0.028 | 0.222 | 0.904 | 0.047 | -0.639 | -0.621 | 0.616 | 0.017 |
| NKX31_HUMAN_Tsuboyama_2023_2L9R | 2482 | 1.537 | 0.029 | 0.153 | 0.826 | 0.145 | -0.566 | -0.506 | 0.541 | 0.025 |
| DOCK1_MOUSE_Tsuboyama_2023_2M0Y | 2915 | 1.584 | 0.027 | 0.149 | 0.802 | 0.143 | -0.397 | -0.333 | 0.549 | 0.017 |
| CSN4_MOUSE_Tsuboyama_2023_1UFM | 3295 | 1.589 | 0.022 | 0.234 | 0.833 | 0.027 | -0.644 | -0.398 | 0.673 | 0.014 |
| CBX4_HUMAN_Tsuboyama_2023_2K28 | 2282 | 1.598 | 0.021 | 0.164 | 0.858 | 0.084 | -0.633 | -0.456 | 0.592 | 0.012 |
| OBSCN_HUMAN_Tsuboyama_2023_1V1C | 3197 | 1.621 | 0.023 | 0.194 | 0.891 | 0.070 | -0.511 | -0.561 | 0.654 | 0.013 |
| SPTN1_CHICK_Tsuboyama_2023_1TUD | 3201 | 1.672 | 0.018 | 0.286 | 0.868 | 0.035 | -0.583 | -0.482 | 0.698 | 0.011 |
| YNZC_BACSU_Tsuboyama_2023_2JVD | 2300 | 1.690 | 0.017 | 0.180 | 0.900 | 0.093 | -0.442 | -0.632 | 0.670 | 0.017 |
| UBE4B_HUMAN_Tsuboyama_2023_3L1X | 3622 | 1.691 | 0.022 | 0.177 | 0.795 | 0.073 | -0.578 | -0.170 | 0.646 | 0.012 |
| SDA_BACSU_Tsuboyama_2023_1PV0 | 2770 | 1.699 | 0.016 | 0.266 | 0.922 | 0.045 | -0.496 | -0.658 | 0.717 | 0.015 |
| MYO3_YEAST_Tsuboyama_2023_2BTT | 3297 | 1.713 | 0.019 | 0.212 | 0.825 | 0.074 | -0.374 | -0.405 | 0.709 | 0.011 |
| AMFR_HUMAN_Tsuboyama_2023_4G3O | 2972 | 1.724 | 0.021 | 0.264 | 0.848 | 0.081 | -0.665 | -0.414 | 0.682 | 0.010 |
| HECD1_HUMAN_Tsuboyama_2023_3DKM | 5586 | 1.777 | 0.015 | 0.288 | 0.898 | 0.084 | -0.427 | -0.455 | 0.680 | 0.018 |
| POLG_PESV_Tsuboyama_2023_2MXD | 5130 | 1.806 | 0.011 | 0.352 | 0.905 | 0.044 | -0.491 | -0.488 | 0.775 | 0.016 |
| DLG4_HUMAN_Faure_2021 | 6976 | 1.817 | 0.040 | 0.441 | 0.794 | 0.127 | -0.637 | -0.258 | 0.431 | 0.027 |
| RASK_HUMAN_Weng_2022_binding-DARPin_K55 | 24873 | 1.876 | 0.023 | 0.646 | 0.903 | 0.061 | -0.633 | -0.299 | 0.460 | 0.054 |
| RASK_HUMAN_Weng_2022_abundance | 26012 | 1.882 | 0.027 | 0.487 | 0.774 | 0.107 | -0.558 | -0.123 | 0.441 | 0.028 |
| A4_HUMAN_Seuma_2022 | 14811 | 1.946 | 0.032 | 0.365 | 0.785 | 0.106 | -0.710 | -0.206 | 0.562 | 0.005 |
| YAP1_HUMAN_Araya_2012 | 10075 | 1.964 | 0.094 | 0.264 | 0.639 | 0.127 | 0.245 | -0.242 | 0.598 | 0.017 |
| PABP_YEAST_Melamed_2013 | 37708 | 1.969 | 0.038 | 0.446 | 0.856 | 0.106 | -0.441 | -0.145 | 0.631 | 0.059 |
| GRB2_HUMAN_Faure_2021 | 63366 | 1.984 | 0.021 | 0.450 | 0.841 | 0.089 | -0.354 | -0.179 | 0.660 | 0.029 |
| Q6WV12_9MAXI_Sommermeyer_2022 | 31401 | 2.685 | 0.459 | 0.579 | 0.853 | 0.086 | -0.727 | -0.298 | 0.444 | 0.000 |
| D7PM05_CLYGR_Sommermeyer_2022 | 24515 | 3.038 | 0.547 | 0.437 | 0.705 | 0.058 | -0.778 | -0.476 | 0.350 | 0.002 |
| Q8WTC7_9CNID_Sommermeyer_2022 | 33510 | 3.055 | 0.499 | 0.540 | 0.818 | 0.098 | -0.745 | -0.241 | 0.458 | 0.000 |
| F7YBW8_MESOW_Aakre_2015 | 9192 | 3.575 | 0.008 | -0.214 | 0.925 | 0.241 | 0.376 | -0.374 | 0.609 | 0.581 |
| GFP_AEQVI_Sarkisyan_2016 | 51714 | 3.878 | 0.724 | 0.456 | 0.741 | 0.198 | -0.879 | -0.547 | 0.407 | 0.058 |
| CAPSD_AAV2S_Sinai_2021 | 42328 | 4.728 | 0.439 | 0.269 | 0.862 | 0.189 | -0.422 | -0.223 | 0.539 | 0.005 |
| F7YBW8_MESOW_Ding_2023 | 7922 | 5.426 | 0.783 | -0.353 | 0.714 | 0.203 | -0.204 | -0.683 | 0.320 | 0.051 |

| Dataset | Size | Avg. Mutation | $\phi_{\mathrm{lo}}$ | $\rho_a$ | NFC | $I_d$ | $\epsilon\_$reci | FDC | $\phi_{\mathrm{EE}}$ | $\eta$ |
|---|---|---|---|---|---|---|---|---|---|---|
| GCN4_YEAST_Staller_2018 | 2638 | 17.068 | 0.966 | -0.292 | -0.083 | 0.295 | -0.839 | -0.146 | 0.304 | 0.031 |

Table A5: Selected landscape features for RNAGym tasks with mean mutation depth > 1.

| Dataset | RNA Type | Size | $\phi_{\text{lo}}$ (%) | $\rho_a$ (%) | NFC | $I_d$ | $\epsilon_{\text{reci}}$ | FDC | $\phi_{\text{EE}}$ | $\eta$ |
|---|---|---|---|---|---|---|---|---|---|---|
| Andreasson2020 | ribozyme | 7343 | 0.078 | 0.080 | 0.404 | 2.748 | 0.043 | -0.243 | 0.515 | 0.766 |
| Beck2022 | ribozyme | 21321 | 0.022 | 0.498 | 0.917 | 0.335 | 0.123 | 0.081 | 0.522 | 0.092 |
| Domingo2018 | tRNA | 4175 | 0.021 | 0.127 | 0.777 | 0.628 | 0.182 | -0.506 | 0.681 | 0.076 |
| Guy2014 | tRNA | 25491 | 0.393 | 0.106 | 0.484 | 2.434 | 0.191 | -0.377 | 0.571 | 0.677 |
| Janzen2022 | ribozyme | 1953 | 0.031 | 0.129 | 0.742 | 0.770 | 0.097 | -0.285 | 0.627 | 0.004 |
| Janzen2022 | ribozyme | 1953 | 0.048 | 0.182 | 0.823 | 0.666 | 0.065 | -0.250 | 0.644 | 0.003 |
| Janzen2022 | ribozyme | 1953 | 0.041 | 0.176 | 0.516 | 1.434 | 0.087 | -0.017 | 0.617 | 0.001 |
| Janzen2022 | ribozyme | 1953 | 0.032 | 0.177 | 0.529 | 1.107 | 0.075 | -0.186 | 0.625 | 0.001 |
| Janzen2022 | ribozyme | 1953 | 0.049 | 0.170 | 0.695 | 0.915 | 0.100 | -0.226 | 0.609 | 0.002 |
| Ke2017 | mRNA | 5533 | 0.078 | 0.292 | 0.906 | 0.394 | 0.150 | -0.485 | 0.634 | 0.129 |
| Kobori2015 | ribozyme | 255 | 0.008 | -0.174 | 0.881 | 0.550 | 0.089 | -0.317 | 0.800 | 0.135 |
| Kobori2015 | ribozyme | 255 | 0.020 | -0.194 | 0.805 | 0.582 | 0.117 | -0.523 | 0.778 | 0.249 |
| Kobori2015 | ribozyme | 1023 | 0.041 | -0.239 | 0.407 | 0.895 | 0.339 | -0.148 | 0.422 | 0.957 |
| Kobori2016 | ribozyme | 10296 | 0.009 | 0.284 | 0.842 | 0.505 | 0.031 | -0.003 | 0.519 | 0.053 |
| Kobori2018 | ribozyme | 16383 | 0.016 | -0.025 | 0.722 | 0.767 | 0.227 | 0.027 | 0.671 | 0.080 |
| Li2016 | tRNA | 65536 | 0.356 | 0.051 | 0.357 | 0.707 | 0.095 | -0.578 | 0.411 | 0.033 |
| McRae2024 | ribozyme | 74942 | 0.758 | 0.195 | 0.754 | 0.512 | 0.055 | -0.510 | 0.427 | 0.006 |
| McRae2024 | ribozyme | 47503 | 0.439 | 0.363 | 0.750 | 0.304 | 0.107 | -0.387 | 0.499 | 0.006 |
| Peri2022 | ribozyme | 16383 | 0.001 | 0.113 | 0.955 | 0.472 | 0.066 | -0.387 | 0.799 | 0.763 |
| Roberts2023 | ribozyme | 33930 | 0.042 | 0.521 | 0.884 | 0.414 | 0.149 | 0.038 | 0.512 | 0.041 |
| Roberts2023 | ribozyme | 21321 | 0.022 | 0.495 | 0.919 | 0.332 | 0.087 | -0.242 | 0.521 | 0.077 |
| Roberts2023 | ribozyme | 9045 | 0.035 | 0.391 | 0.852 | 0.455 | 0.122 | -0.051 | 0.501 | 0.054 |
| Roberts2023 | ribozyme | 22578 | 0.033 | 0.483 | 0.906 | 0.440 | 0.172 | -0.172 | 0.520 | 0.065 |
| Roberts2023 | ribozyme | 10296 | 0.016 | 0.344 | 0.876 | 0.436 | 0.079 | -0.184 | 0.522 | 0.072 |
| Soo2021 | ribozyme | 63430 | 0.021 | 0.039 | 0.752 | 0.725 | 0.238 | -0.015 | 0.641 | 0.008 |
| Tome2014 | aptamer | 417 | 0.317 | 0.011 | 0.421 | 0.060 | 0.500 | 0.294 | 0.402 | 0.005 |
| Tome2014 | aptamer | 2652 | 0.049 | 0.392 | 0.831 | 0.236 | 0.078 | -0.122 | 0.362 | 0.018 |
| Zhang2020 | ribozyme | 111417 | 0.464 | 0.408 | 0.707 | 0.567 | 0.134 | -0.441 | 0.546 | 0.009 |
| Zhang2024 | ribozyme | 61393 | 0.492 | 0.461 | 0.656 | 0.612 | 0.239 | -0.284 | 0.530 | 0.026 |
| Zhang2024 | ribozyme | 69583 | 0.452 | 0.488 | 0.654 | 0.605 | 0.175 | -0.219 | 0.538 | 0.012 |
| Zhang2024 | ribozyme | 149710 | 0.450 | 0.222 | 0.387 | 0.885 | 0.143 | -0.268 | 0.539 | 0.066 |