# OpenReview forum: "Augmenting Biological Fitness Prediction Benchmarks with Landscapes Features from GraphFLA"
_NeurIPS.cc/2025/Datasets_and_Benchmarks_Track — NeurIPS 2025 Datasets and Benchmarks Track spotlight_

### Official Review · Reviewer_At6t · 2025-06-19

**Rating:** 6
**Confidence:** 3

**Summary:**

A popular way to understand and benchmark models of biological sequences is using their ability to predict the effects of mutations as measured in deep mutational scans. But different models perform dramatically differently on different DMS's. What are the features that effect model performance. Previous splits looked at the function being measured or the taxa of the host of the biomolecule of interest. These authors instead measure simple, interpretable statistics of the landscape that predict the performance of some models very accurately.

**Dataset Code Accessibility:**

Yes

**Dataset Code Comments:**

The data can be easily accessed on the github.

**Ethical Considerations:**

No, there are no or only very minor ethics concerns

**Final Justification:**

see comment below

**Limitations Weaknesses:**

1. How accurate are your estimate of landscape statistics? If you only had half the data, would you arrive at the same statistic for ruggedness for example? Are these metrics therefore useful metrics for predicting how accurate model predictions will be?

2. It seems in many cases in figure 3, model robustness to these landscape parameters seems to correlate strongly with mean performance. In this case, in practice, can we expect that stratifying by these metrics is useful for comparing models as opposed to just describing how "hard" a particular task is? This is partially considered in fig 5 but how common is a flip in performance?

3. [This paper](https://arxiv.org/html/2410.22296v1) proposes so-called "Erlich functions" to simulate challenging optimization landscapes. Could these functions be analyzed by the same metrics as in your Fig 6?

**Strengths Contributions:**

1. The authors create a method to efficiently calculate these landscape statistics.

2. The authors analyze the correlations of statistics with each other.

3. The authors show some statistics very strongly correlate with prediction performance for some models.

---

> ### Author Rebuttal · Authors · 2025-07-29
>
> ***Abbreviations:** **"W"** for Limitations & Weaknesses.*
>
> We sincerely thank the reviewer for their constructive comments and recognition of our work. We address each comments below with additional experiments.
>
> ---
> ### **W1-1: Robustness of Landscape Features to Missing Data**
>
> We thank the reviewer for this insightful question. We definitely agree that validating the robustness of our calculated features to incomplete data is crucial. To address this, we conducted a new set of experiments:
> - We generated Kauffman's *NK* landscapes of varying dimensions (n={5, 10, 15, 20}) and ruggedness levels (from k=0 to k=n-1).
> - For each landscape, we simulated missing data by randomly removing a fraction $\alpha$ of the variants, where $\alpha$ ranged from 10% to 50%.
> - We then applied GraphFLA to both the complete and incomplete landscapes to measure the resulting landscape features.
> - Each experimental setup was repeated 10 times with different random seeds to ensure reliable estimates.
>
> Due to space constraints, we present a representative case below for a moderately sized (n=15) and rugged (k=7) landscape. The trends observed are consistent across all tested configurations.
>
> As shown in Table 1 below, **most key landscape features are highly robust, showing minimal deviation from the ground truth even with up to 50% of the data missing**.
>
> The one exception is *global optima accessibility*, which systematically decreases as more data is removed. This is expected, as this feature measures the fraction of variants connected to the global optimum via accessible (i.e., fitness-increasing) paths. Removing intermediate variants can "break" these paths, thus lowering the measured accessibility. However, this effect is predictable and can be readily **corrected**. For instance, with 50% missing data, the corrected estimate is 0.3223 / (1-0.5) = 0.6446, which is very close to the true value of 0.6729.
>
> **Table 1.** Key landscape features for an *NK* landscape (n=15, k=7) with varying fractions of missing data.
>
> |Data Missing|Reciprocal sign epistasis|Global optima accessibility|Autocorrelation|FDC|
> |---|---|---|---|---|
> |0%(Ground Truth)|0.1885|0.6729|0.1151|-0.0313|
> |10%|0.1883|0.6203|0.0927|-0.0420|
> |20%|0.1884|0.5276|0.0771|-0.0337|
> |30%|0.1837|0.4965|0.0789|-0.0310|
> |40%|0.1829|0.4005|0.0620|-0.0302|
> |50%|0.1774|0.3223|0.0553|-0.0313|
>
> **`Action 1:`** We will use the additional page allowed for the camera-ready version to add these experiments in a new validation section (after Sec. 4.1) to further strengthen the paper's rigor.
>
> **`Action 2:`** We will add an appendix section providing clear guidelines on which features are sensitive to missing data and how to apply corrections where applicable.
>
> **`See Also:`** Please see our response to Reviewer 4bLZ (W1, W2) for further experiments demonstrating the robustness of GraphFLA's features to measurement noise and different experimental sampling strategies (e.g., random vs. combinatorial mutagenesis).
>
> ---
> ### **W1-2: Predicting Model Performance Using Landscape Features**
>
> The reviewer raises an great point about the predictive utility of our landscape features. To investigate this, we built multiple linear regression models from these features to predict model performance. We did not consider more advanced models to avoid over-fitting and to focus more on the feature quality rather than the modeling process.
>
> From the results in Table 2, we found that **simple linear models are already capable of providing strong fits to model performance data based on landscape features**. Specifically, the first two sets of landscapes are combinatorially complete, thus both the landscape features and model performance are measured in an unbiased manner, resulting in $R^2>0.9$. In addition, although most landscapes from ProteinGym and RNAGym are far from complete, we still observed a reasonably good fit to the performance data ($R^2>0.65$).
>
> These results imply that our landscape features hold the promise to enable dedicated performance predictors for biological fitness prediction tasks.
>
> **Table 2.** $R^2$ scores from multiple linear regression models using landscape statistics to predict model performance across four benchmark datasets.
>
> | Dataset | Target model(s) | R² Score |
> | :--- | :--- | :--- |
> | Combinatorial Library (ours) | Evo2 | 0.940 |
> | TF Binding Landscapes | Evo2 | 0.987 |
> | ProteinGym | Top-10 models | 0.73 ~ 0.87 |
> | RNAGym | All 11 models | 0.67 ~ 0.85 |
>
> **`Action 3:`** We will add these experiments to the Appendix and reference them at the end of Sec. 4.2. This will allow us to discuss this prominent application of our framework and its implications for model selection.
>
> ---
> ### **W2: Deeper Analysis on Landscape Features for Model Comparison**
>
> We understand that the reviewer is interested in seeing whether the results in Fig. 5, which demonstrated that our landscape features can effectively identify respective advantages of different models (i.e., serving as "stratifying metrics"), can generalize to broader scenarios.
>
> Fig. A12 on page 43 partly addressed this, but due to the scale (there are $3,916$ model *pairs* derived from the $88$ evaluated models in ProteinGym) it is infeasible to present all the results visually. To offer a global view of the discriminative power of GraphFLA features and address the reviewer's concern, we instead conducted a quantitative analysis.
>
> Our core idea is that, if a landscape feature is effective in distinguishing two models, then the slope of the regression line in Fig. 5 should *significantly* deviate from zero. In other words, as the landscape feature varies, the performance gap between two models tends to increase or decrease–eventually leading to rank "flip".
>
> This significance can be quantified by the *p*-value of the relationship, and we measured this for all pairs of models benchmarked in ProteinGym, with respect to 5 representative landscape features: $i)$ reciprocal sign epistasis, $ii)$ magnitude epistasis, $iii)$ FDC, $iv) $ neutrality, and $v)$ autocorrelation.
>
> Our results showed that, each of these 5 features alone exhibits a statistically significant (*p*<0.05) correlation with the pairwise performance gap in 24.4% to 32.6% of all 3,916 model pairs. Moreover, 79.8% pairs can be distinguished by at least one of these 5 features. When all 20 landscape features are considered, this increases to 93.6%.
>
> Therefore, **GraphFLA's landscape features can effectively highlight the respective advantage of models in specific types of landscape (addressing Q2, lines 33-34; Sec. 4.3), in addition to offer an overall "hardness" measure of tasks (addressing Q1, line 33; Sec. 4.2).**
>
> **`Action 4:`** We will include this experiment in Sec. 4.3 to serve as a stronger evidence to support our findings from Fig. 5.
>
> ---
> ### **W3: Could the Ehrlich Function be Analyzed by GraphFLA?**
>
> **Yes. GraphFLA is designed to analyze the fitness landscape of arbitrary combinatorial optimization problems, including the Ehrlich function mentioned by the reviewer.**
>
> At the highest level, this Ehrlich function is essentially a synthetic landscape generator like those well-known ones already implemented in the GraphFLA.problems module (e.g., Kauffman's NK landscape, see a full list in Appendix D)
>
> As with other landscape generators, this Ehrlich function comes with several parameters that can be tuned to customize the landscape. One of its key features is its resemblance to real-world biological problems, where many variants may be non-functional and are thus assigned a fitness of $-\infty$.
>
> We have implemented this function in GraphFLA during rebuttal (though we are not allowed to update the repository now), and analyzing it with GraphFLA is straightforward (see Coding Example 1). Results in Table 3 below shows that by varying those control parameters, the Ehrlich landscape instances can exhibit very different topographies. This is an desired property for a test function designed for stress-testing optimizers [1].
>
> This Ehrlich example reinforces the broad applicability of this framework as we discussed in Sec. 4.4.
>
> [1] K. Smith-Miles and M. Muñoz, Instance Space Analysis for Algorithm Testing: Methodology and Software Tools, *ACM Comput. Surv.* (2023)
>
> **Table 3.** Key landscape features for 5 different Ehrlich landscapes with dimension L=8 and vocabulary size V=4, and varying number of motifs (C), motif length (K), and quantization (Q).
>
> | Parameter | Reciprocal sign epistasis | Global optima accessibility | FDC |
> |---|---|---|---|
> | C=1,K=3,Q=3 | 0.6990 | 0.0045 | -0.2038 |
> | C=2,K=3,Q=3 | 0.3624 | 0.0156 | -0.0173 |
> | C=2,K=4,Q=4 | 0.3149 | 0.1536 | -0.0274 |
> | C=2,K=4,Q=1 | 0.9995 | 0.0021 | -0.0126 |
> | C=4,K=2,Q=1 | 0.9881 | 0.0018 | -0.0432 |
>
> **Coding Example 1.** Applying GraphFLA to analyze the landscape of the Ehrlich Function.
>
> ```
> from graphfla.problems import Ehrlich
> from graphfla.landscape import Landscape
> from graphfla.analysis import classify_epistasis
>
> # problem definition
> L = 8
> V = 4
> C = 2
> K = 4
> Q = 4
> problem = Ehrlich(n=L, v=V, c=C, k=K, q=Q)
>
> # data acquisition
> data = problem.get_data()
> X = data["config"].apply(pd.Series)
> fitness = data["fitness"]
> data_types = {col : "categorical" for col in X.columns}
>
> # landscape construction
> landscape = Landscape()
> landscape.build_from_data(X, fitness, data_types=data_types)
>
> # exemplar landscape analysis
> print(classify_epistasis(landscape))
> ```
>
> **`Action 5:`** We will include this Ehrlich function in GraphFLA.problems module in the next release. Appendix D ("Landscape Models") will be updated accordingly.

---

> > ### Comment · Reviewer_At6t · 2025-08-01
> > **Response**
> >
> > This is far more than I expected in a response, and all my concerns are thoroughly addressed, especially actions 1, 2! While other minor critiques and discussions are possible, this work is rigorous, open, thorough, and addresses a pressing need for systematizing benchmarks in protein design. I could easily see this being a mandatory reporting metric for new biological datasets / benchmarks and used to critique and compare in silico surrogates like Erlich functions. I've increased my score to a strong accept.

---

> > > ### Author Response · Authors · 2025-08-01
> > >
> > > We are truly grateful for the reviewer’s engagement with our work and for the time dedicated throughout the review process. Their constructive comments have greatly helped us to further improve the scientific rigor of our manuscript during the rebuttal phase.
> > >
> > > We are glad to hear that our additional experiments and proposed actions have fully addressed the reviewer’s concerns. We are especially encouraged by the reviewer’s strong recognition of the contribution and potential impact of our work.
> > >
> > > Indeed, it is our hope with this paper that landscape analysis—which has traditionally received more attention from the biology community—can also be widely embraced within the AI community as an effective, unique lens for interpreting and comparing biological models. We will continue to maintain and extend the GraphFLA package to further enhance its accessibility and utility for a broader audience.
> > >
> > > Once again, we sincerely thank the reviewer for their time, encouragement, and endorsement of our work!

---

### Official Review · Reviewer_4bLZ · 2025-07-01

**Rating:** 6
**Confidence:** 4

**Summary:**

The paper presents GraphFLA, an open-source Python framework for constructing and analyzing fitness landscapes from deep mutational scanning (DMS) data. It represents the sequence–fitness mapping as a directed, weighted graph and computes 20 biologically meaningful topological features across four dimensions: ruggedness, epistasis, navigability, and neutrality. GraphFLA is applied to over 5,300 landscapes, demonstrating how these features help explain why different predictive models perform variably across tasks. The authors also collect and release 155 combinatorially complete landscapes comprising 2.2 million variants, greatly enriching the community’s benchmarking resources. As a dataset/benchmark paper, it provides a comprehensive illustration and analysis of how and why the dataset was built. It also presents an excellent example code demonstration via Colab. The code structure is clear and clean. I strongly recommend this paper to be accepted!

**Additional Feedback:**

1. For the title, maybe *landscape features* is more appropriate than *landscapes features*?

**Dataset Code Accessibility:**

Yes

**Dataset Code Comments:**

The authors provided Huggingface dataset links, colab tutorials, as well as the GitHub code repo. I tested their implementation and checked the code structure; they all appear to be really good and easily runnable.

**Ethical Considerations:**

No, there are no or only very minor ethics concerns

**Final Justification:**

My concerns have been addressed, and I would like to increase my score.

**Limitations Weaknesses:**

1. While the authors emphasize “combinatorially complete” landscapes, real-world DMS often yields uneven coverage or noisy measurements. The paper’s limitations section briefly acknowledges this but lacks quantitative analyses of how missing or unreliable data affect feature estimates.
2. Many labs generate targeted libraries (e.g., single-site saturation) rather than full combinatorial mutagenesis. It remains unclear how GraphFLA performs on such sparsely populated graphs: Do feature values bias toward underestimation? Providing recommended corrections or confidence intervals would expand practical applicability.
3. The framework identifies associations between landscape features and predictor accuracy but stops short of causal inference. Incorporating multivariate analyses, such as partial correlation controlling for landscape size or employing structural equation modeling, could clarify which features truly drive performance differences.

**Strengths Contributions:**

1. GraphFLA curates 20 biologically interpretable features such as the fraction of local optima, reciprocal sign epistasis, fitness-distance correlation, and neutrality index, which jointly characterize landscape topology and mutation interactions.
2. Utilizing sparse graph representations and the igraph library, GraphFLA constructs landscapes in roughly 20 s with under 2 GB of memory, outperforming existing tools. This efficiency makes it practical to analyze thousands of landscapes in a single study.
3. The framework reads ProteinGym, RNAGym, and UniProbe CSVs out-of-the-box, enabling users to layer landscape feature analysis atop hundreds of published DMS experiments. Moreover, the authors contribute 155 new combinatorially complete landscapes, which cover DNA, RNA, and protein domains.
4. By correlating each landscape feature with Spearman ρ of fitness predictions across models, the paper reveals concrete patterns: high epistasis impairs zero-shot predictors, poor navigability traps greedy searches, and neutral networks influence transfer learning. These findings can directly guide model selection and architectural design.
5. All code, data, and analysis notebooks are publicly available. The code repo and Colab tutorial include explicit documentation of preprocessing steps, hyperparameters, and hardware configurations, making it reproducible.
6. The manuscript is well-written, and the tables and figures are captioned clearly.

---

> ### Author Rebuttal · Authors · 2025-07-28
>
> ***Abbreviations:** **"W"** for Limitations & Weaknesses*, ***"F"** for Additional Feedback.*
>
> We appreciate the reviewer’s thoughtful feedback and are truly encouraged by their strong recognition of our work. Notably, we found all three suggestions extremely useful, which have significantly helped us to improve the quality and scientific rigor of the paper. Below we respond to each point with additional experiment results.
>
> ---
> ### **W1: Robustness of Landscape Features to Missing or Noisy Data**
>
> We thank the reviewer for this constructive suggestion. We definitely agree that quantifying the framework's robustness to missing and noisy data is a crucial validation.
>
> To address this, we have conducted a new set of experiments on Kauffman's *NK* landscapes, which allow for precise control over landscape properties. We generated landscapes with varying dimensions ($n \in \{5, 10, 15, 20\}$) and ruggedness levels ($k \in [0, n-1]$).
>
> For each generated landscape, we performed two separate analyses:
> *   **Incomplete Data:** We created subsampled landscapes by randomly removing a fraction $\alpha \in \{0.1, 0.2, \dots, 0.5\}$ of the variants.
> *   **Noisy Data:** We introduced random noise drawn from a Gaussian distribution $\mathcal{N}(0, (\beta \sigma)^2)$ to each variant's fitness score. Here, $\sigma$ is the s.t.d. of the original fitness values, and we tested noise levels of $\beta \in \{0.01, 0.05, 0.1, 0.2\}$.
>
> We then computed the landscape features for each of these landscapes, repeating each experiment 10 times with different random seeds to ensure reliable results. Due to space constraints, we present representative results for a moderately sized ($n=15$) and rugged ($k=7$) landscape below; the conclusions are consistent across all tested configurations.
>
> **Table 1.** Key landscape features for an *NK* landscape ($n=15, k=7$) under varying levels of data incompleteness and noise. All results are averaged over 10 runs.
> ||Reciprocal sign epistasis|Global optima accessibility|Autocorrelation|FDC|
> |---|---|---|---|---|
> |groundtruth (complete)|0.1885|0.6729|0.1151|-0.0313|
> |incomplete_0.1|0.1883|0.6103|0.0927|-0.0420|
> |incomplete_0.2|0.1884|0.5276|0.0771|-0.0337|
> |incomplete_0.5|0.1774|0.3223|0.0553|-0.0313|
> |noisy_0.01|0.1889|0.6492|0.0965|-0.0314|
> |noisy_0.05|0.1896|0.6542|0.0927|-0.0317|
> |noisy_0.1|0.1921|0.6362|0.0966|-0.0319|
> |noisy_0.2|0.1984|0.6339|0.0867|-0.0414|
>
> The results demonstrate that most key landscape features are **highly robust to both data incompleteness and noise**. Even with 50% of the data missing or with noise equivalent to 20% of the fitness standard deviation, features like *reciprocal sign epistasis* and *FDC* remain stable.
>
> The one exception is *global optimum accessibility*, which, as expected, decreases as more data is removed. This feature measures the fraction of variants with an accessible (fitness-increasing) path to the global optimum. Missing data can "break" these paths by removing intermediate nodes, leading to an underestimation.
>
> However, this effect is predictable and can be partially corrected. By scaling the measured accessibility by the fraction of *remaining* data ($1-\alpha$), we can recover a reasonable estimate of the original value. For example, with 50% missing data ($\alpha=0.5$), the corrected value is $0.3223 / (1-0.5) = 0.6446$. With 25% missing data ($\alpha=0.25$), the corrected value is $0.5276 / (1-0.25) = 0.7035$. Both are close to the ground truth of 0.6729.
>
> **`Action 1:`** We will incorporate these new experiments into a dedicated validation section in the camera-ready version of the paper (using the additional page), placed after Sec. 4.1, to rigorously demonstrate the framework's robustness.
>
> **`Action 2:`** We will add a new section to the Appendix with comprehensive guidelines on the sensitivity of each feature to missing data and noise, including recommended correction methods where applicable.
>
> ---
> ### **W2: Robustness of Landscape Features to Sparsely Sampled Data**
>
> This is a really interesting aspect to discuss! In general, there are three common types of mutagenesis experiments that generate empirical landscape data (we have touched on this in Sec. 2, lines 98-106, but not deep as we are here at an AI venue, but we will amend this in the revised version):
>
> **1. Site-saturation mutagenesis (SSM)**
>
> Landscapes derived from SSM typically contain only single-mutant variants (i.e., local neighborhood) relative to a wild-type (WT) sequence. Consequently, it is quite straightforward to analyze the mutation effects at each site, and advanced landscape analysis as in GraphFLA is not necessarily needed. For example, there is no epistatic interaction information here since there are only single-mutants. We note that this is not a flaw of GraphFLA; rather, it is a result of the information volume carried by the dataset itself. Actually, this is what motivated us (and some recent works, e.g., [4,7]) to curate/generate combinatorial landscapes in the first place .
>
> **2. Random mutagenesis**
>
> This strategy often creates a library that is densely sampled near the WT sequence but sparse elsewhere. To simulate this scenario, we again used an *NK* landscape ($n=15, k=7$, with 32,768 total variants). We generated a sparse, biased library of 1,804 variants by applying a 10% per-site mutation rate to the global optimum sequence, which mimicks a typical experimental outcome.
>
> The results, shown in Table 2, demonstrate that the key landscape features remain remarkably stable, even when calculated on this much smaller, biased subset. We confirmed this finding across multiple repetitions and different landscape configurations.
>
> **Table 2.** Key landscape features for a complete *NK* landscape ($n=15, k=7$) versus a sparse landscape generated via simulated random mutagenesis.
> | |Base landscape|Random mutagenesis landscape|
> |---|---|---|
> |reciprocal sign epistasis|0.1842|0.1823|
> |global optima accessibility|0.7031|0.7246|
> |autocorrelation|0.1199|0.1208|
> |FDC|-0.0843|-0.0837|
>
> **3. Combinatorial mutagenesis**
>
> This strategy conducts sparse sampling within a combinatorially design. It is equivalent to the random removal of variants from our combinatorially complete landscapes. As shown in our response to W1 above, GraphFLA's feature calculations are highly robust to this form of uniform data incompleteness.
>
> **`Action 3:`** In the camera-ready version, we will integrate this new experiment on random mutagenesis libraries into the new validation section proposed in our response to W1.
>
> **`Action 4:`** We will expand Sec. 3 and the Discussion to provide a clearer overview of these different experimental paradigms and discuss GraphFLA's applicability and performance for each.
>
> ---
> ### **W3: Additional Correlation & Causal Analysis**
>
> Again, this is a very constructive advice and we really appreciate the reviewer for pointing this out.
>
> **1. Partial correlation analysis**
>
> To isolate the influence of topological features from the potential confounding effect of landscape size, we performed a partial correlation analysis on our combinatorially complete library.
>
> The results in Table 3 reveal that the strong correlations we initially reported remain largely unchanged after controlling for landscape size. This indicates that the observed relationships between landscape topology and model performance are robust and not merely an artifact of landscape size.
>
> **Table 3.** Comparison of simple Spearman's correlation with partial Spearman's correlation (controlling for landscape size).
> |Feature|Simple Spearman's corr.|Partial Spearman's corr.|
> |---|---|---|
> |reciprocal sign epistasis|-0.9126|-0.9123|
> |global idiosyncratic index|-0.8721|-0.8829|
> |FDC|-0.7582|-0.7552|
> |fraction of local optima|-0.7434|-0.7684|
> |r/s ratio|-0.7062|-0.7514|
> |global optima accessibility|0.3807|0.4078|
> |EE mutations|0.8677|0.8641|
> |autocorrelation|0.8446|0.8512|
> |neighbor-fitness corr.|0.7931|0.7928|
> |magnitude epistasis|0.7699|0.7651|
>
> This finding is further consolidated by our analysis of the 5,016 TF binding landscapes in Figure 4. Since all these landscapes are identical in size (32,896 variants), landscape size is naturally controlled for, yet the clear patterns linking landscape features to model performance persist.
>
> **2. Structural equation modeling (SEM)**
>
> Here, we used SEM to integrate key theories from evolutionary biology literature into a hypothesis for causal analysis. Specifically, based on existing literature, we hypothesized that:
> - Epistasis, ruggedness, and neutrality each have a direct impact on model performance.
> - Epistasis indirectly affects model performance by increasing landscape ruggedness (Sec. 2, line 85).
> - Increased landscape ruggedness reduces landscape navigability (Sec. 2, line 89).
>
> We then evaluated this hypothesis using data from our combinatorially complete library and representative features for each aspect. We evaluated model-data consistency to determine the support for proposed links and found good model-data fit ($\chi^2$ p-value > 0.05 and RMSEA < 0.05). The results also indicate strong support for all hypothesized causal pathways, as summarized in Table 4.
>
> **Table 4.** Parameter estimates and $p$-value from structural equation modeling (SEM)
>
> |Outcome|Predictor|Estimate|p-value|
> |---|---|---|---|
> |spearman corr.|reciprocal sign epistasis|-1.7341|0.0000e+00|
> |spearman corr.|neutrality|-0.2011|1.7382e-05|
> |spearman corr.|autocorr.|0.3719|2.9986e-08|
> |autocorr.|reciprocal sign epistasis|-1.5192|0.0000e+00|
> |global optima accessibility|autocorr.|-1.3598|1.2823e-03|
>
> **`Action 5:`** We will add the full partial correlation and SEM analyses to a new Appendix section. We will summarize these key findings and reference the appendix in the main text (Sec. 4.2).
>
> ---
> ### **F1: Suggestion on Paper Title**
>
> We thank the reviewer for this suggestion. We agree and will correct the title.

---

> > ### Comment · Reviewer_4bLZ · 2025-08-03
> >
> > Thanks for the rebuttal! My concern has been addressed clearly! I would like to increase my score and strongly recommend this paper getting accepted!

---

> > > ### Author Response · Authors · 2025-08-03
> > >
> > > Dear reviewer 4bLZ, we really appreciate your strong recognition of our work and all your efforts dedicated to this review process!
> > >
> > > We will incorporate all the proposed actions above in our rebuttal into the camera-ready version, and thank you so much for raising those constructive suggestions!

---

### Official Review · Reviewer_sGKy · 2025-07-02

**Rating:** 5
**Confidence:** 5

**Summary:**

This paper introduces GraphFLA, a framework augmenting biological fitness prediction benchmarks with 20 landscape features (covering ruggedness, epistasis, navigability, and neutrality). Key contributions include: (1) A scalable framework for multi-modal data (DNA/RNA/protein); (2) A data-driven feature set curated from 1,673 literatures; (3) 155 combinatorially complete landscapes with 2.2M sequences; (4) Empirical validation that features explain model performance (e.g., Evo2 struggles in landscapes with high reciprocal sign epistasis).

**Dataset Code Accessibility:**

Yes

**Ethical Considerations:**

No, there are no or only very minor ethics concerns

**Final Justification:**

This is a paper with high application value in the field of life sciences, and I recommend accepting it.

**Limitations Weaknesses:**

1. Lack of ablation studies to validate critical features (e.g., ϵ_reci, FDC). For instance, no comparison of model interpretability when removing ϵ_reci, hindering clarity on feature indispensability.
2. Neglects 3D structural information for proteins, relying solely on sequence mutations for higher-order epistasis. Integrating PDB data (e.g., in GB1 landscape) could better capture long-range epistatic effects.
3. ALDE-ZS shows robustness in epistatic landscapes but lacks quantification of computational cost-efficiency (e.g., reduction in experimental rounds). Need comparisons of convergence speed under varying ϵ_reci (e.g., 0.2 vs. 0.5).
4. No exploration of combining landscape features with GNNs (e.g., using ϵ_reci as edge weights). GNNs may better model high-order interactions in epistatic landscapes (e.g., Johnston 2024’s TrpB4).
5. Adopts traditional graph representation instead of hypergraphs (e.g., LLM4Hypergraph) for higher-order interactions (e.g., 3+ locus mutations). Hyperedges could better represent multi-component synergies in Polyomino assembly landscapes.

**Strengths Contributions:**

1. The first comprehensive feature set covering four fundamental landscape aspects, outperforming MAGALLEN (which focuses solely on ruggedness). Features (e.g., FDC, γ statistic) are literature-driven and computationally feasible, reproducing prior findings in 5,300+ landscapes (e.g., Papkou 2023’s DHFR landscape).
2. Graph representation and efficient neighborhood generation enable processing million-scale mutants (20s for 1M mutants vs. 5h for baselines). Compatibility with ProteinGym/RNAGym allows task-level analysis (e.g., VenusREM excels in landscapes with FDC < -0.5).
3. Reveals model-performance correlates: Models fail in landscapes with high reciprocal sign epistasis (ϵ_reci > 0.15) or low FDC (> 0). Guides directed evolution: MLDE-ZS shows robustness in epistatic landscapes, outperforming greedy DE.

---

> ### Author Rebuttal · Authors · 2025-07-29
>
> ***Abbreviations:** **"W"** for Limitations & Weaknesses.*
>
> We thank the reviewer for their detailed and constructive feedback. We are encouraged that the reviewer has found GraphFLA to be of high-quality and useful for yielding new insights. We also appreciate reviewer's several thoughtful suggestions from the predictive modeling perspective.
>
> Actually, viewing landscape analysis and modeling as *twin flames* is the primary motivation that brought us to this paper, it is thus especially valuable for us to receive advice from the modeling side.
>
> Below, we address each of the reviewer's comments in detail.
>
> ---
> ### **W1: Clarification on Ablation Study**
>
> We thank the reviewer for raising this point. Actually, we designed most of our analyses in Sec. 4 ("Results") specifically to understand the **independent contribution of each landscape feature, which directly aligns with the goal of an ablation study**.
>
> Our approach was to systematically isolate the effect of each feature from the outset. Instead of including all features in a single predictive model and then removing them, our core analyses examine each feature individually. For example:
>
> - Fig. 2d depicts the direct correlation between model performance and each of the 20 landscape features separately.
> - Figs 3, 5, and 6 are all fundamentally structured to visualize a performance metric against one specific feature at a time, thereby isolating its impact.
> - Fig. 4 explores pairwise feature interactions, but still visualizes the independent contribution of each feature along the axes.
>
> Since our methodology is centered on this feature-by-feature examination, it already provides the granular insights that an ablation study aims to deliver.
>
> **`See Also:`** To further investigate feature interdependence, we have also conducted a new causal analysis using partial correlation and structural equation modeling. We kindly refer the reviewer to our response to Reviewer 4bLZ (W3) for these additional results.
>
> ---
> ### **W2: Does Not Consider 3D Structural Information**
>
> We thank the reviewer for this insightful suggestion. To address the reviewer's concern, we would like to clarify that **GraphFLA in its current form can compute epistatic interactions at arbitrary (e.g., from pairwise up to the highest) order.**
>
> We fully understand and appreciate the reviewer's idea that incorporating additional information like 3D structures could benefit the capture of long-range, complex epistatsis. Yet, the context of this hypothesis is mostly **predictive modeling**, which aims to *learn* to predict the true fitness from input data. Intuitively, providing richer information beyond sequence mutations to the model can likely increase performance.
>
> However, the context in this paper is **landscape analysis**, which is centered on *data mining* on empirical sequence-fitness data, i.e., we already know the groundtruth fitness value for each sequence. Thus, using a generalized Walsh-Hadamard transform [1], GraphFLA can recover (which is about faithfully *report*, instead of *predict*) all epistatic interactions in the sequence-fitness mapping without other information.
>
> This can be done with the following code:
>
> ```
> from graphfla.analysis import walsh_hadamard_coefficient
> walsh_hadamard_coefficient(
>     landscape,    # the constructed landscape
>     max_order=4  # max order of interaction to analyze
> )
> ```
>
> That said, the reviewer raises an excellent point about the origin of these interactions. Our analysis reveals *what* interactions exist (e.g., positive or negative), *where* they occur (e.g., among which sites), and *how* strong they are. However, to understand *why* these specific interactions arise from biophysical principles, incorporating 3D structural data is indeed a crucial next step. We agree this is a promising avenue for future research that can build directly upon the precise landscape characterizations our work provides.
>
> [1] A. Faure *et al.*,  An extension of the Walsh-Hadamard transform to calculate and model epistasis in genetic landscapes of arbitrary shape and complexity, *PLoS Comput. Biol.* (2024)
>
> **`Action 1:`** We will add this discussion on the promise of co-analysis with other forms of data, e.g., 3D structural information, in Sec. 3 & 5.
>
> ---
> ### **W3: Convergence Speed of ALDE-ZS**
>
> We thank the reviewer for raising this interesting question about convergence speed. To address this, we conducted new experiments comparing ALDE-ZS against the standard ALDE. Specifically, we selected two combinatorially complete landscapes from the TrpB3 series ($20^3=8,000$ variants each) to control for size:
> - TrpB3E: A rugged landscape representing a challenging optimization task (($\epsilon_{\text{reci}}=0.316$, the highest in the TrpB3 series)
> - TrpB3I: A relatively smooth landscape representing an easier task (($\epsilon_{\text{reci}}=0.171$, the lowest in the TrpB3 series)
>
> While the reviewer suggested a landscape with $\epsilon_{\text{reci}}=0.5$, this is hardly seen in empirical landscapes. Even for a most rugged synthetic *NK* landscape with $n=15$ and $k=14$, the $\epsilon_{\text{reci}}$ would be around only $0.33$. For each experiment, we ran 10 iterations with a batch size of 96, for a total budget of 960 evaluations per run. All other settings match those in Appendix F.
>
> The results in Table 1 demonstrate the benefit of ALDE-ZS:
> - On the rugged TrpB3E landscape, warm-starting with the zero-shot predictor provides a significant performance boost in the initial iterations. It accelerates convergence by approximately 2 iterations, saving 192 fitness evaluations.
> - On the smoother TrpB3I landscape, ALDE-ZS still offers a consistent advantage across all iterations, though the speed-up is less pronounced, as the standard ALDE is already effective at finding high-fitness variants on this benign landscape.
>
> **Table 1.** ALDE-ZS vs. ALDE performance (fitness achieved, as a percentile, averaged across 30 repetitions) at each evaluation budget on the TrpB3E and TrpB3I landscapes.
>
> |Iteration|TrpB3E-ALDE|TrpB3E-ALDE-ZS|TrpB3I-ALDE|TrpB3I-ALDE-ZS|
> |---|---|---|---|---|
> |1|0.574±0.054|0.664±0.033|0.907±0.048|0.917±0.093|
> |2|0.649±0.056|0.732±0.089|0.953±0.026|0.963±0.045|
> |3|0.719±0.066|0.779±0.045|0.967±0.024|0.972±0.034|
> |4|0.758±0.082|0.903±0.124|0.980±0.021|0.983±0.029|
> |5|0.771±0.108|0.841±0.101|0.984±0.025|0.986±0.018|
> |6|0.815±0.108|0.875±0.122|0.992±0.001|0.994±0.009|
> |7|0.847±0.112|0.895±0.139|0.992±0.004|0.994±0.007|
> |8|0.875±0.090|0.903±0.065|0.993±0.002|0.995±0.003|
> |9|0.902±0.097|0.908±0.074|0.994±0.000|0.995±0.001|
> |10|0.907±0.105|0.915±0.091|0.994±0.000|0.995±0.000|
>
> **`Action 2:`** We will add this experiment on convergence speed in the Appendix and reference it Sec. 4.4.
>
> ---
> ### **W4: No Exploration of GNN for Further Enhancing Modeling**
>
> The reason we did not explore GNN-based modeling is that our paper's primary goal, as a submission to the DB Track, is to establish that *landscape features can be used to better interpret and compare model performance* (lines 43-44). Therefore, developing novel modeling methodologies, while important, falls outside this specific scope as well as the Call for Paper of the DB track, and we hope the reviewer could kindly understand this.
>
> We also want to clarify a technical point about the features. All 20 landscape metrics in GraphFLA, including reciprocal sign epistasis ($\epsilon_{\text{reci}}$), are global properties measured for an entire landscape. As such, they describe the landscape's overall character and cannot be used as individual edge weights in a graph.
>
> Our current GraphFLA framework already constructs graphs where edges are weighted by the fitness difference between variants, with edge direction indicating the path of adaptation, thus inherently captures local fitness gradients. We agree with the reviewer that this is well-suited for a GNN to learn from, and we believe this is a promising direction for future work that can directly leverage the datasets we collected and landscapes built with GraphFLA.
>
> **`Action 3:`** We will discuss how can our current graph representation of fitness landscapes could potentially benefit predictive modeling with GNNs in Sec. 3 & 5.
>
> **`Action 4:`** We will explicitly distinguish *global* and *local* landscape features in Sec. 3 to avoid confusion.
>
> ---
> ### **W5: Why Not Use Hypergraphs as Graph Representation?**
>
> We thank the reviewer for this insightful comment. We chose a simple graph representation because it is already capable to provide the necessary function, efficiency, and interpretability required for our goal of landscape analysis.
>
> Specifically, our design choice is based on three key factors::
> - **Sufficiency:** The simple graph model, where nodes are genotypes and edges are single mutations, is completely sufficient for our *analytical goals*. It contains all the information needed (as node/edge attributes) to compute complex landscape properties, including high-order epistasis via the Walsh-Hadamard transform [1]. Though hypergraphs can be useful for modeling purpose, they do not provide direct added value for landscape analysis.
> - **Efficiency:** This standard representation allows us to leverage highly optimized graph traversal algorithms (e.g., see an example in Appendix C.3.1), thereby making our analyses computationally efficient.
> - **Interpretability:** This model has been widely accepted in evolutionary biology since 1970 [2]. This makes our framework intuitive and accessible to domain experts.
>
> Albeit these, we understand the reviewer's point that hypergraphs could capture higher-order interactions more explicitly during predictive modeling. We see this as a prominent future direction for the modeling community, and we hope our landscape features will be a valuable resource for evaluating such methods.
>
> [2] J. Smith,  Natural selection and the concept of a protein space. *Nature* (1970)

---

### Official Review · Reviewer_4bbB · 2025-07-03

**Rating:** 5
**Confidence:** 2

**Summary:**

The manuscript describes GraphFLA, a Python library for analyzing the fitness function landscapes of biological sequences (RNA, DNA, protein etc.). Evaluating the topology/landscape of fitness functions within a neighborhood is important for downstream analysis tasks. There is currently no library for standard open-source implementations of many landscape analysis features (ruggedness, navigability, epistasis, and neutrality) and the manuscript attempts to fill this gap.

Experimental results highlight the computational speed improvement over prior implementations, evaluation of fitness landscapes of existing databases and replication of previously reported results in literature. Beyond main applications, GraphFLA can be used to directed evolution, i.e., finding high fitness variants, and fitness analysis in biological networks at other scales.

**Additional Feedback:**

While the authors present a very clearly documented and described library, I was unsure what would be the impact of release such a library? In the conclusion, authors state "GraphFLA augments current benchmarks like ProteinGym and RNAGym", but I did not understand what the augmentation/improvement is.

**Dataset Code Accessibility:**

Yes

**Dataset Code Comments:**

The library appears to be very easily accessible and well-documented.

**Ethical Considerations:**

No, there are no or only very minor ethics concerns

**Final Justification:**

I thank the authors for a comprehensive rebuttal. I believe the paper will be a valuable contribution to the conference.

**Limitations Weaknesses:**

It would be helpful if authors could briefly describe the downstream tasks that require fitness landscape analysis, this would help readers who are not familiar with the topic.

The survey at the end is very long, and it would be helpful to summarize key findings/trends for ease of reading.

**Strengths Contributions:**

It appears that the library addresses a needed gap in software for analysis of biological sequence fitness functions.

A survey of techniques used for fitness analysis is presented.

The library is already available open-source on GitHub, including sample use cases in Google Colab, and appears to be well documented.

The manuscript is very well-written and easy to navigate.

---

> ### Author Rebuttal · Authors · 2025-07-28
>
> ***Abbreviations:** **"W"** for Limitations & Weaknesses*, ***"F"** for Additional Feedback.*
>
> We thank the reviewer for their valuable time and constructive feedback. Their comments have helped us identify areas where the manuscript can be improved, particularly for readers less familiar with fitness landscape analysis. Below, we address each point in detail to clarify the applications and impact of GraphFLA.
>
> ---
> ### **W1 & F1: Applications of GraphFLA and Impact**
>
> GraphFLA offers high-impact applications for two key audiences: it empowers biologists to uncover fundamental evolutionary principles and enables computer scientists to perform nuanced, task-level interpretation and comparison of biological models.
>
> **1. Understanding Evolutionary Mechanisms**
>
> - ***Significance:*** Fundamentally, the fitness landscape serves as the conceptual sandbox where Darwinian evolution has unfolded for billions of years. Analysis on its topography can provide valuable insights into the course of adaptation and speciation [1-18].
> - ***Target audience:*** Evolutionary biologists studying evolutionary dynamics, genomicists investigating the effects and interactions of mutations, among many others.
> - ***The gap:*** A standard, efficient, and comprehensive framework for landscape analysis is missing. Consequently, researchers must frequently re-implement analyses from scratch, resulting in code that is fragmented across languages, difficult to reuse, and often sub-optimal in performance (Sec. 1, lines 45-51; Sec. 2, lines 92-97).
> - ***Our contribution:*** GraphFLA integrates over 20 distinct analysis methods (surveyed in Appendix. B, C) into a single, unified, and efficient framework. This empowers biologists to easily conduct large-scale meta-analyses across numerous datasets, enabling more general findings that was previously impractical.
> - ***Demonstration in paper:*** In Sec. 4.1, we showcase this capability by applying GraphFLA to analyze over 5,300 empirical fitness landscapes curated from hundreds of biological studies.
>
> **2. Augmenting Model Performance Interpretation (Sec. 4.2)**
>
> - ***Significance:*** Effective benchmarking is critical for advancing fitness prediction models. A key desiderata is to move beyond *aggregate* scores (which can be misleading, see an interesting example at lines 29-32) to answer a crucial question: "Why does a model perform well on one task but poorly on another?" (Q1, page 1). Answering this allows us to gain *task-level* understanding of a model's specific strengths and weaknesses, guiding future research.
> - ***Target audience:*** Computer scientists and AI researchers developing or evaluating fitness prediction models in the AI4Bio community.
> - ***The gap:*** Existing fitness prediction benchmarks like ProteinGym and RNAGym only provide minimal meta-info. regarding each task (Sec. 1, lines 35-37; Sec. 2, lines 112-114). This renders the benchmarking results are often a long spreadsheet with only task IDs and the corresponding performance scores (Fig. 1b). Without effective features characterizing the property of each task, it is impossible to interpret performance variations across tasks (e.g., *why my model achieved a score of 0.9 on task A but only 0.3 on task B?*)
> - ***Our contribution:*** GraphFLA directly addresses this gap by providing quantitative landscape features that serve as task-level metadata. By integrating these features, we **augment** (see Fig. 1d vs. 1b) existing benchmarks, transforming them from simple score tables (Fig. 1b) into rich resources with both scores and task characteristics (Fig. 1d). This finally enables researchers to correlate model performance with landscape properties to answer Q1 (Fig. 1e, top).
> - ***Demonstration in paper:*** In Sec. 4.2 (Figs. 2d, 3, 4), we show that GraphFLA features correlate strongly with model performance across 5,300+ landscapes. This analysis yields specific, nuanced insights—for example, that landscapes that are more rugged, epistatic, or neutral are significantly harder to predict. Such conclusions are unattainable with existing benchmarks alone.
>
> **3. Augmenting Model Comparison (Sec. 4.3)**
>
> - ***Significance:*** Beyond understanding a single model, robust benchmarking requires answering a comparative question: "Why does my model outperform a baseline on task A, but not on task B?" (Q2, page 1). Answering this is essential for identifying the specific advantages and disadvantages of different model architectures.
> - ***Target audience:*** Computer scientists and AI researchers developing or evaluating fitness prediction models in the AI4Bio community.
> - ***The gap:*** Current benchmarks are insufficient for this kind of analysis. Without task-level features, researchers can only observe that one model outperforms another on a given set of tasks, but not why. The underlying properties that differentiate these task sets remain a black box.
> - ***Our contribution:*** By augmenting (as in the previous point) benchmarks with GraphFLA's landscape features, we move beyond simple win/loss counts. We can now systematically analyze how the performance gap between two models changes as a function of landscape topography. This allows us to identify the specific types of landscapes where one model excels over another, directly addressing Q2 (Fig. 1e, bottom).
> - ***Demonstration in paper:*** We provide a clear example in Sec. 4.3 (Fig. 5), showing that the performance gap between models is strongly dependent on landscape characteristics. For instance, while ProSST outperforms VenusREM on benign landscapes (low reciprocal epistasis), its advantage disappears and reverses as the landscapes become more rugged and challenging. This level of granular insight is only possible with GraphFLA.
>
> **4. Broader Applications (Sec. 4.4)**
>
> To further illustrate the library's impact, we highlight three areas where GraphFLA can be applied beyond the core use cases of model interpretation and comparison:
>
> - **Beyond learning:** While the points above focus on interpreting *learning* models, GraphFLA is equally valuable for understanding *optimization* processes. For example, it can analyze the outcomes of directed evolution (Fig. 6), which are analogous to navigating the fitness landscape to find its highest peaks.
> - **Beyond DNA, RNA, and protein:** GraphFLA is designed to be data-agnostic. Its ability to handle arbitrary sequence data allows it to analyze landscapes beyond DNA, RNA, and protein, such as those describing microbial community function (Table A2, page 43).
> - **Beyond fitness prediction:** At the end of Sec. 4.4, we also discussed that GraphFLA can be applied to analyze other forms of landscapes, e.g., phenotype landscapes for RNA secondary structure or protein tertiary structure, with illustrations in Appendix E.
>
> **`Action 1:`** Based on the reviewer's valuable feedback, we will revise the final paragraphs of the Introduction (Sec. 1). The new text will more explicitly outline these applications to ensure that the broad impact and applications of GraphFLA is clear from the outset.
>
> ---
> ### **F1: What is the "Augmentation?"**
>
> Based on the previous descriptions, here we answer this question in a more direct way: Existing benchmarking leaderboards often only contain task IDs and performance scores of different models (Fig. 1b). Consequently, interpretations of these benchmarks are predominantly based on *averaged* scores across all tasks, which can be misleading (please kindly see an interesting statistic at lines 28-32).
>
> GraphFLA addresses this gap by treating each task in a benchmark not just as a list of tasks and scores, but as an empirical fitness landscape. Our library then analyzes the "shape" or topography of this landscape and computes a comprehensive suite of 20 features (summarized in Table 1). These features are not arbitrary statistics; they are well-established metrics from decades of evolutionary biology research that quantify fundamental landscape properties.
>
> By adding these landscape features as rich meta-data to each task in ProteinGym or RNAGym, GraphFLA transforms them from simple leaderboards (Fig. 1b) into powerful diagnostic tools (Fig. 1d, e). This augmentation allows researchers to answer two long-standing questions:
>
> 1. *Why did one model perform well on one set of tasks but poorly on another?* (Q1, line 33; Application 2 above)
> 2. *Why did one model outperform the baseline on task A, but not on task B?* (Q2, lines 33-34; Application 3 above)
>
> In this way, we augment existing benchmarks to fully leverage their impressive scale of tasks and broad array of evaluated models.
>
> **`Action 2:`** We will polish our writing (mostly in Sec. 1) to explicitly describe the "augmentation" so that general readers can better grasp the idea.
>
>
> ---
> ### **W2: Regarding the Lengthy Survey in the Appendix**
> We thank the reviewer for this valuable suggestion. Perhaps the "survey" that the reviewer referred to is Appendix C, where we provide details about each of the 20 landscape features included in GraphFLA. The reason that this section is quite long at this moment because is contains:
> 1. Introduction to each feature.
> 2. Interpretation of each feature.
> 3. Mathematical formulations of each feature.
> 4. Coding implementations of each feature, if applicable.
>
> Among these:
> - #1 and #2 are general contents suitable for non-expert, general readers who are interested in landscape analysis.
> - #3 and #4 are intended for domain experts and developers who want to dive deeper.
>
> Based on this, to enhance readability:
>
> **`Action 3:`** We will split Appendix C into two separate sections based on the grouping above, so that general readers will not be overwhelmed by the lengthy maths whilst experts can quickly locate the information they are looking for.

---

> > ### Comment · Reviewer_4bbB · 2025-08-06
> >
> > Thank you for the comprehensive rebuttal. In addition to the comments above, I'd recommend you provide some examples of common tasks that such a benchmark can be used for (please explain this in layman's terms to a novice reader who would want to get started with the proposed benchmark).
> > I believe the paper is would be a very valuable addition to the conference and will keep my high rating.

---

> > > ### Author Response · Authors · 2025-08-07
> > >
> > > Dear Reviewer 4bbB,
> > >
> > > Thank you so much for your time and efforts throughout the review and discussion phase, and we sincerely appreciate your high recognition of our work.
> > >
> > > We definitely agree that adding more accessible examples (as well as revising the writing of the paper as suggested in your initial comments) is critical for maximizing the impact of GraphFLA. We will implement this using the additional page allowed in the camera-ready version. More importantly, we will continue to expand and maintain online examples/tutorials on Google Colab so that users can better engage with the practical use cases of GraphFLA.
> > >
> > > Thank you again for these constructive comments, which have really helped us strengthen our work and make it accessible to more potential users!
> > >
> > > Best regards.
> > > The Authors

---

### Decision · Program_Chairs · 2025-09-18

**Decision:**

Accept (spotlight)

**Comment:**

This paper proposed GraphFLA which is a library for landscape analysis. The framework supports the analysis of fitness function landscapes of different biological sequences (RNA, DNA, proteins) and fills a gap since it seems there is no standard open-source library that offers this analysis. I was not familiar with landscape analysis before, but all the reviewers and me now agree that this is an important problem and a valuable contribution.
The paper also collected and released 155 combinatorially complete landscapes comprising 2.2 million variants, which is a valuable dataset and benchmark.  There was a good discussion and the authors did an excellent job in addressing the reviewer concerns in great detail.

===== FINAL UPDATE FROM DB Track PCs ====

The final decision for this paper has been taken by the program chairs after consultation with the SACs. All Senior Area Chairs have ranked papers according to the feedback from the AC during the review process. We decided to leave the original meta-review to reflect the opinion of the AC in light of the initial discussions with reviewers and SAC.